# Radially patterned cell behaviours during tube budding from an epithelium

**Yara E Sanchez-Corrales[1], Guy B Blanchard[2]\*, Katja Röper[1]\***

[1]MRC Laboratory of Molecular Biology, Cambridge, United Kingdom; [2]Department of Physiology, Development and Neuroscience, University of Cambridge, Cambridge, United Kingdom

**Abstract** The budding of tubular organs from flat epithelial sheets is a vital morphogenetic process. Cell behaviours that drive such processes are only starting to be unraveled. Using live-imaging and novel morphometric methods, we show that in addition to apical constriction, radially oriented directional intercalation of cells plays a major contribution to early stages of invagination of the salivary gland tube in the *Drosophila* embryo. Extending analyses in 3D, we find that near the pit of invagination, isotropic apical constriction leads to strong cell-wedging. Further from the pit cells interleave circumferentially, suggesting apically driven behaviours. Supporting this, junctional myosin is enriched in, and neighbour exchanges are biased towards the circumferential orientation. In a mutant failing pit specification, neither are biased due to an inactive pit. Thus, tube budding involves radially patterned pools of apical myosin, medial as well as junctional, and radially patterned 3D-cell behaviours, with a close mechanical interplay between invagination and intercalation.

DOI: https://doi.org/10.7554/eLife.35717.001

**\*For correspondence:**
gb288@cam.ac.uk (GBB);
kroeper@mrc-lmb.cam.ac.uk (KR)

**Competing interests:** The authors declare that no competing interests exist.

## Introduction

During early embryonic development, simple tissue structures are converted into complex organs through highly orchestrated morphogenetic movements. We use the formation of a simple tubular epithelial structure from a flat epithelial sheet as a model system to dissect the processes and forces that drive this change. Many important organ systems in both vertebrates and invertebrates are tubular in structure, such as lung, kidney, vasculature, digestive system and many glands. The formation of the salivary glands from an epithelial placode in the *Drosophila* embryo constitutes such a simple model of tubulogenesis (*Girdler and Röper, 2014*; *Sidor and Röper, 2016*). Each of the two placodes on the ventral side of the embryo (*Figure 1A*) consists of about 100 epithelial cells, and cells in the dorso-posterior corner of the placode begin the process of tube formation through constriction of their apical surfaces, leading to the formation of an invagination pit through which all cells eventually internalise (*Girdler and Röper, 2014*; *Sidor and Röper, 2016*).

Apical constriction, a cell behaviour of epithelial cells that can transform columnar or cuboidal cells into wedge-shaped cells and can thereby induce and assist tissue bending, has emerged as a key morphogenetic module utilised in many different events ranging from mesoderm invagination in flies, *Xenopus* and zebrafish to lens formation in the mouse eye (*Lee and Harland, 2007*; *Martin and Goldstein, 2014*; *Martin et al., 2009*; *Plageman et al., 2011*). Apical constriction relies on the apical accumulation of actomyosin, that when tied to junctional complexes can exert pulling forces on the cell cortex and thereby reduce apical cell radius (*Blanchard et al., 2010*; *Mason et al., 2013*). Two pools of apical actomyosin have been identified: junctional actomyosin, closely associated with apical adherens junctions, as well as apical-medial actomyosin, a highly dynamic pool underlying the free apical domain (*Levayer and Lecuit, 2012*; *Röper, 2015*).

**eLife digest** Tubes form many of the organs in the animal body, from lungs to kidneys to intestines; but how are these structures created during development? For example, the tube that composes the salivary gland of the fruit fly emerges from a flat patch of cells. First, a dimple develops in the cell layer and moves inwards to create the tube pit. Then, like water down a plughole, the rest of the cells flow towards this point and fold downwards into the tube. However, it is still unclear exactly which mechanisms drive this process.

Here, Sanchez-Corrales et al. combine microscopy and computational approaches to follow how cells behave in a fruit fly embryo as they build the future salivary gland. The results confirmed that the tube starts forming because of 'apical constriction': cells, which normally look like prisms, squeeze into a wedge shape. This helps the tissue to bend and create the tube pit.

Other mechanisms contribute to the extension of the tube by turning the flat surface of cells into a curved one. In particular, further away from the dimple, cells become tilted towards the cylinder as they move into it. Another process reshuffles how these cells are connected to each other – a mechanism known as neighbour exchange – which leads to an overall movement towards the dimple. As the tube develops, this creates an increasing number of smaller rings of cells around the pit, which helps the cells to form the walls of the cylinder.

Many developmental processes are similar across organs and even species, and a next step could be to explore whether the mechanisms described by Sanchez-Corrales et al. are also present outside of the fruit fly's salivary glands. If so, this could shed light on what happens when tubes fail to form correctly in an embryo, and on how we could create these structures in the laboratory.

DOI: https://doi.org/10.7554/eLife.35717.002

Another prominent cell behaviour during morphogenesis in all animals is cell intercalation, the directed exchange of neighbours, that is for instance the driving force behind events such as convergence and extension of tissues during gastrulation. Also during cell intercalation apical actomyosin activity is crucial to processes such as junction shrinkage and junction extension that underlie this cell behaviour (*Bertet et al., 2004*; *Collinet et al., 2015*; *Rauzi et al., 2010*; *Zallen and Wieschaus, 2004*). Importantly, all cell behaviours during morphogenesis require close coordination between neighbouring cells. This is achieved on the one hand through tight coupling of cells at adherens junctions, but also through coordination of actomyosin behaviour within groups of cells, often leading to seemingly supracellular actomyosin structures in the form of interlinked meshworks and cables (*Blankenship et al., 2006*; *Röper, 2012*, *2013*).

The coordination between cells at the level of adherens junctions as well as actomyosin organisation and dynamics allows a further important aspect of morphogenesis to be implemented: the force propagation across cells and tissues. There is mounting evidence from different processes in *Drosophila* that force generated in one tissue can have profound effects on morphogenetic behaviour and cytoskeletal organisation in another tissue. For instance, during germband extension in the fly embryo, the pulling force exerted by the invagination of the posterior midgut leads to both anisotropic cell shape changes in the germband cells (*Lye et al., 2015*) and also assists the junction extension during neighbour exchanges (*Collinet et al., 2015*). During mesoderm invagination in the fly embryo, anisotropic tension due to the elongated geometry of the embryo leads to a clear anisotropic polarisation and activity of apical actomyosin within the mesodermal cells (*Chanet et al., 2017*).

We have previously shown that in the salivary gland placode during early tube formation when the cells just start to invaginate, the placodal cells contain prominent junctional and apical-medial actomyosin networks (*Booth et al., 2014*). The highly dynamic and pulsatile apical-medial pool is important for apical constriction of the placodal cells, and constriction starts in the position of the future pit and cells near the pit continue to constrict before they invaginate (*Booth et al., 2014*). The GPCR-ligand Fog is important for apical constriction and myosin activation in different contexts in the fly (*Kerridge et al., 2016*; *Manning et al., 2013*), and *fog* expression is also clearly upregulated in the salivary gland placode downstream of two transcription factors, Fkh and Hkb (*Chung et al., 2017*; *Myat and Andrew, 2000b*). Fkh is a key factor expressed in the placode

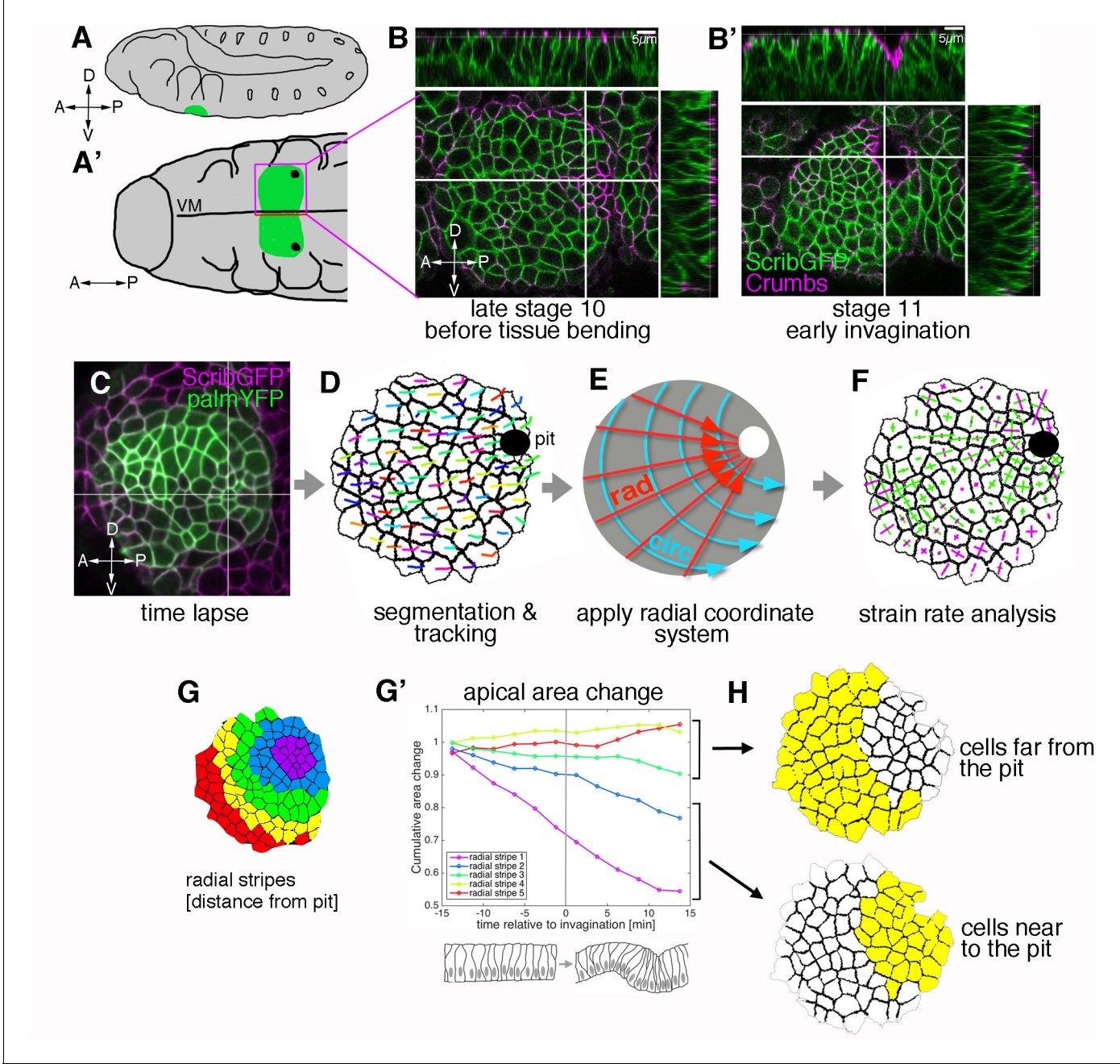

**Figure 1.** Morphogenetic events in the salivary gland placode show a radial organisation. (A, A'). Schematic of a stage 11 *Drosophila* embryo highlighting the position of the salivary gland placode (green) in lateral (A) and ventral (A') views; A: anterior, P: posterior; D: dorsal; V: ventral. (B, B'). Surface and cross-section views of the salivary gland placode just prior to the first tissue bending (B) and once the initial pit of invagination has formed (B'). Lateral membranes are labelled by ScribbleGFP (green) and apical cell outlines by Crumbs (magenta). Scale bars are 5 µm. (C-F) Workflow of the morphometric analysis. Early salivary gland placode morphogenesis is recorded by time-lapse analysis using markers of cell outlines (C), cells are segmented and tracked over time (D; exemplary tracks of individual cells are indicated by coloured lines). Data are recorded and expressed in a radial coordinate system with the invaginating pit as the origin (E; 'rad' is the vectorial contribution radially towards the pit, 'circ' is the vectorial circumferential contribution). Various derived measures are projected onto the radial coordinate system. Here, example projected small domain deformation (strain) rates are shown (F; contraction in green, expansion in magenta; see *Figure 2B*). See also *Videos 1* and *2*. (G-H) Cumulative time-resolved analysis of apical area change for nine embryos (G'), the cumulative area strain is plotted, grouping the placodal cells into radial stripes (G) for the analysis. This reveals a clear split in behaviour between cells far from the pit (H, top; negligible area change) and cells near the pit (H, bottom; constriction, consistently negative area change). See also *Figure 1—figure supplement 1*.

DOI: https://doi.org/10.7554/eLife.35717.003

*Figure 1 continued on next page*

*Figure 1 continued*

The following source data and figure supplement are available for figure 1:

**Source data 1.** Apical area change in radial stripes.

DOI: https://doi.org/10.7554/eLife.35717.005

**Figure supplement 1.** Morphogenetic events in the salivary gland placode show a radial organisation.

DOI: https://doi.org/10.7554/eLife.35717.004

directly downstream of the homeotic factor Sex combs reduced (Scr) that itself is necessary and sufficient to induce gland fate (*Myat and Andrew, 2000a*; *Panzer et al., 1992*). *fkh* mutants fail to invaginate cells from the placode, with only a central depression within the placode forming over time (*Myat and Andrew, 2000a*).

Here, we use morphometric methods, in particular strain rate analysis (*Blanchard et al., 2009*), to quantify the changes occurring during early tube formation in the salivary gland placode. Many morphogenetic processes are aligned with the major embryonic axes of anterio-posterior and dorso-ventral. An excellent example is germband extension in the fly embryo, where polarised placement of force-generating actomyosin networks is downstream of the early anterio-posterior patterning cascade (*Blankenship et al., 2006*; *Simões et al., 2010*). In the case of the salivary gland placode, the primordium of the secretory cells that invaginate first is roughly circular, with an off-center focus due to the invagination point being located in the dorsal-posterior corner, prompting us to assess changes within a radial coordinate framework. In addition to previously characterised apical constriction (*Booth et al., 2014*; *Chung et al., 2017*), we demonstrate that circumferentially oriented directional intercalation of placodal cells plays a major contribution to ordered invagination at early stages.

In addition, we compare quantitative planar strain rate analysis at different apical-basal depths of the tissue, and link cells between the planes to calculate rates of change of local geometry in depth, as a proxy for a full 3D analysis. We uncover that cell geometries and behaviours in 3D are also radially patterned: near the pit of invagination, where apical-medial myosin II is strong (*Booth et al., 2014*), cells are isotropically constricting apically leading to apical cell wedging, and with distance from the pit cells progressively tilt towards the pit. Cells also interleave apically towards each other in a circumferential direction, which can lead to different neighbour connectivity along their basal-to-apical length, equivalent to a T1 transition in depth. This strongly suggests apically driven active intercalation behaviours. We further show that several measures of 'geometrical stress' have signatures indicating that circumferential intercalation in the cells away from the pit is active. In addition, across the placode junctional myosin II is enriched in circumferential junctions, suggesting polarised initiation of cell intercalation through active junction shrinkage. This is followed by polarised resolution of exchanges towards the pit, thereby contributing to tissue invagination. *forkhead* (*fkh*) mutants, that fail to form an invagination, still show cell intercalations within the placode at a high rate, further supporting the active nature of the intercalations. Thus, tube budding depends on a radial organisation of 3D cell behaviours, that are themselves patterned by the radially patterned and polarised activity of apical myosin II pools, with apical-medial myosin dominating near the invagination point and polarised junctional myosin dominating further away from the pit. The continued initiation of cell intercalation but lack of polarised resolution in the *fkh* mutant, where the invagination is lost, could

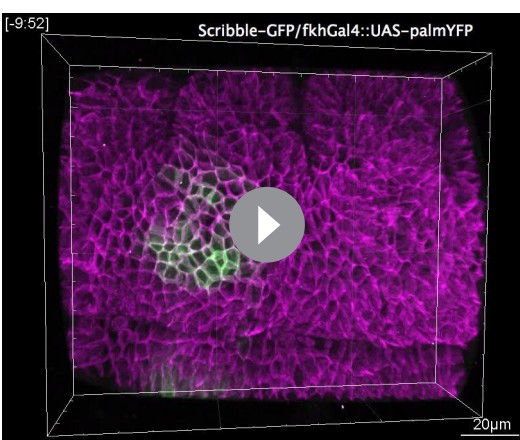

**Video 1.** Example movie of early salivary gland placode morphogenesis in 3D. Embryo of the genotype *Scribble-GFP/fkhGal4::UAS-palmYFP* as shown if *Figure 1C* and *Figure 3A*. Time stamp indicates time before and after initiation of tissue bending at t = 0. Scale bar 20 μm.

DOI: https://doi.org/10.7554/eLife.35717.007

**Table 1.** Number of cells in time-lapse experiments.

| Figure | Cells | Number of embryos | number of cells per time point at [−18, −13.5, −9, −4.5, 0, 4.5, 9, 13.5, 18] min |
|---|---|---|---|
| *Figure 2* | Apical cells near to pit | 9 | [123, 599, 724, 1132, 980, 737, 468, 219, 80] |
| *Figure 2* | Apical cells far from pit | 9 | [156, 837, 1183, 1880, 1731, 1484, 1151, 744, 319] |
| *Figure 3* | Basal cells near to pit | 6 | [260, 453, 726, 569, 546, 506, 360, 89] |
| *Figure 3* | Basal cells far from pit | 6 | [382, 762, 1309, 1200, 1145, 1133, 770, 247] |
| *Figure 4* | 3D proxy cells near to pit | 5 | [36, 307, 420, 479, 336, 410 260 148 58] |
| *Figure 4* | 3D proxy cells far from pit | 5 | [37, 350, 515, 652, 479, 564, 394, 243, 131] |
| *Figure 8* | Apical fkh6 cells near to pit | 5 | [27, 161, 248, 269, 215, 238, 233, 186, 85] |
| *Figure 8* | Apical fkh6 cells far from pit | 5 | [52, 343, 626, 735, 615, 706, 655, 550, 239] |
| Figure | Cells | Number of embryos | number of cells per time point at [−13.5, −9, −4.5, 0, 4.5, 9, 13.5, 18] min |
| *Figure 7* | Uni and Bi-polarity WT | 4 | [207, 522, 1115, 918, 1091, 1125, 1047, 484] |
| *Figure 9* | Uni- and Bi-polarity fkh6 | 5 | [528, 783, 815, 712, 809, 786, 708, 307] |

DOI: https://doi.org/10.7554/eLife.35717.006

suggest that a tissue-intrinsic mechanical interplay also contributes to successful tube budding.

## Results

### Apical cell constriction is organised in a radial pattern in the salivary gland placode

Upon specification of the placode of cells that will form the embryonic salivary gland at the end of embryonic stage 10, the first apparent change within the apical domain of placodal cells is apical constriction at the point that will form the first point of invagination or pit (*Figure 1A,B*; [*Booth et al., 2014*; *Myat and Andrew, 2000b*]). Apical constriction at this point in fact preceded actual tissue bending (*Figure 1B*). We have previously shown that apical constriction is clustered around the pit and is important for proper tissue invagination and tube formation (*Booth et al., 2014*). In order to discover if any further cell behaviours in addition to apical constriction contribute to tissue bending and tube invagination in the early placode, we employed quantitative morphometric tools to investigate the process in comparable wild-type time-lapse movies using strain rate analysis (*Blanchard et al., 2009*). We imaged embryos expressing a lateral plasma membrane label in all epidermal cells as well as a marker that allows identification of placodal cells (*Figure 1C* and *Video 1*; for genotypes see *Table 1*). Time-lapse movies were segmented and cells tracked using previously developed computational tools that allow for the curvature of the tissue to be taken into account (*Figure 1C–F*) (*Blanchard et al., 2009*; *Booth et al., 2014*). The cells of the salivary gland placode that will later form the secretory part of the gland are organised into a roughly circular patch of tissue prior to invagination and maintain this shape during the process (*Figure 1C*). Within this circular patch, the invagination pit is located at the dorsal-posterior edge rather than within the centre of the placode. We therefore employed a radial coordinate system, with the forming pit as its origin, in which to analyse and express any changes (*Figure 1E*). We locally projected 2D strain (deformation) rates and other oriented measures onto these radial ('rad') and circumferential ('circ') axes (*Figure 1F*).

We focused our analysis on the early stages of tissue bending and invagination, defining as t = 0 the time point before the first curvature change at the point of invagination at the tissue level could be observed. Dynamic analysis of the changes in apical area of placodal cells, in the time interval of 18 min prior to and 18 min after the first tissue bending, revealed distinct zones of apical cell behaviour. Grouping the placodal cells into ~ 2 cell-wide stripes concentric to the pit for apical area change analysis (*Figure 1G*; *Figure 1—figure supplement 1* and *Video 2*) revealed a split into cells whose apical area over this time interval did not change or changed only slightly (*Figure 1G'*, red, yellow and green lines) and cells whose apical area progressively decreased (*Figure 1G'*, blue and purple lines). The clear split into two zones of differing cell behaviours, defined by radial distance

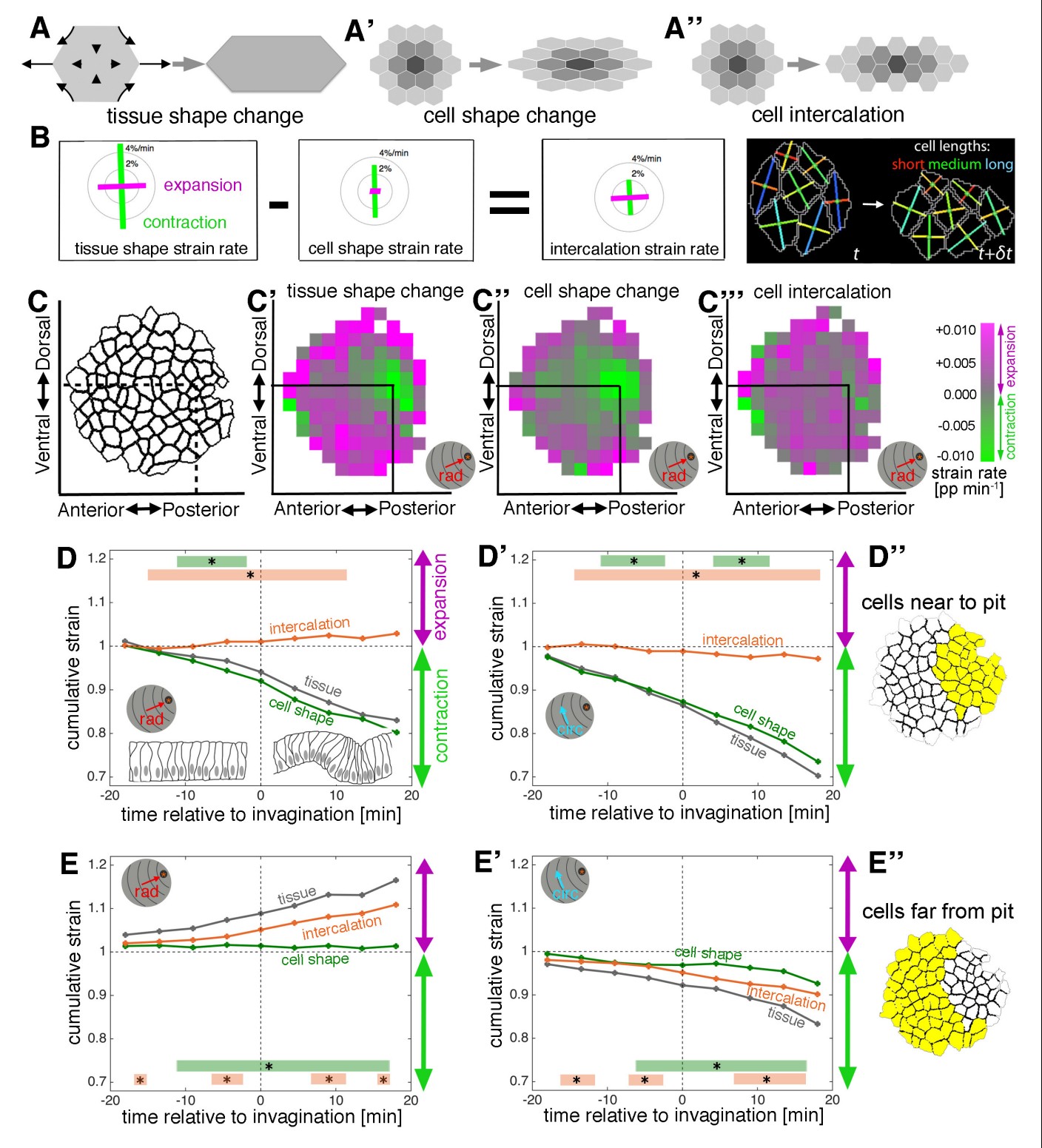

**Figure 2.** Apical strain rate analysis of early events during salivary gland placode invagination. (A–A'') During early salivary gland placode development the change in tissue shape (A) can be accounted for by cell shape changes (A'), cell intercalation (A'') or any combination of the two. During this phase, cell division and gain or loss of cells from the epithelium (other than cells invaginating into the pit) do not occur. (B) For small domains of a focal cell and its immediate neighbours, tissue shape change is the sum of two additive contributions, cell shape change and cell intercalation (*Blanchard et al., 2009*). Both tissue shape change and cell shape change can be measured directly from the segmented and tracked time-lapse movies, so the amount

*Figure 2 continued on next page*

*Figure 2 continued*

of cell intercalation can be deduced. Strain rates are depicted as orthogonal lines, with contraction in green and expansion in magenta, and the length of the line proportional to the rate of strain (grey circles mark 2 and 4% change per minute). The last panel shows the group of cells (central cell and first corona of neighbours) for which the example strain rates in (B) were calculated, with orthogonal lines indicating the lengths of the major and minor axes of the cells. The vertical contraction in the group of cells can be clearly seen as a combination of vertical cell shortening and continuous rearrangement/intercalation (evidenced by one topological change). (C–C''') Spatial maps summarising the strain rate analysis covering 18 min prior to 18 min post commencement of tissue bending. Mapped onto the shape of the placode (C) the strain rate contribution towards the pit ('rad') is shown, quantified from data from nine embryos (see *Figure 2—figure supplement 1*). (C') Tissue contraction (green) dominates near the pit, with expansion (magenta) towards the periphery. The tissue contraction near the pit is mostly due to cell constriction near the pit (C''), whereas the tissue expansion is contributed to by both cell expansion and cell intercalation far from the pit (magenta in C'' and C'''). Strain rates are given in proportion (pp) of change per minute. (D–E'') Regional breakdown of time-resolved cumulative strain. In the area near the pit (D'') tissue constriction dominates (grey curves in D and D') and is due to isotropic constriction at the cell level (green curves in D and D'), whilst intercalation only plays a minor role in this region (orange curves in D, D'). Far from the pit (E''), the tissue elongates towards the pit (E, grey curve), with a corresponding contraction circumferentially (E', grey curve), and this is predominantly due to cell intercalation (orange curves in E and E'). Statistical significance based on a mixed-effects model (see Materials and methods) and a $p<0.05$ threshold (calculated for instantaneous strain rates [see *Figure 2—figure supplement 1*]), is indicated by shaded boxes at the top or bottom of each panel (tissue vs cell shape is green and tissue vs intercalation is orange). See also *Video 3*. Cumulative strains here and in all subsequent plots are calculated as the exponent of the cumulative instantaneous strain rates, which are shown in Figure supplements.
DOI: https://doi.org/10.7554/eLife.35717.009

The following source data and figure supplements are available for figure 2:

**Source data 1.** Cumulative strain and statistics for cells near to the pit and far from the pit.
DOI: https://doi.org/10.7554/eLife.35717.012
**Figure supplement 1.** Apical strain rate analysis of early events during salivary gland placode invagination.
DOI: https://doi.org/10.7554/eLife.35717.010
**Figure supplement 1—source data 1.** Cumulative strain for all cells and instantaneous strain rates for cells near to the pit and far from the pit.
DOI: https://doi.org/10.7554/eLife.35717.011

from the invagination point at zero minutes (when the pit starts to invaginate), prompted us to analyse these regions independently in our strain rate analyses (*Figure 1H*).

## Both apical constriction and cell intercalation are prominent during the early stages of tube morphogenesis

Methods have been developed previously to calculate strain (deformation) rates for small patches of tissue, and to further decompose these into the additive contributions of the rates of cell shape change and of the continuous process of cell rearrangement (intercalation) (*Blanchard, 2017*; *Blanchard et al., 2009*) (*Figure 2A,B*). The levels of both contributions vary dramatically between different morphogenetic processes (*Blanchard et al., 2010*; *Bosveld et al., 2012*; *Butler et al., 2009*; *Etournay et al., 2015*; *Guirao et al., 2015*). Cell divisions have ceased in the placode around the time of specification, and there is also no cell death, therefore none of these processes contributed to the overall tissue deformation in this case. Upon segmentation of cell outlines, the rate of tissue deformation was calculated from the relative movement of cell centroids in small spatio-temporal domains of a central cell surrounded by its immediate neighbours and over three movie frames (~6 min). Independently, for the same domains, rates of cell shape change were calculated mapping best-fitted ellipses to actual cell shapes over time. The rate of cell intercalation could then be deduced as the difference between these two measures (*Figure 2A,B* and *Video 3*; see Materials and methods for a detailed description). The three types of strain rate measure were then projected onto our radial coordinate system.

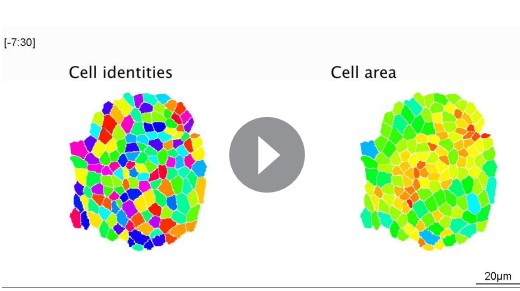

[-7:30]

Cell identities          Cell area

20μm

**Video 2.** Example movie showing evolution of apical cell identities and apical area. Movie ExpID0356 with 100% tracking, is shown. Cell identities are randomly colour-coded on the left, apical area on the right. Area colour codes are identical to *Figure 8C*, please refer to this scale. Time stamp indicates time before and after initiation of tissue bending at t = 0. Scale bar 20 μm.
DOI: https://doi.org/10.7554/eLife.35717.008

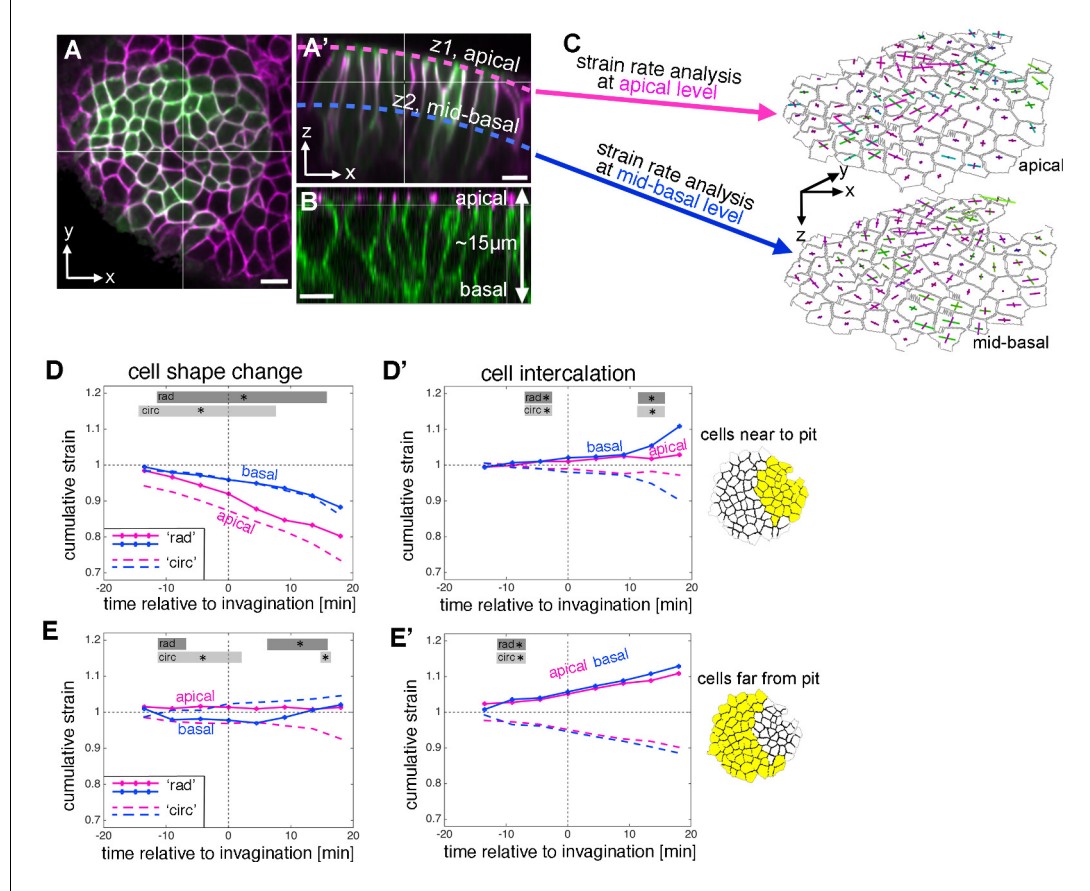

**Figure 3.** Strain rate analysis of early tubulogenesis using a 3D proxy. (A–B) Epithelial cells of the placode are about 2–5 µm in apical diameter but extend about 15 µm into the embryo along their apical-basal axis. To assess behaviour of the tissue and cells at depth, we analysed a mid-basal level, about 8 µm basal to the apical surface. Scale bars are 5 µm. (C) Illustration of the method used: we segmented and tracked cells at a depth of ~8 µm, and repeated the strain rate analysis at this mid-basal level (blue) to compare with apical (pink). (D–D') Comparison of apical (pink, as shown in *Figure 2*) and basal (blue) cumulative strains, for cell shape changes (D, E) and cell intercalation (D', E'), near the pit (D,D') and far from the pit (E,E'). Note how cell shape changes apical versus basal near the pit suggest progressive cell wedging as apices contract isotropically more rapidly than basally (D), and how the rate of cell intercalation across the tissue is highly coordinated between apical and basal, especially in the cells far from the pit (D',E'). Statistical significance based on a mixed-effects model and a $p<0.05$ threshold (calculated for instantaneous strain rates [see *Figure 2—figure supplement 1* and *Figure 3—figure supplement 2*]), is indicated by shaded boxes at the top of each panel: apical 'rad' vs basal 'rad' (dark grey) and apical 'circ' vs basal 'circ' (light grey).

DOI: https://doi.org/10.7554/eLife.35717.014

The following source data and figure supplements are available for figure 3:

**Source data 1.** Cumulative strain and statistics for cells near to the pit and far from the pit at apical and mid-basal depth.
DOI: https://doi.org/10.7554/eLife.35717.019

**Figure supplement 1.** Strain rate analysis of early tubulogenesis using a 3D proxy.
DOI: https://doi.org/10.7554/eLife.35717.015

**Figure supplement 1—source data 1.** Cumulative strain for all cells at apical and mid-basal depth and tissue strain rate for cells near to the pit and far from the pit from an apical and basal depth.
DOI: https://doi.org/10.7554/eLife.35717.016

**Figure supplement 2.** Strain rate analysis of early tubulogenesis using a 3D proxy.
DOI: https://doi.org/10.7554/eLife.35717.017

**Figure supplement 2—source data 1.** Instantaneous strain rate for mid-basal depth.
DOI: https://doi.org/10.7554/eLife.35717.018

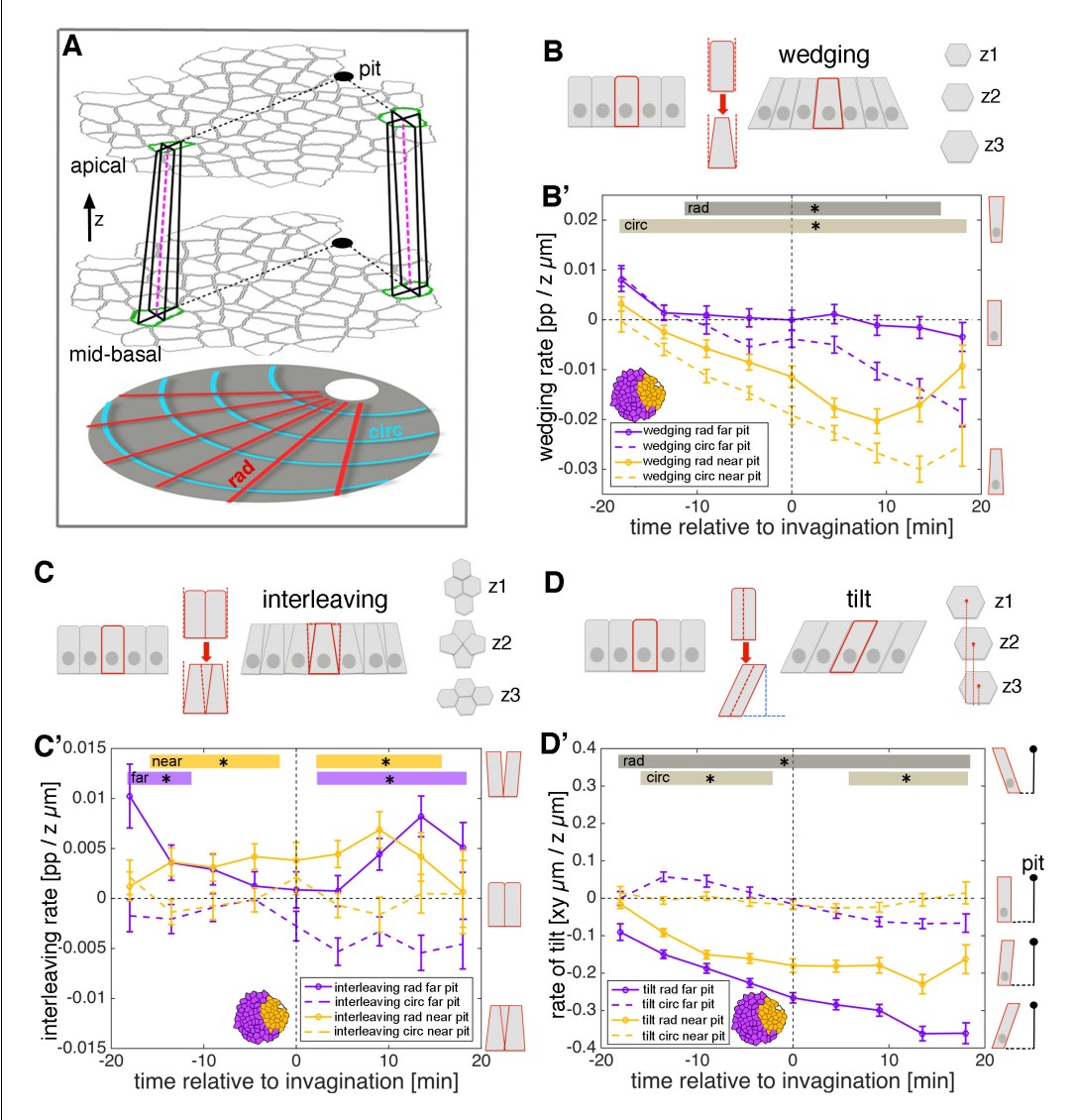

**Figure 4.** 3D cell geometries during early tube budding. (A) Illustration of the method used for calculating 3D domain geometries using tracked cells at apical and mid-basal levels throughout the salivary placode. Although for simplicity only two matched cells are shown here (green outlines in both z-levels), all cells were matched accurately between levels (see text and *Video 4*). Small domains of a focal cell and its immediate neighbours were used to calculate local rates of wedging, interleaving and tilt (from cell 'in-lines', pink dashed lines) using z-strain rate methods (see text and *Figure 4—figure supplement 1A–D*). Measures were projected onto the pit-centred radial (dotted black lines) and circumferential axes. In this tilted side-view cartoon, the z distance between apical and mid-basal levels has been exaggerated. (B) Schematic of cell wedging in a cross-section and as individual z-levels (z1–z3) in the marked red cell. (B') Cell wedging is patterned across the placode, increasing most strongly in cells near to the pit, in accordance with the isotropic apical constriction observed here (orange lines). Away from the pit, wedging contributes mainly in the circumferential direction (purple dashed line). (C) Schematic of cell interleaving in a cross-section and as individual z-levels (z1–z3) in the group of marked red cells. (C') Radial cell interleaving (solid line) is always more positive than circumferential interleaving (dashed lines), often significantly so. Interleaving contributes to a radial expansion apically and a concomitant circumferential contraction. (D) Schematic of cell tilt in a cross-section and as individual z-levels (z1–z3) in the marked red cell. (D') Cell tilt increased continuously in the radial direction (solid lines) apically towards the pit over the period of our analysis. A stronger rate of tilt was observed for the cells further from the pit (purple solid line), which is expected from the radial wedging seen near the pit. (B', C', D') Error bars show the mean of the within-embryo variances for five movies. Significance at p<0.05 using a mixed-effect model (see Materials and methods) is depicted as shaded boxes at the top of the panel: B') wedging in radial orientation, near to vs far from pit (dark grey, 'rad') and wedging in circumferential orientation, near to vs far from pit (light grey, 'circ). (C') Interleaving near to the pit, radial vs circumferential (orange, 'near') and interleaving far from the pit, radial vs circumferential (purple, 'far'). (D') Tilt in radial orientation, near to vs far from pit (dark grey, 'rad') and tilt in circumferential orientation, near to vs far from pit (light grey, 'circ).

DOI: https://doi.org/10.7554/eLife.35717.020

*Figure 4 continued on next page*

*Figure 4 continued*

The following source data and figure supplements are available for figure 4:

**Source data 1.** Wedging, interleaving and tilt mean ± confidence intervall (CI) and statistics.
DOI: https://doi.org/10.7554/eLife.35717.023
**Figure supplement 1.** 3D cell geometries during early tube budding.
DOI: https://doi.org/10.7554/eLife.35717.021
**Figure supplement 1—source data 1.** Total strain rate in Z and statistics.
DOI: https://doi.org/10.7554/eLife.35717.022

Strain rate analysis of changes within the apical domain of the epithelial cells in the early salivary gland placode clearly confirmed the existence of two distinct zones of cell behaviour. Spatial plots summarising ~ 36 mins of data centred around the first tissue bending event (combined from the analysis of 9 wild-type embryos; *Figure 2—figure supplement 1A*) revealed strong isotropic tissue contraction near the invagination pit (*Figure 2C'* and *Figure 2—figure supplement 1B*, green) that was mainly contributed by apical cell constriction (*Figure 2C''* and *Figure 2—figure supplement 1B'*). At a distance from the pit, tissue elongation dominated in the radial direction (*Figure 2C'*, magenta) and was contributed mostly by cell intercalation (*Figure 2C'',C'''*). The split in behaviour and the strong contribution of cell intercalation to the early invagination and tube formation was even more apparent from time resolved strain rate analyses. At the whole tissue level, apical cell constriction began more than 10 min before any curvature change at the tissue level (*Figure 2—figure supplement 1C,C'*, green curve), and this change was most pronounced in the cells near the invagination pit (*Figure 2D,D'* versus E, E', green curve). In addition, cell intercalation also commenced about 10 min prior to tissue bending (*Figure 2—figure supplement 1C,C'*, orange curve), but in this case, the stronger contribution came from cells far from the pit (*Figure 2E,E'* versus D,D', orange curve).

Although constriction was isotropic near the pit, with equally large magnitudes contributing both radially and circumferentially (*Figure 2D,D'*, green curves), intercalation was clearly polarised towards the invagination pit, with expansion radially in the orientation of the pit (*Figure 2D,E* and *Figure 2—figure supplement 1C*, 'rad', orange curves), and contraction circumferentially (*Figure 2D',E'* and *Figure 2—figure supplement 1C'*, 'circ', orange curves; see also *Figure 2—figure supplement 1F*).

Thus, in addition to apical constriction, directional cell intercalation constitutes a major second cell behaviour occurring during tissue bending and invagination of a tube. Furthermore, the amount of both cell behaviours occurring was radially patterned across the placode. However, it was not clear whether both these behaviours were entirely being driven by active apical mechanisms, or whether further basal events contributed, so we next investigated the 3D behaviours of placode cells.

## A quasi-3D tissue analysis at two depths shows coordination of cell behaviours in depth

The apical constriction as observed in the placodal cells near the pit is indicative of a redistribution of cell volume more basally, which could result in a combination of cell wedging and/or cell elongation in depth. The 3D-cell behaviour of acquiring a wedge-like shape is of particular interest in the placode because it is capable of tissue bending. Apical constriction (coupled with corresponding basal expansion if cell volume is maintained, as in the case of the salivary gland placode [data not shown]) is known to deform previously columnar or cuboidal epithelial cells

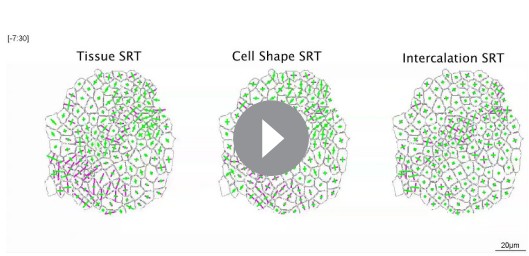

**Video 3.** Example movies of Tissue strain rate tensor (SRT), Cell Shape SRT and Intercalation SRT. Local strain rates are extracted from segmented and tracked movies. Rates of tissue shape change (left), cell shape change (centre) and intercalation (right) are shown. Green vectors represent contraction and magenta vectors represent expansion. Time stamp indicates time before and after initiation of tissue bending at t = 0. See also *Figure 2*.
DOI: https://doi.org/10.7554/eLife.35717.013

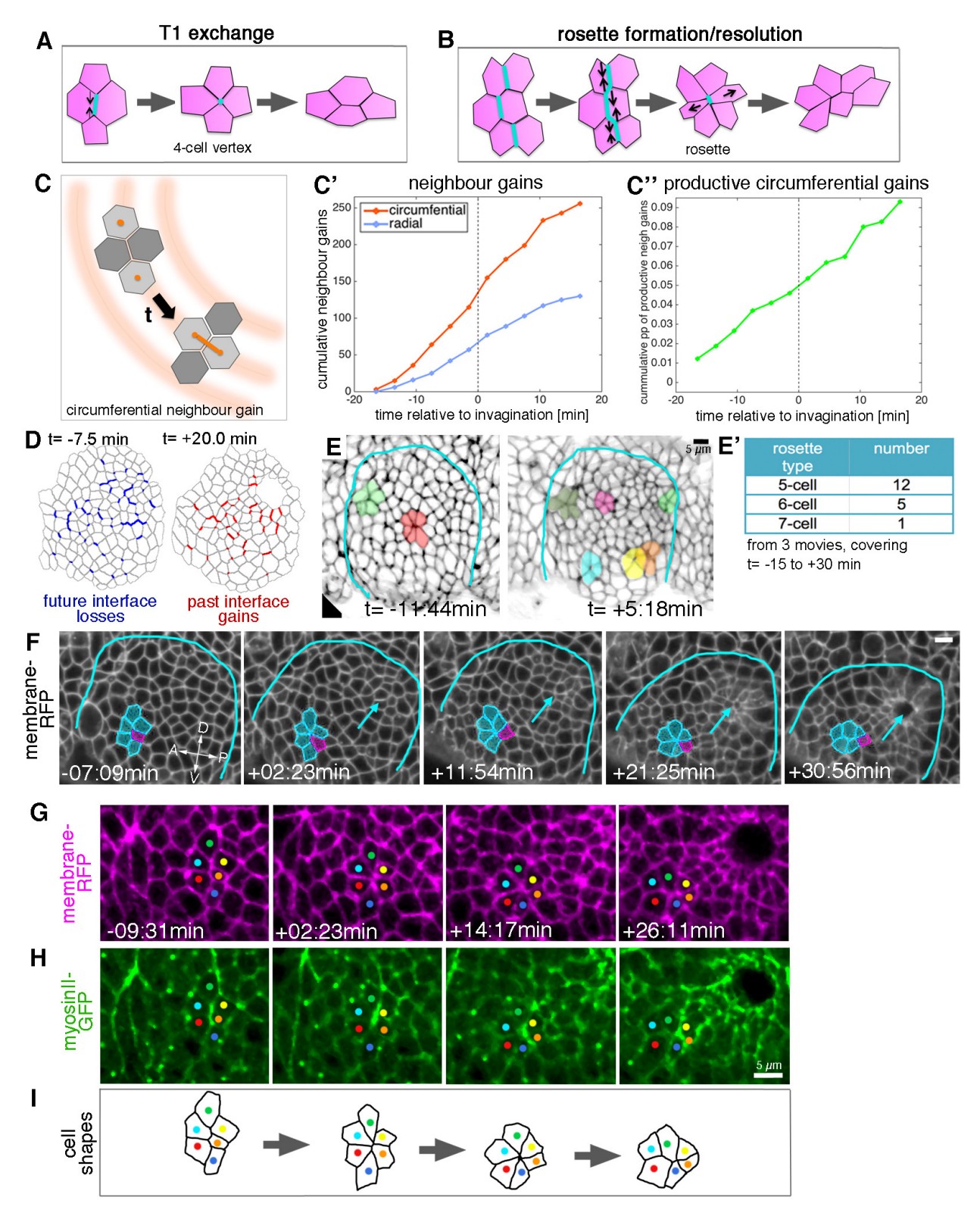

**Figure 5.** Cell intercalation during tube formation combines T1 exchanges and rosette-formation/resolution. (A,B) Depending on the number of cells involved, intercalation can proceed through formation and resolution of a four-cell vertex, a so-called T1-exchange (A), or through formation and resolution of a rosette-like structure than can involve 5–10 or more cells (B). (C–C'') Quantification of neighbour gains as a measure of T1 and intercalation events, with an example of a circumferential neighbour gain (leading to radial tissue expansion) shown in (C). (C') Circumferential

*Figure 5 continued on next page*

*Figure 5 continued*

neighbour gains dominate over radial neighbour gains. (C'') Rate of productive gains, defined as the amount of circumferential neighbour gains leading to radial tissue elongation and expressed as a proportion (pp) of cell-cell interfaces tracked at each time point. Data are pooled from seven embryo movies. (D) Represented for a single exemplary movie analysed (ExpID0356), interfaces that will be lost (blue) or have been gained (red) are shown, mapped onto a segmented version of the placode at the start of the movie for future losses (t = −7.5 min) and at the end of the movie for past gains (t = +20.0 min). Note that lost interfaces are mostly oriented circumferentially, whereas gained interfaces are mostly radial (see alternative visualisation in *Figure 5—figure supplement 1*). (E) Stills of two time-lapse movies used for the strain rate analysis, illustrating the appearance of rosette structures prior to (t= −11:44 min) and after (t = +5:18 min) tissue bending. (E') The majority of rosettes observed in the salivary gland placode consist of five to six cells. Data are pooled from three embryo movies. (F–H) Stills of a time-lapse movie of an example of rosette formation-resolution. (F) The cluster of cells contracts along the circumferential direction of the placode, and resolution from the six-cell-vertex is oriented towards the pit (arrow). Label is *Ubi-RFP-CAAX*. (G,H) The same rosette as in (F) in close-up, showing *Ubi-RFP-CAAX* to label membranes in magenta (G) and myosin II-GFP (*sqhGFP*) in green (H). Note the prominent but transient myosin accumulation at the centre of the rosette-forming group of cells. (I) is a schematic of the rosette formation-resolution analysed in (F–H). See also *Video 5*. Scale bars in (E, F, H) are 5 μm.
DOI: https://doi.org/10.7554/eLife.35717.025

The following source data and figure supplement are available for figure 5:

**Source data 1.** Cumulative neighbour gains and cumulative proportions of circumferential productive neighbour gains.
DOI: https://doi.org/10.7554/eLife.35717.027

**Figure supplement 1.** Cell intercalation during tube formation combines T1 exchanges and rosette-formation/resolution.
DOI: https://doi.org/10.7554/eLife.35717.026

into wedge-shaped cells (*Martin and Goldstein, 2014*; *Wen et al., 2017a*). This is also true in the salivary gland placode, where cross-sections in xz of fixed samples and time-lapse movies confirm the change from columnar to wedged morphology (*Figure 1B,B'*) (*Girdler and Röper, 2014*; *Myat and Andrew, 2000b*). Most analyses of morphogenetic processes are conducted with a focus on events within the apical domain given the prevalent apical accumulation of actomyosin and junctional components (*Figure 3A,A'*) (*Bosveld et al., 2012*; *Butler et al., 2009*; *Martin et al., 2009*; *Rauzi et al., 2010*; *Simões et al., 2010*). However, this apico-centric view neglects most of the volume of the cells. For instance, in the case of the salivary gland placode, cells extend up to 15 μm in depth (*Figure 3B*). We thus decided to analyse cell and tissue behaviour during early stages of tube formation from the placode in a 3D context. Automated cell segmentation and tracking of 3D behaviours is still unreliable in the case of tissues with a high amount of curvature, such as in the salivary gland placode once invagination begins. To circumvent this issue, we used strain rate analysis at different depth as a proxy for a full 3D analysis (*Figure 3A',C*). After accounting for the overall curvature of the tissue, we segmented and tracked placodal cells not only within the apical region (as shown in *Figures 1* and *2*; *Figure 3A',C* pink), but also at a more basal level, ~8 μm below the apical domain (*Figure 3A',C* blue), and repeated the strain rate analysis at this depth (*Figure 3D–E''* and *Figure 3—figure supplement 1* and *Figure 3—figure supplement 2*). We used the same radial coordinate system with the pit as the origin for both layers, so we were able to compare cell behaviours between depths. Segmentation and analysis at the most-basal level of placodal cells was not reliably possible due low signal-to-noise ratio of all analysed membrane reporters at this depth (data not shown).

Overall, the spatial pattern of change at mid-basal level was similar to changes within the apical domain during 33 min centred around the start of tissue bending (*Figure 3—figure supplement 1B,C*). Isotropic tissue contraction at mid-basal depth was clustered around the position of the invagination pit (*Figure 3—figure*

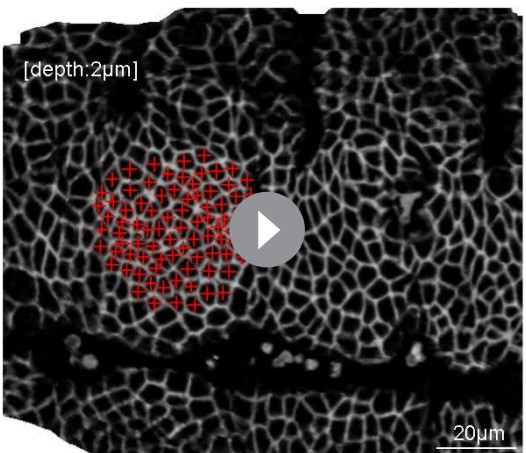

**Video 4.** Example of matching cells through depth. Apical cell identities matched to basal cell identities are shown within the placode. Stamp refers to the depth in the tissue. Scale bar 20 μm. See also *Figure 4*.
DOI: https://doi.org/10.7554/eLife.35717.024

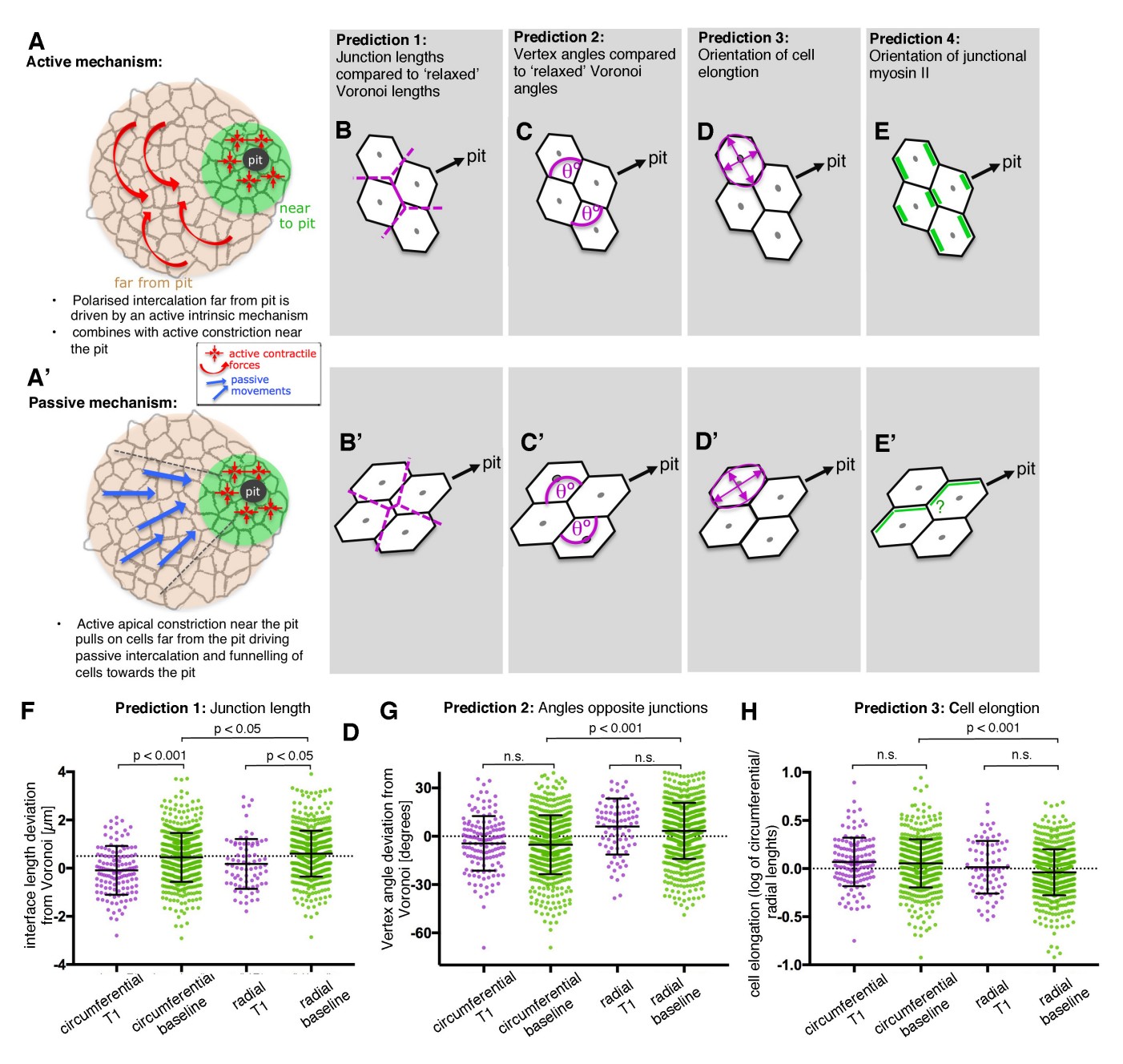

**Figure 6.** Analysis of signatures of active versus passive cell intercalation. (**A, A'**). Cell intercalation in cells far from the pit could either be an active process, helping to draw cells together circumferentially and to extend tissue towards the pit (**A**), or the active pulling from the contractile pit could lead to a passive funnelling of cells towards the pit with associated passive intercalations (**A'**). We considered four measures should allow to distinguish between the above active and passive mechanisms. (**B, B'**) The lengths of actively shrinking circumferential junctions we predicted would be shorter than equivalent junction lengths of a Voronoi tessellation seeded with real cell centroid locations. We used the Voronoi tessellation as a proxy for 'relaxed' (not actively or passively altered) junction lengths and angles. Junctions pulled radially by the pit would be longer than predicted by Voronoi tessellation. (**C,C'**) Angles at vertices opposite actively shrinking circumferential junctions we predicted to be more acute than angles derived from a relaxed Voronoi tessellation, with the opposite predicted for pulled radial junctions. (**D, D'**) Active intercalation should lead to circumferential elongation of cells connected at vertices to shrinking junctions, whereas pulling from the pit would lead to radial cell elongation. (**E, E'**) Junctional actomyosin is predicted to be enriched in circumferential junctions that are actively shrinking. A response to pulling from the pit could lead to no enrichment or radially enriched junctional myosin (see text). (**F**) The junction lengths of both circumferentially and radially oriented junctions undergoing a T1 exchange showed an active signature (shorter than expected) compared to similarly oriented non-intercalating junctions. N = 135, 523, 78 and 618 junctions for the four distributions, respectively, from seven embryos. (**G**) Circumferentially oriented junctions far from the pit had an active signature of

*Figure 6 continued on next page*

*Figure 6 continued*

their vertex angles being more acute compared to radially oriented junctions, though junctions undergoing a T1 did not have more acute angles than junctions not directly involved in a T1. N = 139, 571, 84 and 675 junctions for the four distributions, respectively, from seven embryos. (**F**) Cells far from the pit had an active signature of being more elongated in the circumferential orientation. Cells at the ends of junctions involved in a T1 did not behave differently to junctions not involved in a T1. N = 135, 523, 78 and 618 junctions for the four distributions, respectively, from seven embryos. Error bars in (**F–H**) are ±SD.

DOI: https://doi.org/10.7554/eLife.35717.030

The following source data and figure supplements are available for figure 6:

**Source data 1.** Predictions (junction length, angles opposite shrinking junctions and cell elongation) for circumferential T1, circumferential baseline, radial T1 and radial baseline and their statistics.
DOI: https://doi.org/10.7554/eLife.35717.033
**Figure supplement 1.** Analysis of signatures of active versus passive cell intercalation.
DOI: https://doi.org/10.7554/eLife.35717.031
**Figure supplement 1—source data 1.** Interface length during a T1 transition and baselines for predictions indicated above.
DOI: https://doi.org/10.7554/eLife.35717.032

*supplement 1B,C*, green), and this was due to cell constriction at this level (*Figure 3—figure supplement 1B',C'*), whereas the tissue expansion at a distance from the pit in the radial direction (*Figure 3—figure supplement 1C*) was due primarily to cell intercalation (*Figure 3—figure supplement 1C''*). Similarly to events at the apical level, this radial expansion towards the pit was accompanied by a circumferential contraction, again due primarily to cell intercalation in the region away from the pit (*Figure 3—figure supplement 1B,B''*).

Comparing apical and basal strain rates at the cell and tissue level with respect to their radial and circumferential contributions revealed an interesting picture. In temporally resolved plots, isotropic cell constriction dominated apically in cells near the pit (*Figure 3D*, pink), but with a slower rate of constriction at the mid-basal depth (*Figure 3D*, blue). This was confirmed by cross-section images that show that, near the pit and once tissue-bending had commenced, the basal surface of the cells was displaced even further basally than in the rest of the placode, and cells were expanded at this level, leading to an overall wedge-shape (*Figure 1B'*). In cells away from the pit, similar to the apical region, cell shapes did not change much (*Figure 3E*). In contrast to cell shape changes that diverged at depth at least in the cells near the pit, intercalation behaviour appeared to be highly coordinated between apical and basal levels with near identical contributions at both particularly in cells far from the pit (*Figure 3D', E'* and *Figure 3—figure supplement 1D''*).

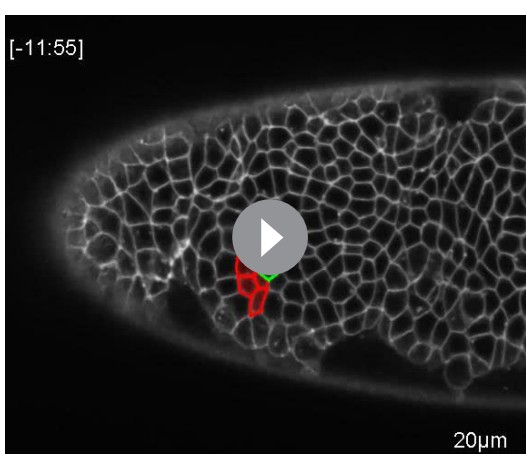

**Video 5.** Example movie of apical rosette formation/resolution. Embryo of the genotype *sqh[AX3]; sqh::sqhGFP42, UbiRFP-CAAX*, only the UbiRFP label is shown. A group of cells going through rosette formation/resolution is highlighted, still of the movie are shown in *Figure 5F*. Time stamp indicates time before and after initiation of tissue bending at t = 0. Scale bar 20 µm.
DOI: https://doi.org/10.7554/eLife.35717.028

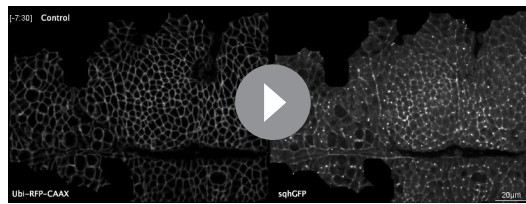

**Video 6.** Example movie of cell shape and myosin II localisation in a control embryo. Embryo of the genotype *sqh[AX3]; sqh::sqhGFP42, UbiRFP-CAAX* used for the myosin II uni- and bi-polarity quantifications as shown in *Figure 7*. Time stamp indicates time before and after initiation of tissue bending at t = 0. Scale bar 20 µm.
DOI: https://doi.org/10.7554/eLife.35717.029

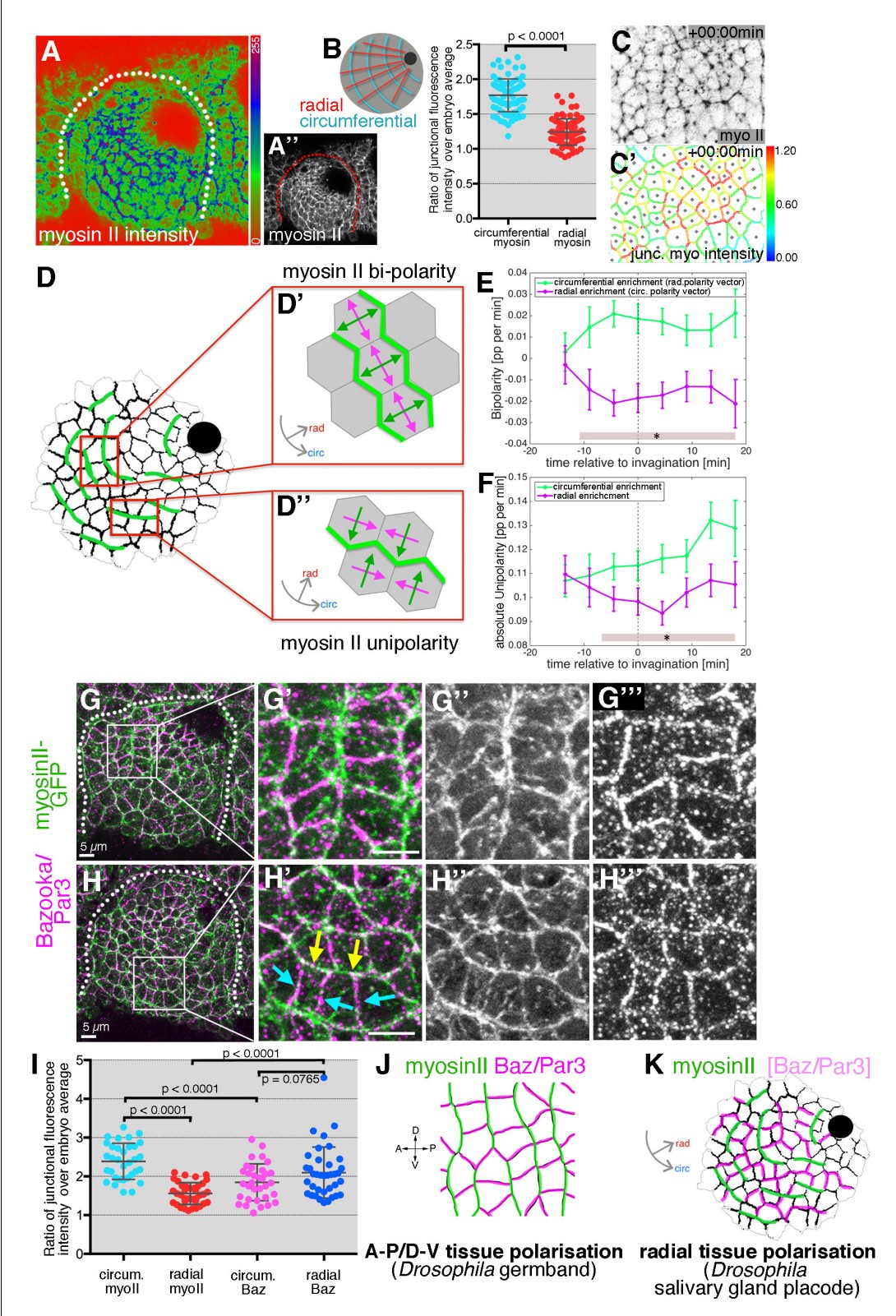

**Figure 7.** Junctional myosin II shows a strong circumferential polarisation during tube formation. (**A–B**) Example of junctional myosin II intensity, visualised using *sqh[AX3]; sqhGFP* in fixed embryos; (**A**) shows a heat map of intensity. (**B**) Quantification of apical junctional myosin polarisation, depicted as the ratio of junctional intensity (circumferential or radial) over embryo average (5 placodes from 5 embryos, 83 circumferential junctions versus 83 radial junctions; significance calculated using unpaired t-test; embryo average is average apical myosin intensity in the epidermis outside the

*Figure 7 continued on next page*

*Figure 7 continued*

placode). (C, C') Myosin II enrichment was quantified from segmented and tracked time-lapse movies. (C) depicts the sqhGFP signal from a single time point of a movie, (C') shows the calculated junctional myosin II intensity. See also *Videos 6* and *7*. (D–F) Myosin enrichment at junctions can occur in two flavours: (D', E) Myosin II bi-polarity is defined as myosin II enrichment at two parallel oriented junctions of a single cell, calculated as the magnitude of a vector pointing at the enrichment (D'). Circumferential myosin II bi-polar enrichment (i.e. the radial bi-polarity vector, green in (D') pointing at myosin II enrichment), increases between −15 min and + 18 min (E), green curve). (D'', F) Myosin II unipolarity is defined as myosin II enrichment selectively on side of a cell (D''). Circumferential myosin II uni-polar enrichment (i.e. the radial uni-polarity vector, green, in (D'') pointing at myosin II enrichment), increases between −15 min and +18 min (F), green curve), whereas radial uni-polar enrichment (i.e. the circumferential uni-polarity vector in magenta in D''), does not increase (F), magenta curve). (E, F) Error bars represent the mean of within-embryo variances of four movies, and significance between radial and circumferential bi- and uni-polarity at $p<0.05$ using a mixed-effects model (see Materials and methods) is depicted as shaded boxes at the bottom of panels E and F. See also *Video 7*. (G–H''') Examples of myosin II and Bazooka/Par3 polarisation in fixed embryos, two different regions of two placodes are shown. Note the complementary localisation of myosin II (green) and Bazooka (magenta). Scale bars are 5 μm. (I) Quantification of myosin II and Bazooka/Par3 polarisation in areas of strong junctional myosin II enrichment at circumferential junctions as shown in (G') and (H'). Both circumferential (yellow arrows in H') and radial (turquoise arrows in H') fluorescence intensity are shown as ratios over embryo average, both for myosin II and Bazooka in the same junctions (from five embryos; mean and SEM are shown; paired t-test for comparison in the same junctions, unpaired t-tests for comparison of circumferential myo vs radial myo and circumferential Baz vs radial Baz; N = 33 circumferential junctions and N = 38 radial junctions across five embryos; embryo average is average apical myosin II or bazooka intensity in the epidermis outside the placode). (J, K) In contrast to the well-documented A-P/D-V polarity of cells in the *Drosophila* germband during elongation in gastrulation (J), the salivary gland placode appears to show a radial tissue polarisation, with a radial-circumferential molecular pattern imprinted onto it that instructs the morphogenesis (K).

DOI: https://doi.org/10.7554/eLife.35717.034

The following source data is available for figure 7:

**Source data 1.** Myosin and Bazooka fluoresce intensity in radial and circumferential junctions and Myosin unipolary and bipolarity over time.
DOI: https://doi.org/10.7554/eLife.35717.035

## Patterns of cell wedging, interleaving and tilt in the placode

In cells near to the pit, the faster rate of apical constriction implies that cell wedging is occurring, but we have not confirmed this in the same dataset by measuring the relative sizes of apical and basal cell diameters. Similarly, although the rates of apical and basal intercalation are remarkably similar, this does not rule out that the actual arrangement of cells at one level is 'tipped' ahead of the other, through interleaving in depth (akin to a T1 transition along the apical-basal axis, see z-sections in *Figure 4C* and *Figure 4—figure supplement 1Dd"*). For example, if cell rearrangement is being driven by an active apical mechanism (see below and Figure 6), we predict that apical cell contours would be intercalating ahead of basal cell contours, even while their rates remain the same. Both wedging (*Figure 4B*) and interleaving (*Figure 4C*) have implications for the tilt, or lean, of cells relative to epithelial surface normals (*Figure 4D* and *Figure 4—figure supplement 1A–C*), with a gradient of tilt expected for constant wedging or interleaving in a flat epithelium. We therefore set out to quantify 3D wedging, interleaving and cell tilt with new methods.

We used the placodes (n = 5) for which we have tracked cells at both apical and mid-basal levels. First, we developed a semi-automated method to accurately match cell identities correctly between depths (*Figure 4A*, *Video 4*, and see Materials and methods). We then borrowed ideas from recent methods developed to account for epithelial curvature in terms of the additive contributions of cell wedging and interleaving in depth (*Deacon, 2012*). In the early developing salivary gland placode, average tissue curvature is very slight, so we simplified the above methods for flat epithelia. We applied exactly the same methods that we have used in *Figures 2* and *3* to calculate strain rates for small cell domains, but rather than quantifying rates of deformation over time, now we quantify rates of deformation in depth (see z-level illustrations in *Figure 4B,C,D*

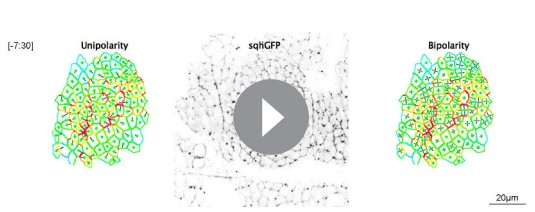

**Video 7.** Example movie of automatic junctional myosin II quantification. Myosin II uni-polarity vectors (left) and bi-polarity vectors (right) are shown. Junctions are colour coded according to their myosin II intensity levels as shown in *Figure 7C'* (middle). Time stamp indicates time before and after initiation of tissue bending at t = 0. Scale bar 20 μm. See also *Figure 7*.
DOI: https://doi.org/10.7554/eLife.35717.036

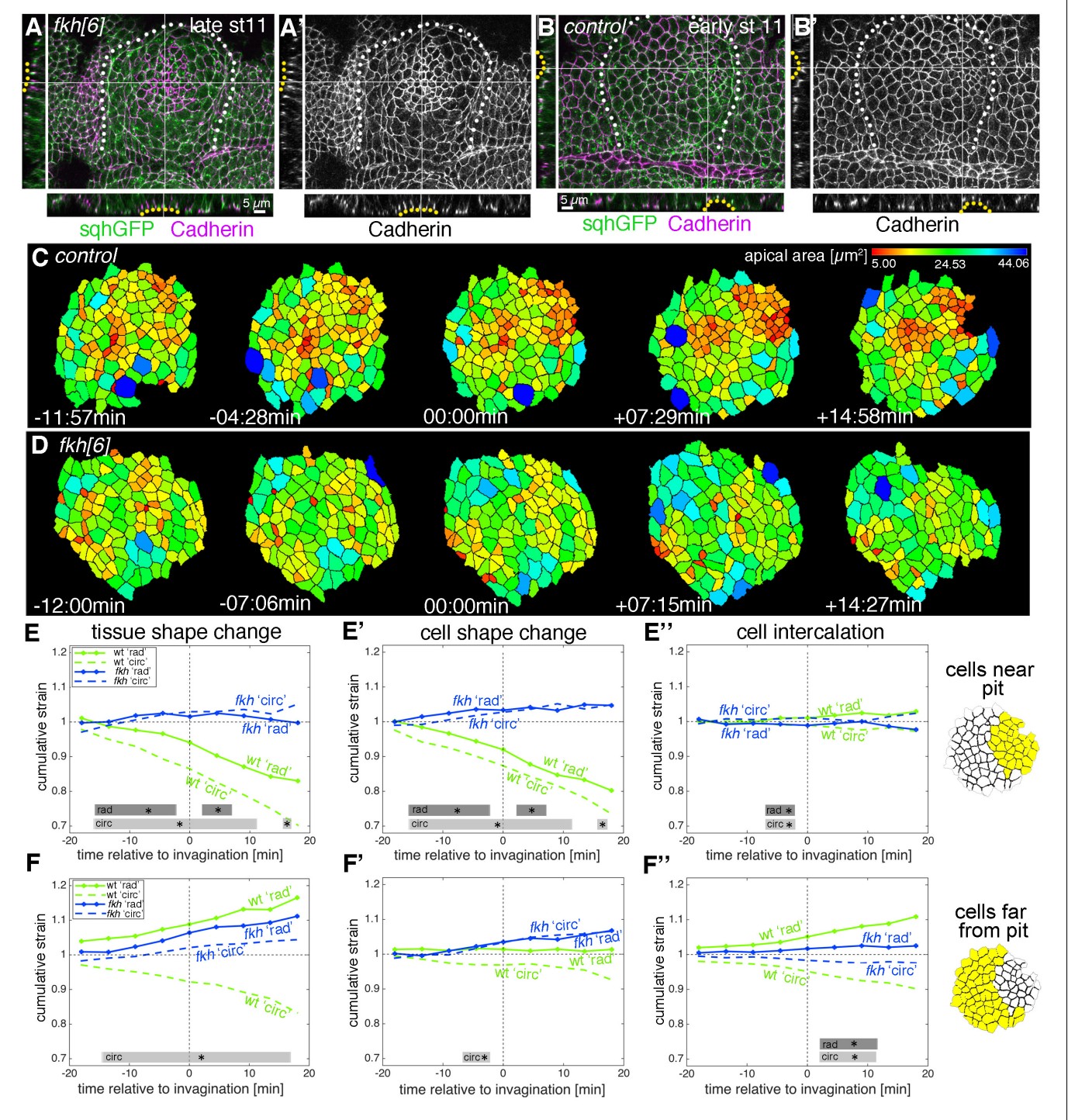

**Figure 8.** Loss of radial patterning of cell behaviours in salivary gland placodes lacking Fkh. (**A–B'**) Examples of *fkh* mutant and wild-type placodes illustrating the lack of pit formation. A control placode at early stage 11 shows clear constriction and pit formation in the dorsal-posterior corner (yellow dotted line in cross-sections in **B**, **B'**), whereas even at late stage 11 a *fkh* mutant placode (beyond the time frame of our quantitative analysis) does not show a pit in the dorsal-posterior corner, instead a shallow central depression forms with some constricted central apices (yellow dotted line in cross-sections in **A**), (**A'**). Myosin II (sqhGFP) is in green and DE-Cadherin in magenta, white dotted lines in main panels outline the placode boundary. (**C, D**) Stills of segmented and tracked time lapse movies for control (**C**) and *fkh[6]* mutant (**D**) placodes, apical cell outlines are shown and colour-coded by apical cell area. Note the lack of contraction at the tissue and cell level in the *fkh[6]* mutant. (**E–F''**) Analysis of cumulative apical strains in *fkh* mutant placodes (from five movies) in comparison to the analysis of wild-type embryos (as shown in *Figure 2*). Over the first 36 min of tube budding centred around the first appearance of tissue-bending in the wild-type and an equivalent time point in the *fkh* mutants, the *fkh* mutant placodes show only a

*Figure 8 continued on next page*

*Figure 8 continued*

slight expansion at the tissue level in the cells far from the predicted pit position (**F**) due to cell shape changes (**F'**) with very little intercalation contributing to the change (**E''**, **F''**). Statistical significance based on a mixed-effects model and a p<0.05 threshold (calculated for instantaneous strain rates [see *Figure 2—figure supplement 1* and *Figure 8—figure supplement 2*]), is indicated by shaded boxes at the top of each panel: wt 'rad' vs *fkh* 'rad' (dark grey) and wt 'circ' vs *fkh* 'circ' (light grey).

DOI: https://doi.org/10.7554/eLife.35717.037

The following source data and figure supplements are available for figure 8:

**Source data 1.** Cumulative strain and statistics for cells near to the pit and far from the pit for wild type and *fkh[6]* mutant.
DOI: https://doi.org/10.7554/eLife.35717.041
**Figure supplement 1.** Loss of radial patterning of cell behaviours in salivary gland placodes lacking Fkh.
DOI: https://doi.org/10.7554/eLife.35717.038
**Figure supplement 2.** Loss of radial patterning of cell behaviours in salivary gland placodes lacking Fkh.
DOI: https://doi.org/10.7554/eLife.35717.039
**Figure supplement 2—source data 1.** Instantaneous strain rates for *fkh[6]* mutant.
DOI: https://doi.org/10.7554/eLife.35717.040

and *Figure 4—figure supplement 1D*). The cell shape strain rate becomes a wedging strain in depth, in units of proportional shape change per micron in z (*Figure 4B,B'* and *Figure 4—figure supplement 1Dc, c'*), the intercalation strain rate becomes the interleaving strain in depth, in the same units (*Figure 4C,C'* and *Figure 4—figure supplement 1Dd, d'*), and translation velocity becomes the cell tilt (*Figure 4D,D'* and *Figure 4—figure supplement 1A*,Da,a'; see Materials and methods for details). Once again, we projected the z-strain rates and tilt onto our radial coordinate system.

Cells across the placode started out at −18 min before pit invagination unwedged and mostly untilted in radial and circumferential orientations (*Figure 4B',D'*). Cells near the pit became progressively wedge-shaped over the next 30 min, with smaller apices (*Figure 4B'*, orange lines; *Figure 4—figure supplement 1C*). Cell wedging was reasonably isotropic, but with circumferential wedging always stronger than radial. Away from the pit, progressive wedging was less rapid, again with a strong circumferential contribution but nearly no radial contribution (*Figure 4B'*, purple lines). That cells were less wedged radially might be because this is the orientation in which cells move into the pit, releasing radial pressure due to apical constriction near the pit.

The wedging anisotropy is also compatible with active circumferential contraction. Indeed, circumferential interleaving was always more negative than radial interleaving, often significantly so (*Figure 4C'*, solid vs dashed lines). Thus, interleaving contributes a circumferential tissue contraction apically, with a concomitant radial expansion. This pattern is thus also compatible with an apical circumferential contraction mechanism, possibly driving cell rearrangements.

Cell tilt, a measure of the divergence of a cell's in-line from the surface normal (*Figure 4D* and *Figure 4—figure supplement 1A,B,D*), increased continuously in the radial direction towards the pit over the period of our analysis (*Figure 4D'*, solid lines). A stronger tilt was observed for the cells further from the pit (*Figure 4D'*, purple solid line), which is expected from the radial wedging seen near the pit (*Figure 4B'*, solid orange line).

Hence, the relatively isotropic rates of apical constriction near the pit and the very similar rates of intercalation apically versus basally were in fact grounded in anisotropic wedging near the pit and in an interleaving difference between apical and basal. 3D tissue information such as wedging, interleaving and tilt are therefore essential to fully understand planar cell behaviours such as cell shape change and intercalation. Overall, our combined analysis so far suggests that isotropic apical constriction near the pit combines with an apically led circumferential contraction

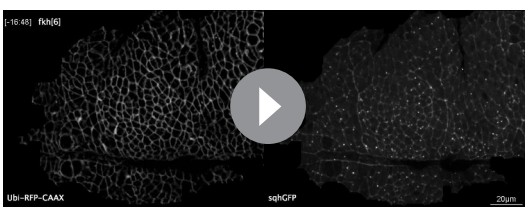

**Video 8.** Example movie of cell shape and myosin II localisation in a fkh[6] mutant embryo. Embryo of the genotype *sqh::sqhGFP42, UbiRFP-CAAX; fkh[6]* used for the myosin II uni- and bi-polarity quantifications as shown in *Figure 9*. Time stamp indicates time before and after initiation of tissue bending at t = 0. Scale bar 20 μm.

DOI: https://doi.org/10.7554/eLife.35717.042

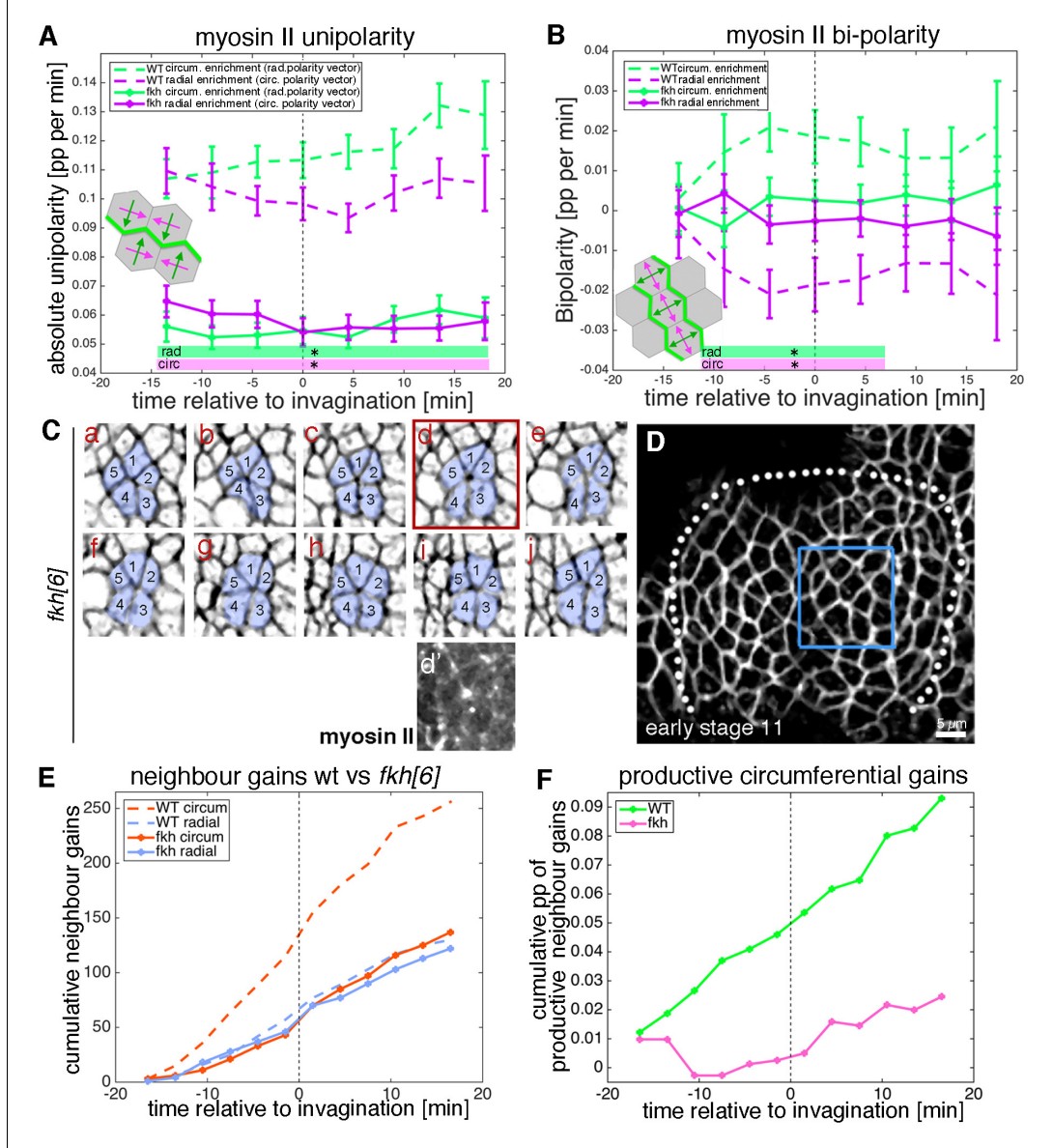

**Figure 9.** Analysis of myosin patterns and intercalation behaviour in fkh mutants. (**A,B**) Analysis of myosin II unipolarity (**A**) and bi-polarity (**B**) in *fkh[6]* mutant placodes compared to wild-type (as shown in *Figure 7*, see also *Videos 7* and *8*). Overall myosin II levels in *fkh[6]* mutant placodes are lower (not shown) and both unipolarity and bi-polarity are decreased, with bi-polarity near zero (**B**, solid curves). Error bars show intra-embryo variation of five embryo movies for *fkh[6]* and four embryo movies for wt. Statistical significance at p<0.05 using a mixed-effect model is indicated as shaded boxed at the bottom of the panels: wt vs *fkh[6]* for the radial vector/circumferential enrichment (green, 'rad') and wt vs *fkh* for the circumferential vector/radial enrichment (purple, 'circ'). (**C–F**) Neighbour exchanges still occur in *fkh[6]* mutant placodes. (**C**) Sill pictures from a tracked and segmented movie of a *fkh[6]* mutant placode (labelled with membrane-RFP), frames (labelled **a–j**) are 1:55 min apart; the full placode view in (**D**) is from time point (**a**, **d'**) shows the myosin accumulation at the centre of the rosette structure observed in (**d**). (**E**) In comparison to wt where circumferential neighbour gains dominate over radial ones (dashed lines), in the *fkh[6]* mutant placodes both occur with equal frequency (solid lines). This leads to nearly no productive circumferential neighbour gains in the mutant, significantly fewer than in the wt (**F**); Kolmogorov-Smirnov two sample test D = 0.5833, p=0.0191).

DOI: https://doi.org/10.7554/eLife.35717.043

The following source data is available for figure 9:

**Source data 1.** Unipolarity and bipolarity and dynamics of neighbour gains in *fkh[6]* mutant.
DOI: https://doi.org/10.7554/eLife.35717.044

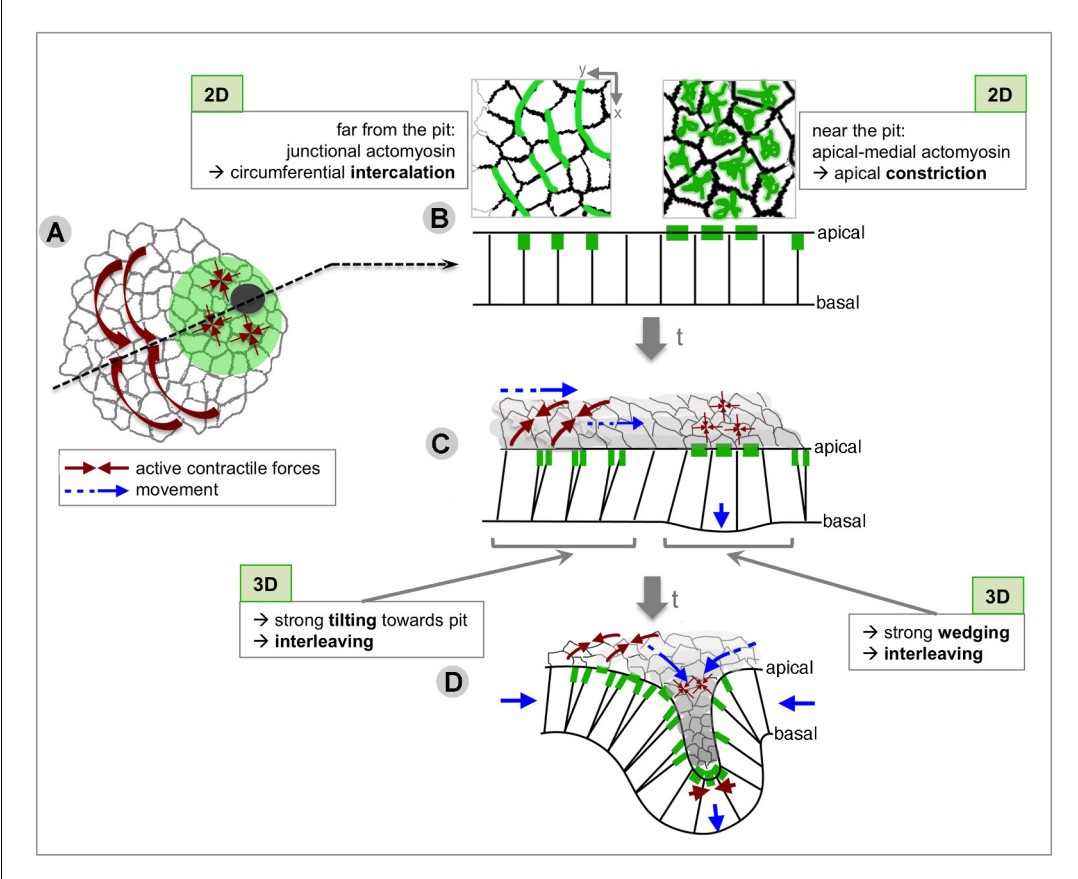

**Figure 10.** Summary of the radial patterning of 2D and 3D cell behaviours and actomyosin pools across the salivary gland placode during early tube formation. (**A**) Circumferential tissue convergence through intercalation and apical constriction at the pit combine to result in radial tissue expansion towards the invagination point. (**B**) These 2D behaviours are associated with different actomyosin pools: far from the pit, circumferential junctional actomyosin underlies active intercalation through junction shrinkage, and near to the pit a pulsatile apical-medial actomyosin underlies apical constriction (**Booth et al., 2014**). (**C**) Quasi-3D analyses revealed that active apical intercalation (thick brown arrows) and isotropic constriction (thin brown arrows) lead in 3D to strong wedging of cells near the pit, strong tilting of cells far from the pit always towards the pit, as well as interleaving (i.e. change of neighbour connectivity along the apical-basal axis) across the tissue, aiding circumferential convergence and radial extension (cell and tissue movement indicated by blue arrows). (**D**) Once tissue bending has commenced at the pit, active apical constriction in and around the pit (thick and thin brown arrows) cooperates with active circumferential intercalations (medium brown arrows) to feed the elongation of the pit tube (blue arrows).
DOI: https://doi.org/10.7554/eLife.35717.045

mechanism. We now investigate the possible origins of the latter.

## Radially polarised T1 and rosette formation and resolution underlie cell intercalation in the placode

Our strain rate analysis has revealed that the intercalation strain rate, representing the continuous process of slippage of cells past each other, was a major contributor to early tube formation and was highly coordinated between apical and basal domains. 3D domain interleaving further revealed that cell rearrangement convergence is more advanced apically in the circumferential orientation. Both these findings are measured from small groups of cells across the placode but are agnostic about neighbour exchange events or more complicated multi-neighbour exchanges. Neighbour exchanges or cell intercalations are usually thought to occur through one of two mechanisms: groups of four cells can exchange contacts through the formation of a transient four-cell vertex structure, in a typical T1 exchange (**Figure 5A**), whereas groups of more than four cells can form an intermediary structure termed a rosette, followed by resolution of the rosette to create new neighbour contacts (**Figure 5B**). During convergence and extension of tissues in both vertebrates and invertebrates, the

formation and resolution of these intermediate structures tend to be oriented along embryonic axis, with the resolution occurring perpendicular to the formation (*Blankenship et al., 2006*; *Lienkamp et al., 2012*). We therefore decided to also analyse intercalation in the placode by following individual events to identify the underlying mechanistic basis.

From our database of apical cell tracks and their connectivity, we identified all T1 transitions, classifying the time point when pairs of cells become new neighbours as 'neighbour gains'. We further sub-classified neighbour gains as being either radially or circumferentially oriented, depending on which orientation was closest to the centroid-centroid line of two cells involved in a neighbour gain (*Figure 5C*). T1s occurred at a constant rate over our study period, and we observed T1s in both orientations, revealing that neighbour connectivity was quite dynamic (*Figure 5C',D*). Nevertheless, over two-thirds of neighbour gains were oriented circumferentially, a bias that correlates with the intercalation strain rate contraction circumferentially (see for example *Figure 2E'*). This bias was also evident when visualising interface losses and gains over time for an individual placode, with losses preferentially occurring for circumferential interfaces and gains for radial interfaces (*Figure 5D* and *Figure 5—figure supplement 1*). We defined the number of productive neighbour gains as the difference between circumferential and radial gains, since equal numbers of both would cancel each other out. In order to control for any variability between placodes or between the number of cells tracked per placode, we expressed the number of productive gains as a proportion of the number of cell-cell contacts that were available to perform a circumferential T1 per time step (see Materials and methods). The proportion of productive circumferential gains was approximately constant, which lead to a steady net gain over time (*Figure 5C''*). Furthermore, the cumulative strain attributable to discrete T1s calculated here ($e^{0.09}$ = 1.09) is in good agreement with the cumulative strain over the same time window that is attributed to the continuous process of intercalation (1.1 in *Figure 2E*).

In addition to typical T1 exchanges, multicellular rosette structures could easily be identified amongst the placodal cells (*Figure 5E–F*). Rosette formation began prior to the first sign of tissue-bending, but the number of rosettes per placode increased afterwards (*Figure 5E,E'*). Rosettes were usually formed of five to seven cells, with most involving only five cells (*Figure 5E', F*). By contrast, rosettes observed during *Drosophila* germband extension can be formed of up to 12 cells (*Blankenship et al., 2006*). The strain rate analysis already indicated that, overall, intercalation events should be polarised to produce a contraction in the circumferential orientation with the corresponding expansion polarised towards the pit (*Figure 2C–E'*). Analysis of rosette formation and resolution in our time-lapse datasets of embryos expressing a membrane marker demonstrated that groups of cells contracted in a circumferential orientation to form a rosette, the resolution of which then moved individual cells towards the invaginating pit (*Figure 5F* and *Video 5*), thereby leading to the expansion observed in the strain rate analysis.

Non-muscle myosin II is the major driver of cell shape changes in many different contexts (*Levayer and Lecuit, 2012*; *Röper, 2013, 2015*), and it has been shown to play an important role in T1 transitions and in rosette formation driving the convergence and extension events during germband extension in the *Drosophila* embryo (*Blankenship et al., 2006*; *Fernandez-Gonzalez et al., 2009*; *Rauzi et al., 2010*). We imaged embryos expressing palmitoylated RFP (*Ubi-TagRFP-CAAX*) as a membrane label and a GFP-tagged version of non-muscle myosin II regulatory light chain (called spaghetti squash, *sqh*, in *Drosophila*) under control of its own promoter in the null mutant background (*sqhAX[3]; sqhGFP42*) to assess myosin II distribution and intensity as a proxy for myosin activity (*Video 6*). In all individual rosette formation-resolution examples analysed (n = 29), junctional myosin II appeared particularly enriched in the form of short cable-like structures at the central contact sites of the rosette (*Figure 5G,H*). These cables initially spanned several cell diameters and shortened concomitant with the cells being drawn into a central vertex (*Figure 5I*). The orientation of the short myosin cables correlated with the direction of rosette-formation, but in contrast to germband extension was not always oriented parallel to the dorsal-ventral axis, but rather following the circumferential coordinates of the placode.

We now investigated whether these circumferential intercalations were actively driven within the apical domain to promote invagination.

**Table 2.** Genotypes used for Figure Panels.

| Figure | Panel | Embryo genotypes | Number of embryos used for quantitative analysis | |
|---|---|---|---|---|
| *Figure 1* | B, B' | *Scribble-GFP* | panel representative of genotype (>30 embryos inspected) | fixed |
| *Figure 1* | C | *fkhGal4,UASpalm-YFP/Scribble-GFP* | panel representative of genotype (>30 embryos inspected) | live |
| *Figure 1* | G,G' | *fkhGal4,UASpalm-YFP/Scribble-GFP and sqh[AX3]; sqh::sqhGFP42, UbiRFP-CAAX* | 1 and 8 (i.e. 9 in total) | live |
| *Figure 2* | C,D,E | *fkhGal4,UASpalm-YFP/Scribble-GFP and sqh[AX3]; sqh::sqhGFP42, UbiRFP-CAAX* | 1 and 8 (i.e. 9 in total) | live |
| *Figure 3* | A,B | *fkhGal4,UASpalm-YFP/Scribble-GFP* | panels representative of genotype (>30 embryos inspected) | live |
| *Figure 3* | D,E,F | *sqh[AX3]; sqh::sqhGFP42, UbiRFP-CAAX* | 5 | live |
| *Figure 4* | C,E,G | *fkhGal4,UASpalm-YFP/Scribble-GFP and sqh[AX3]; sqh::sqhGFP42, UbiRFP-CAAX* | 1 and 4 (i.e. 5 in total) | live |
| *Figure 5* | C',C'', C''' | *fkhGal4,UASpalm-YFP/Scribble-GFP and sqh[AX3]; sqh::sqhGFP42, UbiRFP* | 1 and 6 (i.e. 7 in total) | live |
| *Figure 5* | E,E' | *sqh[AX3]; sqh::sqhGFP42, UbiRFP-CAAX* | 3 | live |
| *Figure 5* | F,G,H | *sqh[AX3];sqh::sqhGFP42, UbiRFP-CAAX* | panels representative of genotype (>30 embryos inspected) | live |
| *Figure 6* | F,G,H | *fkhGal4,UASpalm-YFP/Scribble-GFP and sqh[AX3]; sqh::sqhGFP42, UbiRFP* | 1 and 6 (i.e. 7 in total) | live |
| *Figure 7* | A | *sqh[AX3]; sqh::sqhGFP42* | panel representative of genotype (>30 embryos inspected) | fixed |
| *Figure 7* | B | *sqh[AX3]; sqh::sqhGFP42* | 5 | fixed |
| *Figure 7* | C | *sqh[AX3]; sqh::sqhGFP42 UbiRFP-CAAX* | panels representative of genotype (>30 embryos inspected) | live |
| *Figure 7* | E,F | *sqh[AX3]; sqh::sqhGFP42, UbiRFP-CAAX* | 4 | live |
| *Figure 7* | G | *sqh[AX3]; sqh::sqhGFP42* | panels representative of genotype (>30 embryos inspected) | fixed |
| *Figure 7* | H | *sqh[AX3]; sqh::sqhGFP42* | panels representative of genotype (>30 embryos inspected) | fixed |
| *Figure 7* | I | *sqh[AX3]; sqh::sqhGFP42* | 5 | fixed |
| *Figure 8* | A | *sqh::sqhGFP42, UbiRFP-CAAX; fkh[6]* | panels representative of genotype (>30 embryos inspected) | fixed |
| *Figure 8* | B | *sqh::sqhGFP42, UbiRFP-CAAX; fkh[6]/TM3* | panels representative of genotype (>30 embryos inspected) | fixed |
| *Figure 8* | C | *sqh[AX3]; sqh::sqhGFP42, UbiRFP-CAAX* | panels representative of genotype (>30 embryos inspected) | live |
| *Figure 8* | D | *sqh::sqhGFP42, UbiRFP-CAAX; fkh[6]* | panels representative of genotype (>30 embryos inspected) | live |
| *Figure 8* | E,F (as in *Figure 2C,D,E*) | *fkhGal4,UASpalm-YFP/Scribble-GFP and sqh[AX3]; sqh::sqhGFP42, UbiRFP-CAAX* | 1 and 8 (i.e. 9 in total) | live |
| *Figure 8* | E,F | *sqh::sqhGFP42, UbiRFP; fkh[6]* | 5 | live |
| *Figure 9* | A,B | *sqh[AX3]; sqh::sqhGFP42, UbiRFP-CAAX* | 4 | live |
| *Figure 9* | A,B | *sqh::sqhGFP42, UbiRFP-CAAX; fkh[6]* | 5 | live |
| *Figure 9* | C,D | *sqh::sqhGFP42, UbiRFP-CAAX; fkh[6]* | 3 | live |
| *Figure 9* | E | *sqh[AX3];sqh::sqhGFP42, UbiRFP-CAAX* | 6 | live |
| *Figure 9* | E | *sqh::sqhGFP42, UbiRFP-CAAX; fkh[6]* | 5 | live |
| *Figure 9* | F | *fkhGal4,UASpalm-YFP/Scribble-GFP and sqh[AX3]; sqh::sqhGFP42, UbiRFP* | 1 and 6 (i.e. 7 in total) | live |
| *Figure 9* | F | *sqh::sqhGFP42, UbiRFP-CAAX; fkh[6]* | 5 | live |

*Table 2 continued on next page*

*Table 2 continued*

| Figure | Panel | Embryo genotypes | Number of embryos used for quantitative analysis | |
|---|---|---|---|---|
| Figure Supplement | Panel | *Embryo genotype* | Number of embryos used for quantitative analysis | |
| *Figure 2—figure supplement 1* | B-E' | *fkhGal4,UASpalm-YFP/Scribble-GFP and sqh[AX3]; sqh:: sqhGFP42, UbiRFP-CAAX* | 1 and 8 (i.e. 9 in total) | live |
| *Figure 2—figure supplement 1* | F | *sqh[AX3]; sqh::sqhGFP42, UbiRFP-CAAX* | 1 | live |
| *Figure 3—figure supplement 1* | B-F | *sqh[AX3]; sqh::sqhGFP42, UbiRFP-CAAX* | 5 | live |
| *Figure 3—figure supplement 2* | A-B' | *sqh[AX3]; sqh::sqhGFP42, UbiRFP-CAAX* | 5 | live |
| *Figure 4—figure supplement 1* | B,C | *sqh[AX3]; sqh::sqhGFP42, UbiRFP-CAAX* | 1 | live |
| *Figure 4—figure supplement 1* | E | *fkhGal4,UASpalm-YFP/Scribble-GFP and sqh[AX3]; sqh:: sqhGFP42, UbiRFP-CAAX* | 1 and 4 (i.e. 5 in total) | live |
| *Figure 5—figure supplement 1* | A,B | *sqh[AX3]; sqh::sqhGFP42, UbiRFP-CAAX* | 1 | live |
| *Figure 6—figure supplement 1* | | *fkhGal4,UASpalm-YFP/Scribble-GFP and sqh[AX3]; sqh:: sqhGFP42, UbiRFP* | 1 and 6 (i.e. 7 in total) | live |
| *Figure 8—figure supplement 1* | A-B' | *sqh::sqhGFP42* | panels representative of genotype (>30 embryos inspected) | fixed |
| *Figure 8—figure supplement 1* | D-F''' | *sqh::sqhGFP42, UbiRFP-CAAX; fkh[6]* | panels representative of genotype (>30 embryos inspected) | fixed |
| *Figure 8—figure supplement 2* | A-E' | *sqh::sqhGFP42, UbiRFP-CAAX fkh[6]* | 5 | live |

DOI: https://doi.org/10.7554/eLife.35717.046

## Analysis of signatures of active versus passive cell intercalation

Even though the quasi-3D analysis detailed above strongly indicates that the intercalation of cells far from the pit initiates from the apical domain, the process of initiation itself could be either actively driven or a passive response. Intercalation far from the pit would be active if it arose as an intrinsic property of this part of the tissue and actively drove circumferential contraction of the tissue (*Figure 6A*; red curved arrows). The capacity of these cells for such active behaviour would likely be achieved through genetic patterning. Intercalation would be passive if the active apical constriction near the pit drove a passive 'funnelling' of far cells towards the pit, the radial pull leading to passive polarised intercalations (*Figure 6A'*; blue arrows).

In order to distinguish between an active mechanism and a passive mechanism, we made four predictions (*Figure 6B–E'*), comparing metrics of junction length, angles at vertices, cell elongation and junctional myosin II accumulation. For each metric, we compared circumferentially oriented junctions with radial junctions. We also distinguished whether junctions were or were not shrinking junctions, about to be involved in a T1 event (see Materials and methods). We compared T1 data with similarly oriented non-T1 data to ask if there was a more active signature to shrinking junctions. We also compared radial versus circumferential non-T1 data to ask whether junctions as a whole in one orientation had a more active signature.

For the first two predictions (*Figure 6B–C'*), we compared real interface lengths and vertex angles with interface lengths and angles predicted by a Voronoi tessellation (see Materials and methods). We considered that a Voronoi tessellation generated from cell centroid seeds represents a mechanically neutral configuration for the cell-cell junctions and angles, and controlled for variation in local geometry around focal junctions. The direction in which real junction lengths or angles deviated from neutral Voronoi geometries was indicative of an active or a passive mechanism.

The deviation of real junction length from predicted Voronoi junction length (*Figure 6B,B'*) has previously been used as a geometric proxy for junction stress in the germband (*Tetley et al., 2016*). In the salivary gland placode, junction lengths were shorter than Voronoi predicted lengths for

shortening T1 junctions compared to non-T1 junctions, most strongly for circumferentially oriented junctions (*Figure 6F* and *Figure 6—figure supplement 1B*). Circumferential non-T1 interfaces also deviated from neutral geometries by being shorter on average than expected, with radial junctions longer (*Figure 6F*). This suggests that circumferential junctions, and circumferential T1s in particular, were contracted, possibly by an intrinsic contractile mechanism, rather than the cells being pulled away from each other radially by the pit.

We predicted that active circumferential junction contraction would impose more acute angles at their associated vertices (*Figure 6C*, [*Rauzi et al., 2008*]), whereas a radial pull would lead to more obtuse angles (*Figure 6C'*). On average, angles linked to circumferential junctions were indeed more acute, relative to neutral Voronoi geometry, than those linked to radial junctions (*Figure 6G* and *Figure 6—figure supplement 1C*). Similarly, cells were on average more elongated circumferentially (*Figure 6H* and *Figure 6—figure supplement 1D*) as predicted by actively contracting circumferential junctions and incompatible with a radial pull.

These three geometrical measures for cells far from the pit, where intercalation dominates, are therefore consistent with an active intercalation mechanism, leading to circumferential neighbour gains and thus circumferential convergence.

We also predicted that active circumferential junction contraction would be driven by myosin II accumulation at circumferential junctions (*Figure 6E*). Recent studies have indicated that in some tissues, as response to mechanical pulling, myosin II accumulates and becomes polarised at junctions parallel to the pulling force (*Duda et al., 2018*; *Fernandez-Gonzalez et al., 2009*). Hence if intercalation occurred as a passive response, we would expect either no myosin II polarisation or mechanically induced localisation to radially oriented junctions (*Figure 6E'*). The localisation and increase in myosin II observed at the central junctions during rosette formation/resolution in the placode, as shown above (*Figure 5G–I*), indicated that some junctional myosin II was circumferentially enriched. We now set out to analyse junctional myosin II distribution systematically across the whole placode.

## Myosin is enriched in circumferential junctions across the placode

When apical junctional myosin intensity was quantified in fixed samples of *sqhGFP* embryos across the early placode, a clear enrichment of myosin II was apparent in circumferentially compared to radially oriented junctions (*Figure 7A,B*). Such tissue-wide circumferential versus radial polarisation of myosin clearly supported an active mechanism of cell intercalation, with circumferential myosin II likely assisting both rosette and T1 vertex formation during the polarised cell intercalation events.

To compare and correlate junctional myosin II dynamics with the above strain rate analysis, we analysed myosin II polarisation dynamically across the whole tissue (*Figure 7C,C'*). Quantitative tools that allow junctional polarisation of myosin and other players to be quantified have previously been established (*Figure 7D,D''*; [*Tetley et al., 2016*]). Myosin II polarisation at circumferential junctions could either occur through enrichment at two opposite junctions or sides, termed bi-polarity (*Figure 7D'* and *Video 7*), or through enrichment at a single junction or side within a cell, termed unipolarity (*Figure 7D''* and *Video 7*). Starting about 10 min prior to tissue bending, bi-polar enrichment of myosin II at circumferential junctions was significantly stronger than at radial junctions when measured across the whole placode (*Figure 7E*). Similarly, starting ~ 5 min later, unipolar enrichment of myosin II at circumferential junctions dominated over radial enrichment when measured across the whole placode (*Figure 7F*). This clear and increasing circumferential junctional polarisation of myosin II together with the geometrical signatures discussed above strongly supported an active mechanism of T1 vertex and rosette formation (*Figure 6A*). Less clear is whether the resolution of rosettes and vertices is equally actively driven. Instead, this part of the intercalation events could respond to cell-extrinsic cues, such as the pulling of the invaginating pit, akin to the role of the posterior midgut invagination during germ band extension (*Collinet et al., 2015*; *Lye et al., 2015*) or pulling forces generated through micro-aspiration re-orienting intercalations in embryonic mouse tissues (*Wen et al., 2017b*).

In cell intercalation events during germband extension, myosin polarisation is complementary to enrichment of Par3/Bazooka (Baz) as well as Armadillo/β-catenin, with both enrichments controlled by the upstream patterning and positioning of transmembrane Toll receptors and Rho-kinase (Rok) (*Blankenship et al., 2006*; *Paré et al., 2014*; *Simões et al., 2010*). We therefore analysed whether such complementarity was also present in the early salivary gland placode. Antibody labelling of Baz often showed a complementarity in membrane enrichment to myosin II (*Figure 7G–I*), most

pronounced where myosin II was organised into circumferential mini cables during intercalation events (*Figure 7G,H*). Baz was enriched at radially oriented junctions where myosin II was low, and vice versa at circumferential junctions, though in contrast to germband extension, the Baz polarisation did not extend uniformly across the tissue and was overall less strong than the myosin II polarisation (*Figure 7I*).

Thus, in order to adapt to a circular tissue geometry and to the need for ordered invagination through a focal point during the process of tube budding, a conserved molecular pattern of myosin-Baz complementarity is apparently imprinted onto the salivary gland placode in a radial coordinate pattern, rather than the prevailing A-P/D-V pattern of the earlier embryo during germband extension (*Figure 7J,K*).

## Resolution but not initiation of cell intercalation is disrupted in the absence of Forkhead

In order to address how the radial pattern of behaviours and molecular factors across the placode is established, we analysed mutants in a key factor of salivary gland tube invagination, the transcription factor Fork head (Fkh). Fkh is expressed just upon specification of the placodal cells in a dynamic pattern spreading across the whole placode (*Figure 8—figure supplement 1A–C*), directly downstream of the homeotic factor Scr (*Zhou et al., 2001*). In *fkh* mutants, invagination of the placode fails, and towards the end of embryogenesis salivary gland-fated cells undergo apoptosis as Fkh appears to prevent activation of pro-apoptotic factors (*Jürgens and Weigel, 1988*; *Myat and Andrew, 2000a*). Previous studies have concluded that Fkh promotes cell shape changes important for invagination, in particular apical constriction (*Chung et al., 2017*; *Myat and Andrew, 2000a*). Fkh is not the only transcription factor important for correct changes during invagination, but works in parallel to for example Huckebein (Hkb), the lack of which also confers significant problems with apical cell shape changes and invagination (*Myat and Andrew, 2000b*, *2002*).

We combined a *fkh* null mutant (*fkh[6]*) with markers allowing membrane labeling and cell segmentation (*Ubi-TagRFP-CAAX*) as well as myosin II quantification (*sqhGFP*; see *Video 8*). When *fkh* mutant placodes were compared to wild-type ones at late stage 11 (beyond the time window analysed here) then *fkh* mutant placodes showed no sign of a dorsal-posterior invagination point (*Figure 8A–B'*; *Figure 8—figure supplement 1D–E''*). In fact, many placodes beyond late stage 11 showed a centrally located shallow depression (*Figure 8A,A'*, yellow dotted lines). Strain rate analysis of five segmented movies of *fkh[6]* mutant placodes, spanning the equivalent time period to the wild-type movies analysed above (*Figure 8—figure supplement 2C*), showed that there was no constriction at the tissue level near the pit (*Figure 8C–E* and *Figure 8—figure supplement 2D,D'*). In fact, if anything, there was a slight tissue expansion (*Figure 8E*) caused by a slight expansion at the cell level (*Figure 8E'*), with zero intercalation (*Figure 8E''*). Away from the pit *fkh[6]* mutant placodes expanded slightly (*Figure 8F* and *Figure 8—figure supplement 2E,E'*), again mostly due to cell shape changes (*Figure 8F', F''*).

Is this strong reduction of cell behaviours due to a complete 'freezing' of the placode in the *fkh[6]* mutants? We analysed whether junctional myosin II was still polarised in the *fkh[6]* mutant placodes. In fixed and live samples, *fkh[6]* mutant placodes still showed the circumferential actomyosin cable surrounding the placode (*Figure 8—figure supplement 1E''*, compare to D''; [*Röper, 2012*]). When myosin II unipolarity and bi-polarity were quantified from five segmented and tracked movies, at the tissue-level there was a strong and significant reduction in myosin II polarisation, with no difference between radial and circumferential orientations in either measure and myosin II bi-polarity indistinguishable from zero (*Figure 9A,B*). Nonetheless, cells were not static but in fact neighbour exchanges were present, both in form of T1 exchanges and rosettes (*Figure 9C–F*). The quantitative analysis of neighbour gains from segmented and tracked movies in the *fkh[6]* mutant revealed that gains accumulated both circumferentially as well as radially, at a rate comparable to that observed in the control (*Figure 9E*). However, in contrast to the control, where circumferential gains significantly outweighed radial neighbour gains leading to a positive net rate of productive circumferential gains (*Figure 9F*, green line), in the *fkh[6]* mutant circumferential and radial gains occurred in equal amounts (*Figure 9E*) leading to a near zero net gain (*Figure 9F*, pink line). Interestingly, when focusing on individual events such as rosettes, despite a loss of tissue-wide myosin II polarisation, myosin II was still enriched at the central constricting junctions (*Figure 9Cd'*), and in fixed embryos smaller regions of myosin II-Baz complementarity could be identified (*Figure 8—figure supplement 1F–G*).

Therefore, although at the tissue level *fkh[6]* mutant placodes appear near static, the close analysis of individual events revealed a highly dynamic but unpolarised intercalation behaviour across the mutant placodes. Without an actively invaginating pit and focussed apical constriction taking place in the *fkh[6]* mutant, the unpolarised intercalation events are not being resolved radially because of the absence of the pull from the pit. This finding therefore also suggests an active intercalation mechanism, where the remaining local increases in junctional myosin II still support formation of T1 vertices and rosettes, but without any overall directionality to their resolution.

## Discussion

Morphogenesis sculpts many differently shaped tissues and structures during embryogenesis. A core set of molecular factors that are the actual morphogenetic effectors, such as actomyosin allowing contractility or cell-cell adhesion components allowing coordination and mechanical propagation of cell behaviours across tissues, are used iteratively in different tissues and at different times. By contrast, the activity of upstream activating gene regulatory networks leading to tissue identity, but also initial tissue geometry and mechanical constraints, are highly tissue-specific.

During tube formation of the salivary glands in the fly embryo, we observe a clear tissue-level radial organisation of cell behaviours, with the off-centre located invaginating pit as the organising focal point (*Figure 10A*). When analysed in 2D within the apical domain of the placodal epithelial cells, apical constriction dominates at the future pit of invagination and directional cell intercalation dominates further away from the pit. This intercalation achieves a circumferential convergence and radial extension of the tissue towards the invagination point (*Figure 10B*). Interestingly, these cell behaviours (cell shape change and cell intercalation) have previously been shown to drive other morphogenetic processes (*Butler et al., 2009*; *Collinet et al., 2015*; *Lee and Harland, 2007*; *Lye et al., 2015*; *Martin and Goldstein, 2014*; *Martin et al., 2009*; *Plageman et al., 2011*; *Rauzi et al., 2010*), but in our system they are utilised within a radial coordinate system, with the morphogenetic outcome being the formation of a narrow tube of epithelial cells from a round and flat placode primordium.

In order to understand complex organ formation, it is important to understand cause and effect during the process. Cell behaviours can either be actively driven through for instance patterned actomyosin activity or they can be a mechanical response to events, or a combination of the two. An excellent example for active behaviours intersecting and being influenced by nearby events is the extension of the germband during *Drosophila* embryogenesis. In this tissue, an active mechanism of polarised intercalation combines with an extrinsic pulling force from the invaginating posterior midgut that helps the directional resolution of T1s (*Butler et al., 2009*; *Collinet et al., 2015*; *Lye et al., 2015*). During tube formation of the salivary gland placode, a similar intersection of active and passive mechanisms could be taking place: actively initiated intercalations combine with an active apical constriction at the pit that polarises the resolution of the exchanges (*Figure 10C,D*). The ongoing but unproductive intercalations in the *fkh* mutant, that lack directional formation and resolution, support the notion that the pulling of the constricting and invaginating pit reinforces the directionality of resolution in the wild type. The setup of active constriction and active intercalation in adjacent regions combined with some aspect of mechanical coupling between the two would thus be similar to the germband and posterior midgut, although with the placode being radially organised and the germband axially patterned. Thus, our work suggests the existence of iteratively used morphogenetic mechanisms that are highly adaptable to a particular tissue geometry and size. Compared to germband extension (*Tetley et al., 2016*), the signatures of active apical intercalation behaviour analysed here are reduced in magnitude. In particular, interfaces contract much more strongly away from the neutral Voronoi geometry in the dorso-ventral axis in the germband, compared to the circumferential axis in the placode. Furthermore, individual interfaces approaching a T1 are not significantly different from similarly oriented interfaces in the placode, whereas the strongly increasing fluorescence density of myosin associated with junction shortening in the germband distinguishes the geometry of these interfaces from non-T1 interfaces. This could be due to the overall much smaller size of the tissue, with changes accumulating at a tissue level that restrict further drastic deformation prior to a T1. This aspect can be addressed in future studies comparing even more active intercalation events in other tissues or through modelling approaches. It will also be crucial to

determine what underlies the patterning of accumulation of the different myosin pools across the placode that likely allows the mechanical coupling of behaviours (*Figure 10B*).

The quasi-3D analysis comparing apical to basal sections also revealed a radial organisation of cell behaviours, specifically cell wedging, tilting and interleaving (*Figure 10C,D*). The constriction near the pit leads to strong cell wedging, with less wedging at a distance from the pit. Also in 3D, cells are tilting towards the point of invagination, with more tilt at a distance. In addition, interleaving of cells (the continuous change in arrangement of cells along their basal to apical length that can be but is not necessarily associated with discrete changes in neighbour connectivity in depth), occurs across the placode and is compatible with active apical circumferential convergence and radial extension of the tissue overall. Thus, our work revealed novel patterns of cell behaviours that could only be uncovered by considering the 3D context of a developing tissue.

Non-equilibrium 3D cell geometries in a flat epithelium, such as those caused by the wedging, interleaving and tilting analysed here, evolve during early placode morphogenesis in revealing patterns (*Figure 10C,D*). The appearance and progressive increase of these out-of-equilibrium geometries in the placode precede the start of 3D pit invagination by more than 10 min. This could suggest that during this phase a pre-pattern of tension could build up across the placode that would make the pit invagination more efficient, once initiated. Some of the cell behaviours we observe and the resulting complex 3D shapes might then be the integrated results of the balance of forces in the changing mechanical context of the placode. This will be possible to test experimentally by interference with certain behaviours or through ectopic induction of others, and can also be tested in silico in the future.

Our strain rate analysis in depth has revealed many interesting features of epithelial geometry and behaviour. What is still quite unclear during epithelial morphogenesis is whether morphogenetic behaviours are always initiated apically and propagated basally, or whether there is in fact an active contribution through events initiated at lateral or basal sides. A few recent reports suggest that not all is apically initiated (*Monier et al., 2015*; *Sun et al., 2017*), but whether this is a general principle or highly tissue-specific is unclear. It is also unclear how any cell behaviour, whether initiated apically or basally, is communicated and propagated across the length of the cell. In many morphogenetic processes including the tube budding from the salivary gland placode, actomyosin and other morphogenetic effectors are concentrated within the apical junctional domain. Our data at depth reveal that there is close coordination of intercalation rates between apical and mid-basal levels, and that the apically led interleaving and progressive wedging strongly support an active apical mechanism that is followed further basally.

In summary, our work uncovers a dynamic interplay of highly patterned cell behaviours within a radially organised tissue. Future research will show whether such radial patterning of myosin and cell behaviours is conserved across other tube-forming tissue primordia.

## Materials and methods

**Key resources table**

| Reagent type (species) or resource | Designation | Source or reference | Identifiers | Additional information |
|---|---|---|---|---|
| Gene (*D. melanogaster*) | Fork Head | NA | FLYB:FBgn0000659 | |
| Gene (*D. melanogaster*) | non-muscle myosin II/sqh | NA | FLYB:FBgn0003514 | |
| Gene (*D. melanogaster*) | Bazooka/Par3 | NA | FLYB:FBgn0000163 | |
| Genetic reagent (*D. melanogaster*) | Scribble-GFP | Kyoto Drosophila Genomic Research Centre | | |
| Genetic reagent (*D. melanogaster*) | fkh-Gal4 | PMID:10625560 | | |

*Continued on next page*

*Continued*

| Reagent type (species) or resource | Designation | Source or reference | Identifiers | Additional information |
|---|---|---|---|---|
| Genetic reagent (*D. melanogaster*) | Fkh-Gal4::UAS-palmYFP | PMID:10625560, PMID:21297621 and this study | | stock generated upon recombination of Brainbow Cassette stock |
| Genetic reagent (*D. melanogaster*) | sqh[AX3]; sqh::sqhGFP42 | PMID:14657248 | | |
| Genetic reagent (*D. melanogaster*) | sqh::sqhGFP42, UbiRFP-CAAX | Kyoto Drosophila Genomic Research Centre Number 109822 | | |
| Genetic reagent (*D. melanogaster*) | fkh[6] | Bloomington Drosophila Stock Center, PMID:2566386 | FLYB:FBal0004012 | |
| Antibody | anti-DE-Cadherin (rat monoclonal) | Developmental Studies Hybridoma Bank at the University of Iowa | DSHB:DCAD2 | (1:10) |
| Antibody | anti-Crumbs (mouse monoclonal) | Developmental Studies Hybridoma Bank at the University of Iowa | DSHB:Cq4 | (1:10) |
| Antibody | anti-Bazooka (rabbit polyclonal) | PMID:10591216 | | (1:500) |
| Antibody | anti-Forkhead (guinea pig plyclonal) | PMID:2566386 | | (1:2000) |
| Antibody | Alexa Fluor 488/549/649-coupled secondary antibodies | Molecular Probes | | |
| Antibody | Cy3-, Cy5- coupled secondary antibodies | Jackson Immuno Research | | |
| Software | otracks | PMID:19412170, PMID:24914560 | | Software file (custom software written in IDL) |
| Software | *nd-safir* | PMID:19900849 | | Denoising algorithm. Available at http://serpico.rennes.inria.fr/doku.php?id=software:nd-safir:index |

## Fly stocks and husbandry

The following transgenic fly lines were used: *sqhAX3; sqh::sqhGFP42* (*Royou et al., 2004*) and *fkhGal4* (*Henderson and Andrew, 2000*; *Zhou et al., 2001*) [kind gift of Debbie Andrew]; *Scribble-GFP* (DGRC Kyoto), *UAS-palmYFP* (generated from membrane Brainbow; [*Hampel et al., 2011*]), $y^1$ $w^* cv^1 sqh^{AX3}$; P{$w^{+mC}$ = sqh-GFP.RLC}C-42 M{$w^{+mC}$ = Ubi-TagRFP-T-CAAX}ZH-22A (Kyoto DGRC Number 109822, referred to as $sqh^{AX3}$;sqhGFP; UbiRFP); P{w + mC = sqh-GFP.RLC}C-42 M {w + mC = Ubi-TagRFP-T-CAAX}ZH-22A; fkh[6]/TM3 Sb Twi-Gal4::UAS-GFP (fkh[6] allele from Bloomington). See *Table 1* for details of genotypes used for individual figure panels.

## Embryo immunofluorescence labelling, confocal, and time-lapse

Embryos were collected on apple juice-agar plates and processed for immunofluorescence using standard procedures. Briefly, embryos were dechorionated in 50% bleach, fixed in 4% formaldehyde, and stained with primary and secondary antibodies in PBT (PBS plus 0.5% bovine serum albumin and 0.3% Triton X-100). anti-Crumbs and anti-E-Cadherin antibodies were obtained from the Developmental Studies Hybridoma Bank at the University of Iowa; anti-Baz was a gift from Andreas Wodarz (*Wodarz et al., 1999*); anti-Fkh was a gift from Herbert Jäckle (*Weigel et al., 1989*). Secondary antibodies used were Alexa Fluor 488/Fluor 549/Fluor 649 coupled (Molecular Probes) and Cy3 and Cy5 coupled (Jackson ImmunoResearch Laboratories). Samples were embedded in Vectashield (Vectorlabs).

Images of fixed samples were acquired on an Olympus FluoView 1200 or a Zeiss 780 Confocal Laser scanning system as z-stacks to cover the whole apical surface of cells in the placode. Z-stack projections were assembled in ImageJ or Imaris (Bitplane), 3D rendering was performed in Imaris.

For live time-lapse experiments embryos from [*Scribble-GFP,UAS-palmYFP fkhGal4*], [*sqh^AX3*; *sqhGFP,UbiRFP*] or [*sqhGFP,UbiRFP; fkh[6]*] were dechorionated in 50% bleach and extensively rinsed in water. Embryos were manually aligned and attached to heptane-glue coated coverslips and mounted on custom-made metal slides; embryos were covered using halocarbon oil 27 (Sigma) and viability after imaging after 24 h was controlled prior to further data analysis. Time-lapse sequences were imaged under a 40x/1.3NA oil objective on an inverted Zeiss 780 Laser scanning system, acquiring z-stacks every 0.8–2.6 min with a typical voxel xyz size of 0.22 × 0.22 × 1 µm. Z-stack projections to generate movies in Supplementary Material were assembled in ImageJ or Imaris. The absence of fluorescent *Twi-Gal4::UAS-GFP* was used to identify homozygous *fkh[6]* mutant embryos. During the early stages of salivary gland placode morphogenesis analysed here, *fkh[6]* mutants showed no reduction in cell number or initiation of apoptosis (data not shown). The membrane channel images from time-lapse experiments were denoised using *nd-safir* software (*Boulanger et al., 2010*).

## Cell segmentation and tracking

Cell tracking was performed using custom software written in IDL (code provided in (*Blanchard et al., 2009*) or by email from G.B.B.). First, the curved surface of the embryonic epithelium was located by draping a 'blanket' down onto all image volumes over time, where the pixel-detailed blanket was caught by, and remained on top of binarised cortical fluorescence signal. Different quasi-2D image layers were then extracted from image volumes at specified depths from the surface blanket. We took image layers at 1–3 and 7–8 µm for the apical and mid-basal depths, respectively. Image layers were local projections of 1–3 z-depths, with median, top-hat or high/low frequency filters applied as necessary to optimise subsequent cell tracking.

Cells in image layers at these two depths were segmented using an adaptive watershedding algorithm as they were simultaneously linked in time. Manual correction of segmented cell outlines was performed for all fixed and time-lapse data. The segmentation of all the movies used in this study was manually corrected to ensure at least 90% tracking coverage of the placode at all times. Tracked cells were subjected to various quality filters (lineage length, area, aspect ratio, relative velocity) so that incorrectly tracked cells were eliminated prior to further analysis. The number of embryos analysed and number of cells can be found in *Tables 1* and *2* (also see *Figure 2—figure supplement 1* (WT apical), *Figure 3—figure supplement 1* (WT basal) and *Figure 8—figure supplement 1 (f, K, h)*).

## Mobile radial coordinate system for the salivary placode

WT movies were aligned in time using as t = 0 min the frame just before the first sign of invagination of cell apices at the future tube pit was evident. *fkh[6]* mutants were aligned using as a reference of embryo development the level of invagination of the tracheal pits that are not affected in the *fkh[6]* mutant as well as other morphological markers such as appearance and depth of segmental grooves in the embryo. Cells belonging to the salivary placode (without the future duct cells that comprise the two most ventral rows of cells in the primordium) were then manually outlined at t = 0 mins using the surrounding myosin II cable as a guide and ramified forwards and backwards in time. Only cells of the salivary placode were included in subsequent analyses.

At t = 0 min, the centre of the future tube pit was specified manually as the origin of a radial coordinate system, with radial distance (in µm) increasing away from the pit (e.g. *Figure 1G*). Circumferential angle was set to zero towards Posterior, proceeding anti-clockwise for the placode on the left-hand side of the embryo, and clockwise for the placode on the right so that data collected from different sides could be overlaid.

The radial coordinate system was 'mobile', in the sense that its origin tracked the centre of the pit, forwards and backwards in time, as the placode translated within the field of view due to embryo movement or to on-going morphogenesis.

## Morphogenetic strain rate analysis

Detailed spatial patterns of the rates of deformation across the placode and over time quantify the outcome of active stresses, viscoelastic material properties and frictions both from within and outside the placode. We quantified strain (deformation) rates over small spatio-temporal domains

composed of a focal cell and one corona of immediate neighbours over a ~5 min interval ([*Blanchard et al., 2009*] and reviewed in [*Blanchard, 2017*]). On such 2D domains, strain rates are captured elliptically, as the strain rate in the orientation of greatest absolute strain rate, with a second strain rate perpendicular to this (*Figure 2B*).

For the early morphogenesis of the salivary gland placode, in which there is no cell division or gain/loss of cells from the epithelium, three types of strain rate can be calculated. First, total tissue strain rates are calculated for all local domains using the relative movements of cell centroids, extracted from automated cell tracking. This captures the net effect of cell shape changes and cell rearrangements within the tissue, but these can also be separated out. Second, domain cell shape strain rates are calculated by approximating each cell with its best-fit ellipse and then finding the best mapping of a cell's elliptical shape to its shape in the subsequent time point, and averaging over the cells of the domain. Third, intercalation strain rates that capture the continuous process of cells in a domain sliding past each other in a particular orientation, is calculated as the difference between the total tissue strain rates and the cell shape strain rates of cells. Strain rates were calculated using custom software written in IDL (code provided in (*Blanchard et al., 2009*) or by email from G.B.B.).

The three types of elliptical strain rate were projected onto our radial coordinate system (see *Figure 1F*), so that we could analyse radial and circumferential contributions. Strain rates in units of proportional size change per minute can easily be averaged across space or accumulated over time. We present instantaneous strain rates over time for spatial subsets of cells in the placode, and cumulative strain ratios for the same regions over time. These plots were made from exported data using MATLAB R2014b.

We calculated strain rates in layers at two depths for WT placodes. Because we applied a radial coordinate system originating in the centre of the future tube pit to both depths, and used the same reference t = 0 min, we were able to compare strain rates at the same spatio-temporal locations on the placode between the depths.

## Matching cells between apical and mid-basal layers

We also wanted to characterise the 3D geometries of cells within small domains, using the cell shapes and arrangements in the two depths we had tracked. To do this, we needed to correctly match cells between apical and mid-basal depths. We manually seeded 3–5 apico-basal cell matches per placode, then used an automated method to fill out the remaining cell matches across the placode and over time.

We did this by sequentially looking at all unmatched apical cells that were next to one or more matched cell. The location of the unmatched basal centroid was predicted by adding the vector between matched and unmatched apical cell centroids to the matched basal centroid. The nearest actual basal centroid to this predicted basal centroid location was chosen as the match, on condition that the prediction accuracy was within 0.25 of the apical centroid distance. Information from multiple matched neighbours was used, if available, improving the basal centroid prediction. Progressively matching cells out from known matched cells filled out placodes for all embryos. We visually checked apical-basal matches for each embryo in movies of an overlay of apical and basal cell shapes with apico-basal centroid connections drawn (*Video 3*).

## z-strain rates: 3D domain geometries

Having matched cells between apical and mid-basal layers, we have access to information about approximate 3D cell shapes. For single cells, we can measure how wedged the cell shape is in any orientation and how tilted is the apical centroid to basal centroid 'in-line' relative to the surface normal (*Figure 4A*). In particular, we are interested in the amount of wedging and tilt in our radial and circumferential placode orientations (*Figure 4A*). However, an important aspect is missing which is that cells can be arranged differently apically versus basally, independently of any cell wedging. That is they can be to a greater or less extent interleaved. Interleaving in depth is completely analogous to cell intercalation in time (*Figure 4—figure supplement 1D*). The degree of interleaving is a multicell phenomenon, so we return to our small domains of a cell and its immediate neighbours. Within these small domains we want to quantify the amount of cell wedging, interleaving and cell tilt.

To correctly separate these quantities, we borrow from methods developed to separate out the additive contributions of wedging, interleaving and tilt that account for epithelial curvature (*Deacon, 2012*). Deacon shows that for small domains of epithelial cells, curvature across the domain is the sum of their wedging and interleaving, while cell tilt has no direct implication for curvature.

During our study period, salivary placodes have minimal curvature (average curvature is less than $0.05 \ \mu m^{-1}$ in both AP and DV axes). We therefore simplify the problem to flat (uncurved) domains, uncurving any local curvature so that apical and basal cell outlines are flat and parallel. Usefully, we can then treat small domains in exactly the same way as we have done above to calculate strain rates above, but instead of quantifying the rate of deformation over time, here we calculate rate of deformation in depth, or 'z-strain rates' (*Figure 4—figure supplement 1D*).

The cell shape strain rate is equivalent to the wedging z-strain rate (*Figure 4B*) and the intercalation strain rate is the interleaving z-strain rate (*Figure 4D*). Both of these add up to a total or tissue strain rate and a total or tissue z-strain rate (*Figure 4—figure supplement 1E*), respectively. Note that the total z-strain rate of a domain can be the result exclusively of wedging or of interleaving (as in *Figure 4B,D*; as with temporal strain rates, *Figure 2A*), but some combination of the two is more likely. Domain translation and rotation for temporal domains become domain tilt (*Figure 4F*) and twist in the 3D domain geometries, respectively. Domain twist is very weak in spatial or temporal averages of the placode (data not shown).

The units of wedging and interleaving are proportional size change per $\mu m$ in z, and tilt is simply a rate (xy $\mu m$/z $\mu m$). We use the convention that change is from basal to apical, so a bottle-shape has negative cell wedging. The number of embryos and cells analysed for 3D cell geometries can be found in *Tables 1* and *2*.

## Neighbour exchange analysis

We used changes in neighbour connectivity in our tracked cell data to identify neighbour exchange events (T1 processes). Neighbour exchange events were defined by the identity of the pair of cells that lost connectivity in *t* and the pair that gained connectivity at *t + 1*. The orientation of gain we defined as the orientation of the centroid-centroid line of the gaining pair at *t + 1*. We further classified gains as either radially or circumferentially oriented, depending on which the gain axis was most closely aligned to locally. We did not distinguish between solitary T1s and T1s involved in rosette-like structures.

From visual inspection, we knew that some T1s were subsequently reversed, so we characterised not only the total number of gains in each orientation but also the net gain in the circumferential axis, by subtracting the number of radial gains. Furthermore, when comparing embryos and genotypes, we controlled for differences in numbers of tracked cells by expressing the net circumferential gain per time step as a proportion of half of the total number of tracked cell-cell interfaces in that time step. We accumulated numbers of gains, net gains, and proportional rate of gain over time for WT (*Figure 5C*) and *fkh* (*Figure 9E,F*) embryos. Two sample Kolmogorov-Smirnov test was used to determine significance at p<0.05 for data in *Figure 9F*.

## Calculation of signatures of active versus passive intercalation

In the region of the placode away from the pit, we considered that active circumferential intercalation driven by junctional shortening would produce a different signature in the geometries of cells from a pull from the actively constricting cells at the pit (as depicted in *Figure 4A,A'*). We took an approach that avoided using simplistic raw measures of junction lengths and angles as these do not take into account differences in the geometry of cells immediately surrounding a junction.

First, we considered that junctions approaching a T1-transition that are being actively shortened by myosin II motors would be shorter than expected according to some neutral geometry. We used a Voronoi tessellation based on actual cell centroids as our reference neutral geometry, allowing us to compare actual junction lengths with predicted neutral lengths (as used in [*Tetley et al., 2016*]). The Voronoi tessellation provides cell junctions as the set of points equidistant between neighbouring cell centroids (see magenta dashed lines in *Figure 6B,B'*). We calculated the deviation from Voronoi in $\mu m$, with a negative value indicating that junctions were likely to be actively shortened. A positive value would indicate a passively elongated junction. In practice, placode cells (and germ-band cells, see *Figure 5D–F* in [*Tetley et al., 2016*]) are less ordered than a Voronoi tessellation,

with the result that cell-cell interfaces are on average longer than tessellated interfaces. The baseline average value is therefore positive and not zero (see dashed line in *Figure 6F*). In principle, it would be better to generate neutral geometries using a mechanical model but that would require a further set of untested assumptions about the rheology of the placode tissue. We chose the Voronoi tessellation because it offers a very simple way of generating a plausible neutral geometry.

Second, we measured the vertex angles in the cells at opposite ends of focal junctions (see *Figure 6C,C'*). Raw angles would be expected to reduce from 120° at three-way junctions to 90° at a four-way T1-transition. However, again to control for variation in local cell geometry, we compared angles relative to those predicted by a Voronoi tessellation constructed from cell centroid seeds. Our measure is therefore the angular deviation from the Voronoi tessellation, in degrees, with a negative value indicating a narrower angle, which we would interpret as being drawn out by an actively contracting junction. Our angular deviation measure is measured as the average for the two vertices at either end of a focal junction.

Third, we calculated the elongation of the pair of cells at either end of a focal junction (*Figure 6D,D'*). Again, we considered that an actively shrinking focal junction will tend to elongate these cells, whereas a contractile pit would elongate cells in the radial direction towards the pit.

We classified junctions as being in the run up to a T1 (T1 data) or not involved in T1 (baseline data), and according to their (radial or circumferential) orientation and asked two types of question. We compared circumferential baseline to radial baseline data, asking if there was an overall active signature bias in the tissue. We also compared T1 data to baseline data for similarly oriented junctions, asking whether junctions involved in a T1 have a specific effect on local geometries.

For statistical tests, we again employed the mixed-effects models (see Statistics section below). In order to avoid using temporally correlated data in the comparisons we selected one time point for each data set we compared. For baseline data, we could choose any developmental time point because the baseline data for all three measures did not vary with developmental time (*Figure 6—figure supplement 1B–D*). We therefore chose the time point −2–0 min from the start of pit invagination, which maximised the number of placodes (see *Figure 2—figure supplement 1A*) and hence N cells. For T1 data we chose −3 min from neighbour exchange, near the middle of the time window before exchange where junction length proceeds consistently to extinction (*Figure 6—figure supplement 1A*).

## Fluorescence intensity quantifications of Myosin and Bazooka

Embryos of the genotype *sqhAX3; sqh::sqhGFP42* (*Royou et al., 2004*) were labelled with either with anti-Bazooka and anti-DE-Cadherin or anti-DE-Cadherin and phalloidin to highlight cell membranes. For *Figure 7B*, myosin II signal intensity of junctions oriented either circumferentially or radially from five placodes was quantified using Image J. For *Figure 7I* fluoresce intensity of myosin II and Bazooka of junctions such as the ones indicated by arrows in *Figure 7H'*, that were oriented either circumferentially or radially from 10 placodes was quantified using Image J. 3-pixel wide lines were manually drawn along each junction. Intensity values were normalised to average fluorescence outside the placodes. We used a paired t-test for comparison of intensities within in the same junction, unpaired t-tests for comparison of circumferential myosin II vs radial myosin II and circumferential Baz vs radial Baz.

## Automated myosin II quantification and polarity

Whereas previously we quantified apicomedial Myosin II (*Booth et al., 2014*), here we focused on junctional Myosin II. We extracted a quasi-2D layer image from the Myosin II channel at a depth that maximised the capture of the junctional Myosin. We background-subtracted the Myosin images and quantified the average intensity of Myosin along each cell-cell interface within the placode. To calculate the average intensity, we set the width of cell-cell interfaces as the cell edge pixels plus two pixels in a perpendicular direction either side. This captured the variable width of Myosin signal at interfaces.

We further summarised the uni- and bipolarity of Myosin for each cell. Methods to calculate Myosin II unipolarity and bipolarity are described in detail in *Tetley et al. (2016)*. Briefly, the average interface fluorescence intensity around each cell perimeter as a function of angle is treated as a periodic signal and decomposed using Fourier analysis. The amplitude component of period two

corresponds to the strength of Myosin II bipolarity (equivalent to planar cell polarity) (*Figure 7D'*). Similarly, the amplitude of period one is attributed to myosin unipolarity (junctional enrichment in a particular interface) (*Figure 7D''*). The extracted phase of periods 1 and 2 (bi/uni-polarity) represent the orientation of cell polarity. We projected both polarities onto our radial coordinate system. Both polarity amplitudes are expressed as a proportion of the mean cell perimeter fluorescence. Cells at the border of the placode neighbouring the supra-cellular actomyosin cable were excluded from the analysis. The number of embryos and number of cells analysed can be found in *Tables 1* and *2*.

## Statistical analysis of time-lapse data

Statistical tests to determine significance of data shown are indicated in the figure legends. Significance in time-lapse movies was calculated for bins of 4.5 min using a mixed-effects model implemented in R ('lmer4' package as in [*Butler et al., 2009*; *Lye et al., 2015*]) with a significance threshold of $p < 0.05$.

A mixed-effect model has fixed effects and random effects. The former are variables associated with an entire population and are expected to have an effect on the dependent variable. Random effects are factors which are associated with individual experimental units drawn at random from a population and introduce variation that is desirable to account for (*Pinheiro and Bates, 2000*). Here, to test for differences in instantaneous strain rates we used as fixed effect the genotypes (wild type and *forkhead*) while the variation between embryos (from the same genotype) was considered a random effect. The advantage of using a linear mixed-effect model is that this framework allows testing for differences on certain variables while accounting for sources of variation that are present in the full data set.

In the main figures we generally present cumulative strains, which are generated from instantaneous strain rate data shown in Figure supplements. The cumulative strains are calculated only from instantaneous data averages and so do not carry any of the distributions that were used to calculate instantaneous means and associated confidence intervals. We therefore perform all statistics on the instantaneous data, using the mixed-effects model above. For mixed-effects models, one cannot portray a single overall confidence interval. Instead, we have a choice to show one of within- or between-genotype confidence intervals, and we have chosen the former, as has been done previously (for example in [*Butler et al., 2009*; *Gorfinkiel et al., 2009*; *Lye et al., 2015*; *Tetley et al., 2016*]).

Therefore, error bars in time-lapse plots show an indicative confidence interval of the mean, calculated as the mean of within-embryo variances. The between-embryo variation is not depicted, even though both are accounted for in the mixed effects tests.

## Acknowledgements

The authors thank the following people; for reagents and fly stocks: Debbie Andrew, Andreas Wodarz, Herbert Jäckle; for help with image analysis: Jerôme Boulanger. Work in the Röper lab is supported by the Medical Research Council (file reference number U105178780). GBB was supported by grant no. 15.23(k) from the Isaac Newton Trust, by Wellcome Trust grant no. 100329/Z/12/Z awarded to William Harris and Biotechnology and Biological Sciences Research Council Standard Grant BB/J010278/1 to Richard Adams and Bénédicte Sanson. GBB also thanks Wellcome Trust Investigator Award [099234/Z/12/Z] to Bénédicte Sanson and the Department of Physiology, Development and Neuroscience, University of Cambridge, UK.

## Additional information

### Funding

| Funder | Grant reference number | Author |
| --- | --- | --- |
| Medical Research Council | U105178780 | Yara E Sanchez-Corrales<br>Guy B Blanchard<br>Katja Röper |
| Biotechnology and Biological Sciences Research Council | BB/J010278/1 | Guy B Blanchard |

| Wellcome | 100329/Z/12/Z | Guy B Blanchard |
|---|---|---|
| Isaac Newton Trust | 15.23(k) | Guy B Blanchard |
| Wellcome | 099234/Z/12/Z | Guy B Blanchard |

The funders had no role in study design, data collection and interpretation, or the decision to submit the work for publication.

## Author contributions

Yara E Sanchez-Corrales, Data curation, Formal analysis, Investigation, Visualization, Methodology, Writing—original draft; Guy B Blanchard, Conceptualization, Software, Methodology, Writing—original draft; Katja Röper, Conceptualization, Formal analysis, Supervision, Funding acquisition, Validation, Investigation, Visualization, Methodology, Writing—original draft

## Author ORCIDs

Yara E Sanchez-Corrales http://orcid.org/0000-0003-1438-1994
Guy B Blanchard http://orcid.org/0000-0002-3689-0522
Katja Röper http://orcid.org/0000-0002-3361-766X

## Decision letter and Author response

Decision letter https://doi.org/10.7554/eLife.35717.052
Author response https://doi.org/10.7554/eLife.35717.053

## Additional files

### Supplementary files
• Transparent reporting form
DOI: https://doi.org/10.7554/eLife.35717.047

### Data availability
All data generated or analysed during this study are included in the manuscript and supporting files. Source data files have been provided for all figures.

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
