## [Decision Letter]

[Editors’ note: the authors were asked to provide a plan for revisions before the editors issued a final decision. What follows is the editors’ letter requesting such plan.]

Thank you for sending your article entitled "Radially-polarised cell behaviours drive tube budding from an epithelium" for peer review at *eLife*. Your article is being evaluated by four peer reviewers, one of whom is a member of our Board of Reviewing Editors, and the evaluation is being overseen by Didier Stainier as the Senior Editor.

Given the list of essential revisions, including new experiments, the editors and reviewers invite you to respond within the next two weeks with an action plan and timetable for the completion of the additional work. We plan to share your responses with the reviewers and then issue a binding recommendation.

The full reviews are also included for your reference, as they contain detailed and useful suggestions. As you will see, the reviewers clearly find interest in the description of radially polarized cell behaviors during tube invagination. However, a number of issued and concerns are raised, a major one regarding the causal relationship between radially organized cell behavior and pit invagination.

The reviewers raised the following main points:

1) Causality. Cause and consequence are not clear with respect to the role of polarized cell behaviors in tube budding. Radially polarized intercalation of cells surrounding the pit may be, at least in part, a consequence of forces generated by the invaginating pit, rather than vice versa. Additional experiments would be required to dissect pit formation on the one hand and peripheral intercalation and myosin accumulation on the other hand. For instance, optogenetic tools might be used to trigger apical constriction and an ectopic invagination pit. This may allow to test whether intercalation behavior and myosin pattern respond to these changes, supporting a mechanical model, or if they do not respond, supporting intrinsic genetic control. If experimental evidence supporting such a causal role cannot be provided, claims about causality (including in the title) need to be diminished accordingly.

2) The analysis of *forkhead* mutants is interesting, as it reveals cell behavior in the absence of an invagination pit. However, it contributes little, if anything, to the resolution of issues related to causality. This should be discussed and made clearer in the text.

3) The analysis of cell behaviors is not truly "3D", as stated in the manuscript, but in fact cell shape is analyzed in 2D at two different levels along the apical-basal axis of the cells. An analysis of shape changes in 3D, and in particular of cell volume, would be more informative and would qualify as a "3D" analysis.

4) In the first part of the work, cell intercalation is inferred only indirectly from tissue behavior across a field of cells. Even though such measures are interesting to assess the relative contribution of cell shape and intercalation, the data would be more convincing if data are visualized per cell and if cell intercalation was assessed directly by quantifying the number and orientation of intercalation events, as in Figure 5. Reviewer 4 gives detailed suggestions on how this could be done.

5) There are a number of questions related to statistical analysis and the description of statistical methods in the manuscript. Mixed-model statistics needs to be explained. Experimental variance (e.g., confidence intervals) needs to be documented.

6) The description of experimental results should be removed from the Discussion section.

Reviewer #1:

This manuscript provides a comprehensive analysis of cellular behavior during the formation of a tubular epithelial organ, using the embryonic salivary gland of *Drosophila* as a model. Budding of tubular structures from flat epithelial sheets has previously been thought to be mainly driven by apical constriction, leading to the formation of an invagination pit. Here the authors used live imaging and extensive quantitative analyses to demonstrate that, in addition to apical constriction, salivary gland invagination is associated with radially oriented intercalation of cells surrounding the invagination pit. Importantly, the authors extended their analysis to 3D, which revealed additional cellular behaviors (wedging, tilting, interleaving) that are oriented either circumferentially or radially with respect to the invagination center. Interestingly, polarized cell intercalation is associated with accumulation of myosin II along circumferential junctions, suggesting that myosin activity is polarized radially across the placode and contributes to tube invagination. Finally, the authors show that in *fkh* mutants, which fail to form an invagination, intercalation still takes place, but intercalating cell clusters fails to resolve in a directional manner. Here, cause and consequence are not clear – do polarized cell behaviors 'drive' tube budding, as stated in the title, or are they, at least in part, a consequence of forces generated by the invaginating pit that are transmitted across the field of cells?

This work reveals important new insights into the cellular behaviors underlying tube formation and contributes significantly to a better mechanistic understanding of this fundamental process. The findings are original and are very likely to be of general relevance, as the cellular behaviors and topologies described here in *Drosophila* closely resemble those found in developing organs of other animals, including vertebrates.

The work is substantial, of very high standard and thoroughly described, although the description of image analysis methodology and statistics is not readily accessible to the non-specialist reader and makes some sections of the text difficult to read.

The manuscript is likely to be of significant interest to a broad audience of cell and developmental biologists. Overall, this is a strong piece of work that is appropriate for publication in *eLife*. However, the authors need to address the following points.

It is not easily conceivable how a symmetric tube can result from the invagination of cells at an asymmetrically (peripherally) placed invagination pit. What happens with the cells located between the pit and the margin of the placode (upper-right part of placode; Figure 1E)? Do radially polarized cell behaviors spread beyond the boundary of the placode, and do cells outside the placode participate in invagination?

Introduction section: "*forkhead* mutants, that fail to form an invagination, only show unproductive intercalations that fail to resolve directionally, likely due to the lack of an active pit."

This sentence seems to contradict the statement in the title that radially-polarised cell behaviours "drive" tube budding. Does the invagination pit drive the polarized intercalation of nearby cells, or vice versa? The issue of cause and consequence needs to be clarified and discussed in the text. The wording of the title may need to be adjusted accordingly.

Results, subsection “3D tissue analysis at two depth shows coordination of cell behaviours in depth”: "Comparing apical and basal strain rates at the cell and tissue level with respect to their radial and circumferential contributions revealed an interesting picture. In temporally resolved plots, isotropic cell constriction dominated apically in cells near the pit (Figure 3E', magenta), but with a slower rate of constriction at mid-basal depth."

Isn't this precisely what is to be expected if cells near the pit undergo apical constriction (which was known before)?

Figure 6 E, F, and the accompanying text, appear unnecessarily complicated. The text refers to *circumferential* myosin enrichment, whereas Figure 6E, F and the legend refer to *radial* bi- or unipolarity of myosin distribution, respectively. Please simplify.

Is the distribution of myosin shown in Figure 6D a schematic drawing or representative of a real image?

Figure legends often lack information that is necessary to understand the data. Axis labels (e.g., "pp per min", "fluorescence increase over embryo average") need to be explained in the figure legends.

Reviewer #2:

This is an interesting manuscript by Sanchez-Corrales et al. that explores the cell shape changes that underlie the invagination of the salivary gland placode. The authors show that a combination of two common tissue shaping events, apical constriction and cell intercalation, function during early invagination. They also interpret their results in a radial coordinate system, which is nicely appropriate to the tissue context, and observe different topology-altering regimens dominating in different proximal-distal regions of the coordinate axis. Finally, they examine these cell behaviors at two different levels along the apical-basal axis as well as in a mutant that affects specification of the salivary placode. In general, these are interesting measurements and provide insight into tube and salivary placode invagination, although the impact of it is slightly hurt by an inability to tease apart the relative functional contributions of each regimen to invagination (i.e., it is not possible to alternately remove cell intercalation or apical constriction from the tissue) and the fact that the role of apical constriction in placode invagination is well-established. The figures are very nicely laid out, and, in some cases, quite beautiful. Their data does broadly support the interpretations presented in the paper. The attention paid to orienting schematics is appreciated and nicely done. The authors are very well-read and knowledgeable about morphogenesis, and carefully build and extend on the work from other systems, while the observation of circumferential intercalation is very nice. I would give a mild recommendation for publication, but will be interested to see the evaluations of other reviewers (which also partly goes to a level of confusion of what level of impact *eLife* requires). Apologies in advance if data was missed that explains any of the below.

Some issues to be addressed:

I'm not sure that "3D" analysis is accurate – this is 2D analysis done at a fairly limited two different planes (0µm and -8µm) in the tissue. It would be more accurate to say that this is an examination of apical and more basal contributions to cell shape and intercalation. Also, can the authors say whether cell volumes are maintained during invagination? It is a bit strange that similar constrictive behaviors are observed at -8µm; at some point there should be a cellular accommodation of the volume shifted due to apical constriction – either through a widening or deepening of the cell. There is a passing reference to cell deepening, but no hard data is presented. The authors should elaborate on this in the Results and Discussion. The analysis of cell wedging and the relationship to apical constriction and cell intercalation is quite interesting, although it takes significant effort to follow. Any editing and addition to the Discussion on these points will be helpful.

N numbers for the number of cells quantified in each category (for example, "near pit" and "far pit" cells) should be reported in each figure. The current reporting of the number of embryos should be kept.

Much of the statistical analysis is calculated through a mixed-effects model. More information on how mixed-effects models were applied to the data is needed to be able to evaluate the appropriateness of this statistical measurement. There should be at least a brief methods section on this.

Was there a statistical reason for splitting cells into "near" and "far" bins? I didn't see a clear statistical justification for this.

I was somewhat troubled by the inferred "intercalation strain rate" in the first sections of the manuscript, as this is a rather indirect measurement of topological changes that can be concretely measured, but they subsequently do exactly this in later portions of the manuscript.

There is an unofficial "results" section embedded into the Discussion. These should be moved to the results, or alternately, saved for a future publication. This section is a bit incongruous, given the Results sections, as it jumps into a cell fate discussion in the Discussion. The cell fate results are weakened by being correlative, without a functional component to test the hypothesis. An advantage of saving these results for another publication would be the ability to analyze such functional disruptions.

Discussion section penultimate paragraph – I am curious which "non-intuitive" results the authors are referring to?

Reviewer #3:

The study analyzes cellular mechanisms underlying initial events in salivary gland invagination, when a flat patch of cells invaginates to form a tube. Invagination is initiated by apical constriction of a few pit cells. The surrounding cells follow suit. Although the role of apical constriction has been reported previously, the cellular behaviour of the surrounding cells and their function for tube formation has not been dissected.

1) The authors report cell intercalation specifically towards the invagination site and propose that this may contribute to tube budding.

2) The authors aim to establish a three-dimensional view of the process by introducing a second focal layer (mid-basal) in addition to the apical view of the cells.

3) The authors report a polarized distribution of myosin II towards the site of invagination.

4) A mutant (*forkhead*) in which the cells are not specified displays only isotropic cell behaviour and myosin II distribution.

The authors conclude that a radially polarised pattern of cell behaviour, including apical constriction (as previously reported) and cell intercalation drives tube formation. The authors compare this pattern of cell behaviour with a radial pattern of myosin II accumulation and conclude that the radial pattern of myosin II would lead to radially polarised cell behaviour which would drive tube budding.

In my view the data do not allow to derive a causality. I accept that in wild type embryos the polarized myosin II pattern matches globally the preference of cell intercalation events. To address causality, the authors analyse *forkhead* mutants which completely lack tube invagination because cell specification is disrupted. It does not come as a surprise that cell and myosin II behave isotropically. The authors do not address the possibility that the radially polarized pattern of myosin II and cell behaviour is a mere consequence of mechanical pulling by the constricting and invaginating pit cells. *forkhead* mutants to not allow to rule out this option, as no invagination is observed in these mutants. The headline is thus an unjustified overstatement, as the radially polarized cell behaviour is likely to be a mere consequence of the activity of the small group of pit cells and certainly do not drive tube budding.

Specific issues:

In many figures the experimental variance is not visible. I understand that the authors tested the statistical significance. Yet, any sort of representation of the variance (e. g. confidence intervals) would help to assess the quality and the degree of uniformity of the data.

In the first part of the study, the degree of cell intercalation is only indirectly derived from the data as difference of total tissue behaviour and fitted cell shape changes. In Figure 5, the number and orientation of intercalation events is directly counted. The data would be more convincing, if also in the first part the contribution of intercalation would be directly measured. Given that the contribution of intercalation is the central conclusion of the study, it seems to be important that this parameter is directly measured.

3D: This part of the study is not convincing. I am fully aware of the difficulties and importance of a three-dimensional view of cell behaviour. Adding a second layer of recording does not contribute much to the issue however. In my view the presented data are not convincing and, in the end, even weaken the central conclusions. The authors conduct a sort of strain analysis along the axial direction, however with only two data points along the axis (apical and mid-basal).

Time axis: I find it confusing that the specific cell behaviour starts already in negative time. I understand that T=0 is set to the first visible invagination. As the polarized intercalation starts already at t=-15min, it would help to correlate the time axis to events at the time. Is this the first time when apical constrictions are observed?

Reviewer #4:

The paper offers interesting insights into tissue morphogenesis. Even though the single mechanical processes underlying this morphogenesis are not new, it is interesting to see this combination for a radial system. For the image analysis elegant and powerful methods were used. However, the principles underlying these analyses are not common knowledge among biologists. Therefore, extra care should be taken for the presentation of the results in order to make sure that biologists can still understand them. The current presentation is insufficiently intuitive and should be clearly improved. This should also allow readers to better judge the quality of the tracking. In addition, because of the simplicity of the system, the analysis method should be slightly simplified as control and also to contribute to better understanding.

Major points (better presentation and minor simplification) in detail:

1) When reading the manuscript, it becomes immediately clear that cell areas decrease at the pit. However, it is much less clear what happens to the regions further away. In addition, currently, it is not easy to see what happens with single cells. This hinders getting an intuitive picture of what happens to shapes of single cells and it does not become clear what the quality of the tracking was. Therefore, movies should be added that look similar to Figure 7C and D but with different coloring. In one movie, it should become visible what happens to the shapes of the radial stripes (Figure 1H’) and, in another one, what happens to individual cells.

2) The relative simplicity of the system should be exploited more to generate additional controls of the method, to simplify the understanding of the results, and possibly to obtain more precise outcomes. Currently, there is a focus on average strain rates and their accumulation. However, since the system has no apoptosis or division and relatively small strain changes, it should, at least for cellular strain rates, be possible to calculate total changes in strain directly for each cell, by comparing shapes in a certain time step directly with those in the first time step. For calculating the change in area, this would be very straightforward. It seems as if the cumulative area change is now calculated indirectly (but it is not clearly described whether this is indeed the case): ellipses are fitted to cells in such a way that the ellipses have the same areas as the cells. Then for each cell a strain rate tensor is obtained that is constrained to conserve the cell's area change, as assessed by the tensor's trace, which is a first order approximation of area change. Then the cumulative area change is obtained by adding up the traces, thus approximating total relative area change (1-0.1-0.1 is not the same as 1*0.9*0.9). Relative area change can also be obtained by only comparing areas at the first time step with those at a later time step. The authors should compare these values as a control of the method. In addition, they should use the direct method for area change, since it is easier to understand and a reader doesn't have to wonder for example why the cumulative area change is in pp per minute and not just in pp in Figure 1H. In addition, the total relative area change is actually the biologically relevant value in my opinion. Now it should not be a problem to do something similar for the strain change of the cells: here the fitted ellipse of a cell at a certain time step can be directly compared with the fitted ellipse of the same cell at the beginning and this should be used as a control. For calculating tissue shape change, the situation is a bit more complicated, since neighbors, and thus calculations, change during the time sequence. In order to do a direct comparison, the same cells should be compared at the beginning and the end. This should in principle be possible though, as long as the neighboring cells in the beginning (mostly) stay together. If other cells mix too much with the initial cells, definitions of strain do not really make sense any more. The authors should judge whether the direct approach is useful for tissue shape change. If it is, it would be useful to replace cumulative strain rates by these values, since they are a bit more straightforward.

3) The presentation of the figures should be clearly improved. Generally speaking, figures are currently missing that give a clear and intuitive overview of what is happening where in the tissue. In addition, tissue shape change, cell shape change, and cell intercalation are currently visualized by coloring according to strain rates, which is not a very intuitive quantity for many biologists. Instead, the figures should be such that readers can make a direct connection between quantitative values and shapes.

List of detailed suggestions to improve figures:

– Figures 1F and 3C are not very intuitive. First, a longer line is usually not associated with contraction. This should be changed. For example, arrows could be used, or the line length could be 1 as a standard and then be shortened or lengthened based on contraction and expansion, respectively. Secondly, the figure shows cell shapes, even though the strain rates are calculated based on a cell and its neighbors. It would therefore be better to make the cell outlines grey or something. Third, as far as I understand it, instantaneous strain rates are shown, so that there is much noise in the results and looking at the single figures may therefore not give much information. This can be improved by taking the total strain change (see point 2 of main points) or otherwise the cumulative strain rate, which should thus be calculated per cell.

– At the moment, Figure 1 does not clearly show what happens to the shapes of the radial stripes, even though this would be useful to better understand the movements in the tissue. This should be improved. One way could be to replace Figure 1G. This figure gives a good impression about time and position dependent cell shape changes, but may be replaced by another one that contains this information together with information at tissue level: one could for example replace it by a figure that contains a segmented image at the beginning, at t=0 and at the end. The boundaries between the radial stripes are thicker, so that it can be seen directly how their stripe shapes change. Cell shapes are then directly visible and can thus be assessed directly. However, in order to stress differences in area and compensate for interpretation difficulties due to the 2D projection, it would then still be useful to color individual cells according to area change. If desirable, cell aspect ratios could be color coded in additional images.

– The interpretation of Figure 2C, Figure 2—figure supplement 1B, Figure 3D, Figure 3—figure supplement 1B, and Figure 6—figure supplement 1A and B is not straightforward. The reader has to look up what the different colors mean and then couple that to the small inlet, which indicates whether the radial or circumferential direction is at play and then read the type of change that the figure shows. Even though the colors are nice to recognize patterns quickly, they don't confer any intuition on the extent of strain. In order to get such an idea, the data in the color bar need to be combined with reading the text to find out what kind of strain is visualized exactly and what the squares mean exactly. The authors should make these figures more intuitive. For example, they could use a segmented image of the end of a video. Each cell could then have arrows indicating strain changed in the radial (or circumferential) direction. Again, the total strain change (or the cumulative strain rate) can be used. In this way, it is immediately clear whether radial or circumferential strains are visualized and they allow for a direct more quantitative comparison between cells. In order to see quickly what is happening where, cells can be colored according to strain changes similarly to how the squares are colored now. Because of the presence of the arrows, it is then also clearer which color codes what. Depending on what the figure looks like exactly in the end, it may be useful to add a segmented image of the start as well and add radial stripe boundaries, so that the shape changes of single cells and their environment can also be looked at. In this way, shape changes of a cell and its neighbors could be directly compared to the length of the arrows and thus create an intuition for strain change. This would of course only show data of one embryo, but the quantification is in other figures anyway.

– Figure 4 does not give an intuitive overview of where which shape changes are present between the two layers. This should be improved. For example, two segmented z-projections could be shown in different colors in one figure. Then the reader could directly look at shape changes between the levels and see whether he can distinguish the wedging, interleaving and tilt.

– Figure 5 does not give any intuitive overview of where which neighbor exchanges occur. Such an overview should be added. For example, segmented images of the beginning and the end of a movie can be shown. Cell boundaries that will disappear may for example be drawn grey in the first image. New cell boundaries may for example be drawn blue in the image of the end of the movie. To get an idea of the number of reversals, each boundary that disappeared and then appeared again, could get another color.

[Editors’ note: formal revisions were requested, following approval of the authors’ plan of action.]

Thank you very much for your response, which was considered by the handling Senior Editor, the Reviewing Editor and the original reviewers. We are in general satisfied with your response and therefore would like to invite you to proceed with revisions. Please note that a major point that still remains to be resolved is the issue of causality – in the revised manuscript, any statements, including the title, implying that polarised cell behaviours cause ("drive") tube invagination will need to be carefully rephrased and toned down accordingly.

[Editors' note: further revisions were requested prior to acceptance, as described below.]

Thank you for resubmitting your work entitled "Radially-patterned cell behaviours during tube budding from an epithelium" for further consideration at *eLife*. Your revised article has been favorably evaluated by Didier Stainier (Senior Editor), a Reviewing Editor, and three reviewers.

The manuscript has been improved but there are some remaining issues that need to be addressed before acceptance, as outlined below:

In the Abstract and in the manuscript text, there are several remaining instances of misleading statements (*italicized*) inferring a causative role of polarised cell behaviours in tube budding, where no such causative role is supported by experimental evidence:

Last sentence of the Abstract: "Thus, tube budding involves radially-patterned pools of apical myosin, medial as well as junctional, *leading to* radially-patterned 3D-cell behaviours."

Introduction section: "In addition, across the placode junctional myosin II is enriched in circumferential junctions *leading to* polarised initiation of cell intercalation through junction shrinkage."

Subsection “Analysis of signatures of active versus passive cell intercalation”: "*This shows that* circumferential junctions, and T1s in particular, are likely to be *driven by* an intrinsic contractile mechanism, and are not the result of cells being pulled away from each other radially."

Discussion section: "in our system they are utilised within a radial coordinate system, thereby *leading to* the formation of a narrow tube of epithelial cells from a round and flat placode primordium."

These and corresponding claims in the text need to be carefully adjusted and toned down accordingly, so as to avoid misleading implications that polarised cell behaviours are a main driving force of invagination.

---

## [Author Response]

[Editors’ note: what follows is the authors’ plan to address the revisions.]

The full reviews are also included for your reference, as they contain detailed and useful suggestions. As you will see, the reviewers clearly find interest in the description of radially polarized cell behaviors during tube invagination. However, a number of issued and concerns are raised, a major one regarding the causal relationship between radially organized cell behavior and pit invagination.The reviewers raised the following main points:1) Causality. Cause and consequence are not clear with respect to the role of polarized cell behaviors in tube budding. Radially polarized intercalation of cells surrounding the pit may be, at least in part, a consequence of forces generated by the invaginating pit, rather than vice versa. Additional experiments would be required to dissect pit formation on the one hand and peripheral intercalation and myosin accumulation on the other hand. For instance, optogenetic tools might be used to trigger apical constriction and an ectopic invagination pit. This may allow to test whether intercalation behavior and myosin pattern respond to these changes, supporting a mechanical model, or if they do not respond, supporting intrinsic genetic control. If experimental evidence supporting such a causal role cannot be provided, claims about causality (including in the title) need to be diminished accordingly.

We would firstly like to apologise as apparently our phrasing of the title led to an interpretation or focus on an aspect of our findings that we did not intend. Our title was meant to refer to a radial patterning of [levels of] different cell behaviours across the placode, with isotropic constriction dominating in the domain near the pit, and oriented cell intercalation dominating in the region further away from the pit, as we illustrate in the schematic in Figure 8G. We did not mean to imply that the polarised cell intercalations that we describe are the main driving force of the invagination nor that the forces generated by the invagination pit do not play a role.

Second, to address directly the point raised, that the observed intercalation could be either a passive consequence of pit contraction or be intrinsically active, we propose to compare these two mechanisms comprehensively. The mechanisms lead to different predictions about cell behaviours, cell geometries and myosin behaviour (see Author response image 1). As we detail in the ‘action plan’ below, we have already tested one prediction, but propose to test a further three, that we anticipate will considerably improve this aspect of the paper.

In the ‘Active’ mechanism, polarised intercalation far from the pit arises from ‘*intrinsic genetic control*’ and actively drives circumferential contraction of the tissue (Author response image 1). In the ‘Passive’ mechanism, the active apical constriction near the pit drives a passive ‘funneling’ of far cells towards the pit, the radial pull and circumferential crowding compression leading to passive polarised intercalations (Author response image 1). This latter mechanism is what Reviewer 3 suggests with “the radially polarized cell behaviour is likely to be a mere consequence of the activity of the small group of pit cells”.

**Author response image 1. respfig1:** Active versus Passive mechanisms driving intercalation.

Before detailing specific predictions that allow us to distinguish between these mechanisms, we need to consider what we expect to see if placode morphogenesis is an overlay of both rather than one single mechanism. The Active mechanism could interact with apical constriction at the pit, the latter helping to pull out and thereby orient the resolution of T1s. Indeed, we believe this is likely to be the case, pending further confirmation from analyses we propose below. If both Active and Passive mechanisms are equally strong they could cancel, leaving no detectable signature of either. However, we think it likely that one will predominate, in which case we will detect its signature, even if this has been attenuated by the other mechanism.

We think it instructive to consider overlapping forces in another tissue earlier in *Drosophila* embryogenesis. Germband extension results from a combination of active polarized intercalation (set up by the AP-patterning system, work from Lecuit & Zallen labs) and an extrinsic pulling force from the invaginating posterior midgut (Lye et al., 2015; Collinet et al., 2015). These forces are analogous to the Active and Passive mechanisms in the placode, respectively, and in the same relative orientations. In the germband, the signature of both of these forces is detectable in cell geometries, myosin organization and strain rate gradients. We will further develop the discussion and Figure 8H along the above lines, pending what we find in proposed analyses below, and we will discuss in relation to the *fkh* mutant in which there is no invagination and cell neighbour exchange still occurs but it is not directional (see next section).

We believe four lines of evidence will allow us to distinguish which mechanism predominates:

Prediction 1

*Orientation of junctional myosin II*. According to the Active mechanism, we expect junctional myosin to be polarized, located along circumferentially oriented junctions, whereas if intercalation occurs as a passive response, we would expect either no myosin polarisation or mechanically induced localization in the perpendicular orientation (Author response image 1). Therefore, the strong polarisation of junctional myosin along circumferential junctions that we have documented in the paper (Figure 6) strongly supports the Active mechanism.

In response to a point raised by Reviewer 3 (“the possibility that the radially polarized pattern of myosin II […] is a mere consequence of mechanical pulling”), to our knowledge only two studies have analysed myosin subcellular localisation in response to mechanical pulling. These are a very recent study using the *Drosophila* wing disc epithelium subjected to mechanical pulling (Duda et al., 2018) and an earlier study of supracellular actomyosin cables during germband extension (Fernandes-Gonzales et al., 2009). In both cases, myosin accumulates and becomes polarised at junctions parallel to the pulling force. Extrapolating to our placode system, we would therefore expect an accumulation of myosin at radially oriented junctions if this is due to mechanical pulling, when we observe the opposite. Circumferentially polarized junctional myosin thus supports the Active mechanism.

Prediction 2

*Cell elongation.* The Active mechanism predicts cells will have a preferential elongation in the circumferential orientation, drawn out by myosin-driven junction contraction, while the Passive mechanism predicts radial elongation, pulled from the pit (Author response image 1). To establish the orientation of cell elongation, we first calculate best-fit ellipses to cell shapes, constraining the area of the ellipse to be the same as the cell area. Then we project the ellipse shape onto the radial axes to get radial and circumferential cell lengths. The strength of radial elongation is then the log-ratio of the radial to circumferential cell lengths, which can be positive (radially) or negative (circumferentially elongated). We will test whether on average cell elongation is significantly different from zero and in which orientation.

Prediction 3

*Angles at vertices* (following Rauzi et al., 2008). The Active mechanism predicts that angles opposite circumferentially oriented junctions will on average be more acute than 90 degrees, stretched by the contracting junctions in between (Author response image 1). Conversely, the Passive mechanism predicts less acute angles as cells are stretched towards the pit.

Prediction 4

*Junction lengths compared to ‘relaxed’ Voronoi lengths*. To distinguish between the junctions that are shortening actively from those that may shorten passively, we will use an inference method to probe geometric stress. We assume that a Voronoi tessellation based on cell centroid locations represents a mechanically neutral configuration for the cell-cell junctions. We will compare real interface lengths with interface lengths predicted by a Voronoi tessellation to extract a length deviation from the Voronoi tessellation, a geometric proxy for local stress (Tetley et al., 2015). Circumferential junctions that are shorter than expected are likely to have been actively shortened by local myosin activity whereas longer junctions are a likely result of cells being stretched towards the pit and subsequently rearranging passively (Author response image 1).

Finally, you suggest using optogenic methods to induce an ectopic pit. We have performed experiments along those lines and have tried the optogenetic tool developed by the de Renzis lab at EMBL to modify myosin activity (Guglielmi et al., 2015). However, this system has the inherent problem that two imaging channels are required for the optogenetic tool, thus we cannot simultaneously image a membrane channel and myosin channel to make our morphometric measurements. We also have tried several other experimental approaches with the purpose of inducing an ectopic constriction point. We have expressed a number of factors that should lead to increased myosin activity (such as an activated form of myosin or the catalytic domain of the myosin-activator Rho-kinase) in a stripe that covers the anterior part of the placode (i.e. opposite the forming pit, the domain where engrailed is expressed, using enGal4 for transgene overexpression). So far, none of these conditions has allowed us to genetically induce strong constriction. Therefore, although the idea of induction of an ectopic pit is tempting, technically this seems to still be impossible with current tools.

Moreover, in the scenario where we would be able to induce a second focus of invagination, and we could not detect a shift in the direction of intercalation, we have no means to rule out the possibility that the ectopic pit did not constrict as strongly as the original pit or that the tissue reached a new equilibrium to buffer an ectopic force. In our view, a quantitative assessment of the relative forces in our tissue deserves a full analysis, for example with various physical manipulations and force inference, which are beyond the scope of this work.

Action plan:

1) Change the title to make our interpretation clearer, using for instance “Radially-patterned cell behaviours drive tube budding from an epithelium”.

2) Explicit introduction of the Active and Passive mechanisms and their predictions (as shown here in Author response image 1) to current Figure 4.

3) Extend our analysis to test Predictions 2-4 listed above.

4) Re-write the manuscript to explain the mechanisms, predictions and new analysis results.

5) Elaborate more in the discussion about the interaction between the Active and Passive mechanisms, how they are synergistic but with distinct signatures, and allude to a combination in germband extension that has some similarities

2) The analysis of forkhead mutants is interesting, as it reveals cell behavior in the absence of an invagination pit. However, it contributes little, if anything, to the resolution of issues related to causality. This should be discussed and made clearer in the text.

We will adjust the results and discussion of *fkh* mutant in light of the two mechanisms proposed above (Author response image 1).

As the reviewers appreciated, the *fkh* mutant still shows intercalations occurring. Although the myosin polarisation is dramatically reduced in the *fkh* mutant, we could detect that the formation of rosettes still correlates with increased junctional myosin at the central junction of the forming rosette, as in the wild-type (example shown in Figure 8C). This finding thus supports an ‘Active’ mechanism in which cell intercalation is an active behaviour, rather than a passive response to activity elsewhere in the tissue.

Moreover, after testing our predictions (Author response image 1), we will contrast our findings in WT and fkh mutant within a radially organised tissue to an axially patterned tissue. In germband extension, an Active mechanism of polarised intercalation combines with an extrinsic pulling force from the invaginating posterior midgut that helps the directional resolution of T1s (Lye et al., 2015; Collinet et al., 2015). Our analysis in the fkh mutant shows that the productive directional neighbour exchange is dramatically decreased when an invaginating pit is not present (Figure 8E-F). Thus, an active intercalation in our tissue could combine with an apical constriction at the pit, the latter helping to pull out the resolution of T1s.

Another recent example of the role of forces to bias rosette resolution comes from the mouse embryo (Wen et al., 2017). Using micropipette aspiration to create an anisotropic force, the authors show that this ectopic stress is sufficient to rescue the resolution of T1s and rosettes in a mutant defective in intercalation (*Fgfr2*). Thus, the discussion of *fkh* mutant in this context will point towards a more generic mechanism of active intercalation combined with forces within the developing embryo to bias the direction of neighbour exchange resolutions.

Action plan:

1) Discuss our findings in *fkh* mutant in relation to possible Active and Passive mechanisms, comparing our findings to germband extension that has an analogous overlay of forces.

3) The analysis of cell behaviors is not truly "3D", as stated in the manuscript, but in fact cell shape is analyzed in 2D at two different levels along the apical-basal axis of the cells. An analysis of shape changes in 3D, and in particular of cell volume, would be more informative and would qualify as a "3D" analysis.

We completely agree that a true 3D analysis would be preferable. We therefore stated in the manuscript: *“*To circumvent this issue, we used strain rate analysis at different depth *as a proxy for a full 3D analysis”.* As we hoped to explain in the manuscript, obtaining a full 3D analysis is still very unreliable (apart from hand-segmentation, which is not feasible – one movie tracing 80 cells over 20 sections in depth and 40 time points would require drawing and linking 64,000 cell outlines by hand). We have tested various available segmentation methods, some of which allow 3D segmentation (such as EDGE4D [Khan et al., 2014, Development]; and RACE [Stegmaier et al., 2016, Dev. Cell]), but these do not track cells in 3D reliably. If unreliable segmentation leads to too much loss of cell identities over time, then our computational methods break down for the strain rate analysis and the data analysed will be too sparse. Therefore, we decided that a 3D proxy as used in our study currently allows much better segmentation and tracking coverage and thus far better strain rate analysis (both in time and in z). We are happy to emphasise even further that our methods are a 3D proxy, or any other term that reflects this approximation better.

It is important to note that our work would thus far be the first published method allowing quantification of changes in cell wedging, cell interleaving or cell tilting as 3D cell behaviours. Adding a second layer (with both layers parallel to the curved epithelial surface) and especially in combination with linking identities of cells between these layers as done in Figure 4 allowed us to approximate changes in 3D cell shape such as cell wedging, cell interleaving and cell tilting. None of these behaviours are obvious or could be deduced from an apical analysis only: apical area shrinkage could well be compensated for by cell elongation in depth, requiring analysis in a different plane; cell interleaving (a change in neighbour contacts along the z-length of the cell) cannot be observed from a single plane, neither can cell tilting. Importantly, we show in Figure 4 that these three changes to 3D cell shape (even if only approximated by the analysis at two z-levels) is clearly radially patterned across the tissue. We will further integrate these results within the framework described here in Author response image 1.

Action plan:

1) Modify the text to emphasise that our method is a 3D approximation.

2) Include in the summary Figure 8G the 3D signatures revealed in our analysis.

4) In the first part of the work, cell intercalation is inferred only indirectly from tissue behavior across a field of cells. Even though such measures are interesting to assess the relative contribution of cell shape and intercalation, the data would be more convincing if data are visualized per cell and if cell intercalation was assessed directly by quantifying the number and orientation of intercalation events, as in Figure 5. Reviewer 4 gives detailed suggestions on how this could be done.

We would first like to point out that the term intercalation is not restricted to one particular measure. The process of intercalation can be measured in two ways, either as the continuous process of slippage of cells past each other (Blanchard et al., 2009; Kabla et al., 2010; Blanchard, 2017) that is captured by our strain rate analysis, or as discrete T1 topological change events. These are equally valid, though can be interestingly different. For example, looking at Figure 4E and 4F from Blanchard (2017) (PMC5379022 https://www.ncbi.nlm.nih.gov/pmc/articles/PMC5379022/figure/RSTB20150513F4/), though the cumulative amount of intercalation measured by both methods during germband extension in *Drosophila* is the same, the time-courses are quite different, with the continuous process starting some 5-10 minutes before T1 events occur, as would be expected from starting from an approximately hexagonal arrangement of cells. One could argue that the continuous process of slippage is a more faithful measure of the underlying myosin-based contractility that drives intercalation than the counting of topological T1 events.

Since in our tissue there is no contribution to tissue strain due to either cell divisions or cell death/cell delaminations, the simple equation of cell shape change plus intercalation equals tissue shape change holds true (Blanchard et al., 2009). So we would not agree that the intercalation strain rate is somehow less real for being ‘inferred’, rather it is precisely calculated as a difference between two directly measurable quantities.

Importantly, our direct measurement and analysis of T1 events in the placode as presented in Figures 5 and 6 closely matches the strain rate analysis. Specifically, the cumulative strain over our observed time window that is attributed to the continuous process of intercalation is 0.1 (Figure 2E) while it is 0.09 for discrete T1s (Figure 8F). This strongly supports the validity and complementarity of both approaches.

Action plan:

1) Highlight in the text that the intercalation strain rates, calculated either as a difference of two directly measured quantities (Figure 2 in the main text) and as amount of neighbour exchange (Figure 5 and 6), are comparable.

In addition:

We thank Reviewer 4 for very helpful suggestions on the presentation of the strain rate data. We propose making the following changes:

1) We will change all cumulative plots to strains, which means they will all start at 1.0 and deviate over time from 1, at the end of the study period showing the proportion of its original size the tissue has reached (due to a particular behaviour). This could be more intuitive, as the reviewer suggests. In Author response image 2 is a version of Figure 2D’ (near pit cells, circumferential strain) in this new representation so that you can see how cumulative panels will look. Note that one aspect of this new presentation that will be different is that a halving of size (to 0.5) and a doubling of size (to 2.0) will not look symmetrical on these new graphs, whereas they do so in our current presentation.

**Author response image 2. respfig2:** New cumulative strain graph to be used.

2) A number of the reviewer’s comments refer to visualising and comparing cells or rings of cells at the start versus at the end of our study period. We agree that this will indeed help to show the changes that occur more intuitively so we will present figure panels for a representative embryo in the various ways suggested (showing rings of near and far from pit cells related to Figure 1H, comparing cell areas and shapes, cumulative strains in the different cell behavioural flavours, overlaying apical and mid-basal cell outlines, colour-coding interfaces that have been involved in a T1). Note however that we are presenting data from multiple placodes and our analysis and conclusions rely on the robustness we get from including as much data as we can. So we will keep the existing multi-placode summaries, though we will now add a detailed case study of one placode.

3) For this case study, we will show that the cumulative strain rates do indeed agree with a comparison of beginning and end sizes. We can do this for cell shape, as the reviewer suggests, but note that the strain rate calculation methods we use rely on small changes per time step of less than 20% strain (Blanchard et al., 2009) and we chose the imaging frame rate to ensure this was the case. It might therefore be difficult to do the same start to end measurement for tissue and intercalation strains.

4) We will try for Figures 1F and 3C to present these as cumulative strains across a movie and see how they look. If differences and patterns are sufficiently clear we will present these as suggested, though in their current form they are useful in showing a snapshot of the instantaneous dynamics, which would otherwise not be shown.

5) There are a number of questions related to statistical analysis and the description of statistical methods in the manuscript. Mixed-model statistics needs to be explained. Experimental variance (e.g., confidence intervals) needs to be documented.

We constructed mixed-effects models to test for differences between regions (far and near pit), directions (radial and circumferential) and genotypes (wild type and *forkhead* mutant) from multi-embryo time-lapse movies.

A mixed-effect model has fixed effects and random effects. The former are variables associated with an entire population and are expected to have an effect on the dependent variable. Random effects are factors which are associated with individual experimental units drawn at random from a population and introduce variation that is desirable to account for (Pinheiro JC and Bates D. (2000) Mixed effects models in S and SPLUS. Springer New York). In our work, to test for differences in instantaneous strain rates we used as fixed effect the genotypes (wild type and *forkhead*) while the variation within-embryo (from the same genotype) was considered a random effect.

The advantage of using a linear mixed-effect model is that this framework allows testing for differences on certain variables while accounting for sources of variation that are present in the full data set. A simpler alternative would have been to pool all data from different embryos of the same genotype, then testing for genotype differences. However, as some of the biologist reviewers will know well, inter-embryo variability in morphogenetic movements cannot be ignored, such as variation in the absolute and relative timings of the onset of mechanical processes and their subsequent morphological effects.

In the main figures we generally present cumulative strains, that are generated from instantaneous strain rate data shown in Supplemental Figures. The cumulative strains are calculated only from instantaneous data averages and so do not carry any of the distributions that were used to calculate instantaneous means and associated confidence intervals. We therefore do all statistics on the instantaneous data, using the mixed-effects model above. For mixed-effects models, one cannot portray a single overall confidence interval. Instead, we have a choice to show one of within- or between-genotype confidence intervals, and we have chosen the former, as has been done previously (for example in Butler et al., 2009; Gorfinkiel et al., 2009; Tetley et al., 2015; Lye et al., 2014).

Action plan:

1) Extend the section in the Materials and methods to explain mixed-model methods, and refer to this more frequently in the text and figure legends.

Reference not cited in the paper:

Kabla AJ, Blanchard GB, Adams RJ, Mahadevan L: Bridging cell and tissue behaviour in embryo development. In *Cell mechanics: from single scale-based models to multiscale modeling*. Edited by Chauviere A, Preziosi L, Verdier C: Chapman & Hall/CRC Mathematical & Computational Biology; 2010. DOI: https://dx.doi.org/10.1098/rstb.2015.0513

[Editors’ notes: the authors’ response after being formally invited to submit a revised submission follows.]

Reviewer #1:This manuscript provides a comprehensive analysis of cellular behavior during the formation of a tubular epithelial organ, using the embryonic salivary gland of Drosophila as a model. Budding of tubular structures from flat epithelial sheets has previously been thought to be mainly driven by apical constriction, leading to the formation of an invagination pit. Here the authors used live imaging and extensive quantitative analyses to demonstrate that, in addition to apical constriction, salivary gland invagination is associated with radially oriented intercalation of cells surrounding the invagination pit. Importantly, the authors extended their analysis to 3D, which revealed additional cellular behaviors (wedging, tilting, interleaving) that are oriented either circumferentially or radially with respect to the invagination center. Interestingly, polarized cell intercalation is associated with accumulation of myosin II along circumferential junctions, suggesting that myosin activity is polarized radially across the placode and contributes to tube invagination. Finally, the authors show that in fkh mutants, which fail to form an invagination, intercalation still takes place, but intercalating cell clusters fails to resolve in a directional manner. Here, cause and consequence are not clear – do polarized cell behaviors 'drive' tube budding, as stated in the title, or are they, at least in part, a consequence of forces generated by the invaginating pit that are transmitted across the field of cells?

As discussed in our initial response, the phrasing of our original title led to an interpretation or focus on an aspect of our findings that we did not intend. Our title was meant to refer to a radial patterning of [levels of] different cell behaviours across the placode, with isotropic constriction dominating in the domain near the pit, and oriented cell intercalation dominating in the region further away from the pit, now illustrated in the new summary Figure 10. We did not mean to imply that the polarised cell intercalations that we describe are the main driving force of the invagination nor that the forces generated by the invagination pit do not play a role (see below).

We have changed the title to now say:

‘Radially-patterned cell behaviours during tube budding from an epithelium’

and hope that this describes the findings more accurately.

As suggested in our initial response, we have now added a description and assessment of an active versus a passive mechanism responsible for the cell intercalation observed in the cells far from the pit. The detailed introduction of these mechanisms and associated predictions can now be found in subsection “Analysis of signatures of active versus passive cell intercalation” of the manuscript and illustrated in the new Figure 6. This figure also contains the data of the analysis of active versus passive signatures (together with Figure 7 (previous Figure 6)). This analysis of geometrical indicators of tissue stress as well as the analysis of junctional myosin polarisation clearly support that intercalations leading to circumferential neighbour gains in the cells far from the pit are actively initiated and actively lead to vertex formation. We cannot at this moment distinguish whether the resolution of exchanges is also intrinsic and actively driven or responds to the pulling from the pit. The continuing intercalation without directed resolution in the *fkh* mutant hint at an extrinsic (passive) component to the resolution, and the analogy to the process of germband extension would predict the same: active initiation of intercalation combines with the pulling from the pit to lead to an overall polarised intercalation event that underlies the circumferential convergence and radial extension at the tissue level. We have also added a short section in the Discussion about why the active signatures in the placode are less strong than in the germband.

These findings and conclusions are now clearly stated in the manuscript and illustrated in Figures 6, 7, and 10.

It is not easily conceivable how a symmetric tube can result from the invagination of cells at an asymmetrically (peripherally) placed invagination pit.

We would suggest that the cell intercalations as well as 3D behaviours we describe are important to achieve this. Although this question is not specifically addressed in this current study, it is of course an important one: why does the placode have an acentral invagination point and is this important for wild-type morphogenesis? The answer to the latter is clearly yes, as previous studies have shown that for instance mutants that commence invagination in the centre of the placode show aberrant tube morphologies and diametres at later stages (Myat and Andrew, 2000a; Myat and Andrew, 2002).

The circumferential convergence far from the pit combined with radial expansion towards the pit is one aspect of how cells move towards the acentral invagination point. Once internalised, previous studies have inferred that further intercalation/convergence and extension must take place within the formed tube as the number of cells around the tube circumference decreases from 12 to about 8 during the later stages of morphogenesis, way beyond the time frame analysed here.

What happens with the cells located between the pit and the margin of the placode (upper-right part of placode; Figure 1E)?

Again, this together with the above question about the acentral invagination point probably warrants a complete additional study. Our preliminary observations, however, suggest that these dorsal posterior cells at the boundary of the placode and right next to the invaginating pit initially remain static and in their original position, but over time these cells also invaginate and move away from the placode boundary. It would be interesting to analyse the behaviour of these cells in the future.

Do radially polarized cell behaviors spread beyond the boundary of the placode, and do cells outside the placode participate in invagination?

No, the whole placode is surrounded by a supracellular actomyosin cable (Röper, 2012, Dev. Cell) that decreases in diameter as the placodal cells disappear from the surface of the embryo.

Introduction section: "forkhead mutants, that fail to form an invagination, only show unproductive intercalations that fail to resolve directionally, likely due to the lack of an active pit."This sentence seems to contradict the statement in the title that radially-polarised cell behaviours "drive" tube budding. Does the invagination pit drive the polarized intercalation of nearby cells, or vice versa? The issue of cause and consequence needs to be clarified and discussed in the text. The wording of the title may need to be adjusted accordingly.

Please see the answer to Reviewer 1 above.

Subsection “3D tissue analysis at two depth shows coordination of cell behaviours in depth”: "Comparing apical and basal strain rates at the cell and tissue level with respect to their radial and circumferential contributions revealed an interesting picture. In temporally resolved plots, isotropic cell constriction dominated apically in cells near the pit (Figure 3E', magenta), but with a slower rate of constriction at mid-basal depth."Isn't this precisely what is to be expected if cells near the pit undergo apical constriction (which was known before)?

Yes, but so far this had never been analysed and quantified dynamically from time-lapse movies and in a 3D proxy analysis. But we agree, this just confirmed an aspect we were expecting.

Figure 6 E, F, and the accompanying text, appear unnecessarily complicated. The text refers to circumferential myosin enrichment, whereas Figure 6E, F and the legend refer to radial bi- or unipolarity of myosin distribution, respectively. Please simplify.

What we are plotting in panels D-F of revised Figure 7 (previously 6) are the vectors that are pointing towards to myosin II enrichment, and by definition these will be orthogonal to the orientation of the junction the myosin is enriched in. The paper describing this method of quantifying and representing myosin junctional polarisation was published in *eLife* previously (Tetley et al., 2016, *eLife*) and we have just kept the conventions as in this paper. We have tried to illustrate clearly what is plotted with the diagrams in Figure 7D’ and D’’.

To simplify or rather clarify, we have now changed the label in panels E, F to say ‘circumferential enrichment (rad. polarity vector)’ as well as ‘radial enrichment (circ. polarity vector)’ to make the plots more intuitive but also retain the correct nomenclature for what is measured and plotted in the method, and have also updated the figure legend accordingly.

Is the distribution of myosin shown in Figure 6D a schematic drawing or representative of a real image?

It is a schematic (drawn from real cell shapes), but representative of real images such as Fgure 7A, G, and H.

Figure legends often lack information that is necessary to understand the data. Axis labels (e.g., "pp per min", "fluorescence increase over embryo average") need to be explained in the figure legends.

The labels were changed and/or explained more clearly in the legend.

Reviewer #2:This is an interesting manuscript by Sanchez-Corrales et al. that explores the cell shape changes that underlie the invagination of the salivary gland placode. The authors show that a combination of two common tissue shaping events, apical constriction and cell intercalation, function during early invagination. They also interpret their results in a radial coordinate system, which is nicely appropriate to the tissue context, and observe different topology-altering regimens dominating in different proximal-distal regions of the coordinate axis. Finally, they examine these cell behaviors at two different levels along the apical-basal axis as well as in a mutant that affects specification of the salivary placode. In general, these are interesting measurements and provide insight into tube and salivary placode invagination, although the impact of it is slightly hurt by an inability to tease apart the relative functional contributions of each regimen to invagination (i.e., it is not possible to alternately remove cell intercalation or apical constriction from the tissue) and the fact that the role of apical constriction in placode invagination is well-established.

Yes, we agree with the reviewer that apical constriction was previously well documented in the salivary gland placode, including by our own work (Booth et al., 2014). But thus far, the analysis of apical constriction was always only focused on the apical domain exclusively, or on a single 2D cross section of tissue in histological sections. Indeed, the Sun et al., (2017) paper revealing basal protrusions in the germband were something of a surprise for those who thought the germband was apically driven. Therefore, we would posit that our dynamic and 3D-proxy analysis, clearly showing strong wedging behaviour near the pit that is restricted to this region and increases over time (even prior to actual tissue-bending) adds to the previous descriptions. Furthermore, the novelty of our data lies in the identification of radially patterned behaviours 2D and 3D, the contribution of directed intercalations in 2D, interleaving, wedging and tilting in 3D and the analysis of active/passive signatures. For this much more in depth analysis of placode tube morphogenesis the apical constriction near the pit, albeit known before, was obviously an important factor that needed to be part of a more complete analysis.

As the reviewer will be well aware, in most morphogenetic processes, it is difficult to selectively affect only part of all actomyosin-dependent processes (i.e. constriction or intercalation), where this has been achieved it is through use of fortuitous mutants that mostly impair one but not the other process. We would argue that in our case the use of the *fkh* mutant achieved this in part: in the mutant, apical constriction is basically absent at early stages, whereas we show that intercalations occur at near wild-type levels but lack the polarised resolution.

Some issues to be addressed:I'm not sure that "3D" analysis is accurate – this is 2D analysis done at a fairly limited two different planes (0µm and -8µm) in the tissue. It would be more accurate to say that this is an examination of apical and more basal contributions to cell shape and intercalation. Also, can the authors say whether cell volumes are maintained during invagination?

As explained in our initial response, we completely agree that a true 3D analysis would be preferable. We already stated in the original manuscript *“*To circumvent this issue, we used strain rate analysis at different *depth as a proxy for a full 3D analysis*”, and have now made sure that this phrasing is used wherever we refer to the method.As we hoped to explain in the manuscript, obtaining a full 3D analysis is still very unreliable (apart from hand-segmentation, which is not feasible – one movie tracing 80 cells over 20 sections in depth and 40 time points would require drawing and linking 64,000 cell outlines by hand). We have tested various available segmentation methods, some of which allow 3D segmentation (such as EDGE4D [Khan et al., 2014, Development]; and RACE [Stegmaier et al., 2016, Dev. Cell]), but these do not track cells in 3D reliably. If unreliable segmentation leads to too much loss of cell identities over time, then our computational methods break down for the strain rate analysis and the data analysed will be too sparse. Therefore, we decided that a 3D proxy as used in our study currently allows much better segmentation and tracking coverage and thus far better strain rate analysis (both in time and in z).

It is important to stress again that our work would thus far be the first published method allowing quantification of changes in cell wedging, cell interleaving or cell tilting as 3D cell behaviours. Adding a second layer (with both layers parallel to the curved epithelial surface) and especially in combination with linking identities of cells between these layers as done in Figure 4 allowed us to *quantify approximate* changes in 3D cell shape such as cell wedging, cell interleaving and cell tilting. Indeed, even if full 3D shapes of all cells were known, the entry-level analysis one would do would likely start with simplifying cell shapes by fitting straight lateral sides to all cells (through regression or similar). The elegance of our approach is that we have effectively constructed this same simplified description of cells in a very efficient manner. Because we only sample at two depths, apically and mid-basally, our measures are likely to be more variable than if detailed 3D cells shapes were known. However, we would like to point out that changes in topology in depth seem remarkably smooth, as can be seen in Video 4 which progressively steps through z, so that sparse sampling in z will in fact capture much of the important 3D information.

None of the 3D geometries are obvious or could be deduced from an apical analysis only: apical area shrinkage could well be compensated for by cell elongation in depth, requiring analysis in a different plane; cell interleaving (a change in neighbour contacts along the z-length of the cell) cannot be observed from a single plane, neither can cell tilting. Importantly, we show in Figure 4 that these three changes to 3D cell shape (even if only approximated by the analysis at two z-levels) is clearly radially patterned across the tissue.

With regards to cell volume, we have analysed this from a single fully hand-traced movie and indeed cell volume appears to be conserved, and we now mention this as ‘data not shown’.

It is a bit strange that similar constrictive behaviors are observed at -8µm; at some point there should be a cellular accommodation of the volume shifted due to apical constriction – either through a widening or deepening of the cell.

In fact, the cell shape strain rate basally is significantly reduced compared to apically (Figure 3D), already indicating wedging behaviour, and the direct analysis of wedging in the 3D proxy analysis using z-strain rates confirms this (Figure 4B’). But the reviewer is correct in that cells are expanded even further more basally, at a depth that we cannot reliably segment and track. This was commented on and explained in the manuscript (Results, subsection “A quasi-3D tissue analysis at two depths shows coordination of cell behaviours in depth”).

The fact that intercalation strain rates are very similar apically and basally (Figure 3E’), especially in the cells away from the pit, prompted the 3D analysis in order to dissect whether apical or basal levels tipped ahead of each other (i.e. were interleaved) despite a nearly identical rate.

There is a passing reference to cell deepening, but no hard data is presented. The authors should elaborate on this in the Results and Discussion.

The slight deepening of the whole placode compared to the surrounding epidermis and even more so in the region of the forming pit is visible in the placodal cross-section in Figure 1B, and we point this out in the section of the manuscript commented on above under 3). This deepening has been described from fixed samples and histological sections by the Andrew lab in already in 2000 (Myat and Andrew, 2000a).

The role of the initial slight columnarisation of the placodal cells compared to the surrounding epidermis is not clear and deserves to be analysed in a future study.

The analysis of cell wedging and the relationship to apical constriction and cell intercalation is quite interesting, although it takes significant effort to follow. Any editing and addition to the Discussion on these points will be helpful.

Together with our introduction of possible active or passive mechanisms responsible for the intercalation away from the pit, we have significantly changed and restructured parts of the Result section and Discussion concerned with these two behaviours and their potential interplay. We hope this has made these sections clearer.

N numbers for the number of cells quantified in each category (for example, "near pit" and "far pit" cells) should be reported in each figure. The current reporting of the number of embryos should be kept.

We have added the total number of cells analysed per time-step of the analysis for the different regions, apical/basal, wt/*fkh[6]* to the Materials and methods section. We felt that adding these lists to the Figure legends would make these confusing.

Much of the statistical analysis is calculated through a mixed-effects model. More information on how mixed-effects models were applied to the data is needed to be able to evaluate the appropriateness of this statistical measurement. There should be at least a brief methods section on this.

Mixed effect models were previously discussed in the Materials and methods section, but we have now expanded the section, which describes the rationale for the use of mixed effect models. We have also clarified legend text where statistics are presented.

Was there a statistical reason for splitting cells into "near" and "far" bins? I didn't see a clear statistical justification for this.

We were prompted to use this split into near and far domains by our early analysis, as for instance shown in Figure 1G, G’, where cumulative apical area change of concentric rings around the pit clearly shows that only the cells near the pit (magenta and blue) are actively constricting at a steady rate, whereas cells far from the pit (green, red, yellow) change little or even slightly expand.

The apical strain rate analysis reported in Figure 2 and Figure 2—figure supplement 1also shows that only when cells are split into these two region can specific cell behaviours be identified as dominant in a region, whereas binning all placodal cells (as in Figure 2—figure supplement 1C, C’) has effects across the tissue cancel each other.

These regions thus distinguish dynamics that were otherwise masked when the tissue was considered as a whole. We cannot rule out the possibility that there are more than two regions within the placode (defined for instance by differential gene expression) and future studies will need to address this.

I was somewhat troubled by the inferred "intercalation strain rate" in the first sections of the manuscript, as this is a rather indirect measurement of topological changes that can be concretely measured, but they subsequently do exactly this in later portions of the manuscript.

As already mentioned in our initial response, we would like to point out that the term intercalation is not restricted to one particular measure. The process of intercalation can be measured in two ways, either as the continuous process of slippage of cells past each other (Blanchard et al., 2009; Kabla et al., 2010; Blanchard, 2017) that is captured by our strain rate analysis, or as discrete T1 topological change events. These are equally valid, though can be interestingly different. For example, looking at Figure 4E, F from Blanchard (2017) (PMC5379022 https://www.ncbi.nlm.nih.gov/pmc/articles/PMC5379022/figure/RSTB20150513F4/), though the cumulative amount of intercalation measured by both methods during germband extension in *Drosophila* is the very similar, the time-courses are quite different, with the continuous process starting some 5-10 minutes before T1 events occur, as would be expected from starting from an approximately hexagonal arrangement of cells. We would argue that the continuous process of slippage is a more faithful measure of the underlying myosin-based contractility that drives intercalation than the counting of topological T1 events.

Since in our tissue there is no contribution to tissue strain due to either cell divisions or cell death/cell delaminations, the simple equation of cell shape change plus intercalation equals tissue shape change holds true (Blanchard et al., 2009). So we would not agree that the intercalation strain rate is somehow less real for being ‘inferred’, rather it is precisely calculated as a difference between two directly measurable quantities.

Importantly, our direct measurement and analysis of T1 events in the placode as presented in Figures 5 and 6 closely matches the strain rate analysis. Specifically, the cumulative strain over our observed time window that is attributed to the continuous process of intercalation is 1.1 (Figure 2E) while it is *e*^0.09^ = 1.094 for discrete T1s (Figure 9F). This strongly supports the validity and complementarity of both approaches.

There is an unofficial "results" section embedded into the Discussion. These should be moved to the Results, or alternately, saved for a future publication. This section is a bit incongruous, given the Results sections, as it jumps into a cell fate discussion in the Discussion. The cell fate results are weakened by being correlative, without a functional component to test the hypothesis. An advantage of saving these results for another publication would be the ability to analyze such functional disruptions.

Thanks, yes, we have removed this section and part of the figure for future use.

Discussion section penultimate paragraph – I am curious which "non-intuitive" results the authors are referring to?

We have found cell interleaving to be a non-intuitive 3D geometry of multi-cellular domains when explained to people for the first time, particularly in comparison to wedging and tilting which can be seen directly for individual cells. Once understood, of course, it becomes intuitive, but we agree it is not a helpful term, so we have removed it.

Reviewer #3:[...]The authors conclude that a radially polarised pattern of cell behaviour, including apical constriction (as previously reported) and cell intercalation drives tube formation. The authors compare this pattern of cell behaviour with a radial pattern of myosin II accumulation and conclude that the radial pattern of myosin II would lead to radially polarised cell behaviour which would drive tube budding.In my view the data do not allow to derive a causality. I accept that in wild type embryos the polarized myosin II pattern matches globally the preference of cell intercalation events. To address causality, the authors analyse forkhead mutants which completely lack tube invagination because cell specification is disrupted.

In the *fkh* mutant, not all cell specification is disrupted. The most upstream transcription factor driving salivary gland placode fate is Scr (in conjunction with Hth and Ext). In the *fkh* mutant, expression of CrebA (as a master activator of all aspects of increased secretory capacity of the glands) is normal, various other factors usually expressed within the placode are still expressed, though what is changed is the spatial pattern of expression of a number of downstream factors (i.e. being restricted to the dorsal-posterior corner at first and then expanding across the tissue).

It does not come as a surprise that cell and myosin II behave isotropically.

This is in fact only true over the time interval analysed here, at later stages the central part of the placode begins to buckle and form a very shallow indented invagination. Also, as visible in the myosin images and movie in the *fkh* mutant, the circumferential actomyosin cable is still specified and assembled, as is junctional myosin accumulation at the centre of groups of cells that still undergo intercalations (see the images of rosette formation in Figure 9). Thus, we would argue that what is lost in the *fkh* mutant is the sub-patterning of the placode with an acentrally positioned point of invagination.

As we discuss in detail now in the revised version, the fact that in the *fkh* mutant T1s still occur, at levels comparable to wild-type (though without directional coordination), strongly supports that the shortening of junctions to a T1 in the placode is indeed an active cell behaviour and not a passive response to other events within the placode.

The authors do not address the possibility that the radially polarized pattern of myosin II and cell behaviour is a mere consequence of mechanical pulling by the constricting and invaginating pit cells.

Please refer to our detailed response to Reviewer 1’s first point. We now introduce, and explicitly test active versus passive contributions.

With regards to myosin polarisation as a passive response, we discuss now in the manuscript that to our knowledge only two studies have analysed myosin subcellular localisation in response to mechanical pulling. These are a very recent study using the *Drosophila* wing disc epithelium subjected to mechanical pulling (Duda et al., 2018) and an earlier study of supracellular actomyosin cables during germband extension (Fernandes-Gonzales et al., 2009). In both cases, myosin accumulates and becomes polarised at junctions parallel to the pulling force. Extrapolating to our placode system, we would therefore expect an accumulation of myosin at radially oriented junctions if this is due to mechanical pulling, when we observe the opposite. As discussed in the revised manuscript, circumferentially polarized junctional myosin thus supports that intercalation is driven by an Active mechanism.

However, we also discuss that the mechanical pulling (due to apical constriction near the invagination point) and intercalation are integrated: polarised junctional actomyosin leads to polarised *formation* of a vertex or rosette, and the mechanical pulling helps to polarise the *resolution* of these events in a directional manner, similar to what has been found to occur during germband extension and posterior midgut invagination (Collinet et al., 2015).

forkhead mutants to not allow to rule out this option, as no invagination is observed in these mutants.

As rosette formation and T1 exchanges continue in the *fkh* mutant (albeit with non-directional resolutions) despite the lack of a pulling pit, we would argue that this supports cell intercalation as an active cell behaviour rather than a passive response, and we discuss this in detail in the revised manuscript.

The headline is thus an unjustified overstatement, as the radially polarized cell behaviour is likely to be a mere consequence of the activity of the small group of pit cells and certainly do not drive tube budding.

As we already pointed out in our initial response, we are sorry that the original title was misinterpreted and we have changed it accordingly.

Please see the detailed response to Reviewer 1 above.

Specific issues:In many figures the experimental variance is not visible. I understand that the authors tested the statistical significance. Yet, any sort of representation of the variance (e. g. confidence intervals) would help to assess the quality and the degree of uniformity of the data.

In the main figures we generally present cumulative strains, that are generated from instantaneous strain rate data shown in figure supplements. The cumulative strains are calculated only from instantaneous data averages and so do not carry any of the distributions that were used to calculate instantaneous means and associated confidence intervals. We therefore do all statistics on the instantaneous data, using the mixed-effects model. For mixed-effects models, one cannot portray a single overall confidence interval. Instead, we have a choice to show one of within- or between-genotype confidence intervals, and we have chosen the former, as has been done previously (for example in Butler et al., 2009; Gorfinkiel et al., 2009; Tetley et al., 2015; Lye et al., 2014).

The variance is therefore visible in the instantaneous strain rates in the figure supplements.

In the first part of the study, the degree of cell intercalation is only indirectly derived from the data as difference of total tissue behaviour and fitted cell shape changes. In Figure 5, the number and orientation of intercalation events is directly counted. The data would be more convincing, if also in the first part the contribution of intercalation would be directly measured. Given that the contribution of intercalation is the central conclusion of the study, it seems to be important that this parameter is directly measured.

Please see the detailed response to Reviewer 2’s point about intercalation strain rate above.

3D: This part of the study is not convincing. I am fully aware of the difficulties and importance of a three-dimensional view of cell behaviour. Adding a second layer of recording does not contribute much to the issue however. In my view the presented data are not convincing and, in the end, even weaken the central conclusions. The authors conduct a sort of strain analysis along the axial direction, however with only two data points along the axis (apical and mid-basal).

Please see our detailed response to Reviewer 2 about the practicalities and what we see as the strengths of our method. We are glad that the other reviewers found the 3D proxy analyses and results interesting. We hope that we have addressed the concerns of this reviewer in the revision and explained the advantages of our methods more succinctly.

Time axis: I find it confusing that the specific cell behaviour starts already in negative time. I understand that T=0 is set to the first visible invagination. As the polarized intercalation starts already at t=-15min, it would help to correlate the time axis to events at the time. Is this the first time when apical constrictions are observed?

As explained in the manuscript, all time lapse movies are aligned to the point in time where the first tissue bending (curvature at the tissue level) can be observed as this emerged as the most reproducible point for such alignment. We thus designated this point as t= 0, which seemed intuitive to us as it represents the first time point of actual invagination.

We agree that it is interesting that changes to cell shapes in the placode commence way before this point, though this was noted and commented on by us already previously (Girdler and Röper, 2014). It is though not completely surprising that changes in the planar epithelium star to occur and build up stresses prior to an out of plane bending of the epithelium. But to dissect the contributions of these pre-bending changes to the efficiency and robustness of the invagination process will be analysed in a further study.

With regards to the first time point analysed and what this relates to: as we state in the results for Figure 4, cells across the placode started out at -18 mins before pit invagination unwedged and mostly untilted in radial and circumferential orientations (Figure 4B’, F’). Cells near the pit became progressively wedge-shaped over the next 30 minutes. Therefore, the pooled beginning of our time lapse analysis appears to capture the earliest changes to placodal cells.

Reviewer #4:The paper offers interesting insights into tissue morphogenesis. Even though the single mechanical processes underlying this morphogenesis are not new, it is interesting to see this combination for a radial system.

To our knowledge, quantitative changes in cell wedging, cell interleaving and cell tilting as measure of 3D cell behaviour have not been reported previously. Certainly, our newly developed methods to asses these changes quantitatively from two or more z levels are novel and will be of general use in the epithelial morphogenesis field.

Furthermore, even within a 2D context, directed cell intercalation within the plane of the epithelial primordium had not been appreciated as a contributor to tube budding from an epithelium previously, to the best of our knowledge.

[…]Major points (better presentation and minor simplification) in detail:1) When reading the manuscript, it becomes immediately clear that cell areas decrease at the pit. However, it is much less clear what happens to the regions further away. In addition, currently, it is not easy to see what happens with single cells. This hinders getting an intuitive picture of what happens to shapes of single cells

To aid this understanding, we had provided various means in the original manuscript, such as videos showing cell outlines, the still time points of apical cell area for wild-type and *fkh* mutant in Figure 8C, D as well as the marked examples of rosette formation/resolutions in Figures 7 and 9.

We agree, though, that even more could be done or at least tested, and so we have followed the reviewer’s suggestions and detail what we have done below.

and it does not become clear what the quality of the tracking was.

The segmentation of all movies used in this study was manually corrected to ensure at least 90% of coverage of each placode at each time point. This statement was now added to the Materials and methods section on segmentation and tracking.

We also added an example movie for one segmented and tracked placode in Video 2 that shows cell identities colour-coded for the length of the movie.

Therefore, movies should be added that look similar to Figure 7C and D but with different coloring. In one movie, it should become visible what happens to the shapes of the radial stripes (Figure 1H’) and, in another one, what happens to individual cells.

We agree that this will help to show changes that occur more intuitively, and so we have generated panels and movies for better illustration.

1) We have added a new Figure 1—figure supplement 1 that shows cell shapes and areas of cells within radial stripes (as in Figure 1G, G’) for an example movie (ExpID0356)

2) We also added a new Video 2 in which each cell identity in the same example experiment (ExpID0356) is colour-coded for identity as well as for apical cell area.

3) We include a new panel in Figure 2—figure supplement 1F that shows the total intercalation strain rate over a whole example experiment (ExpID0356). We did this by calculating strain rates at the middle time point of the movie, over a much larger time window than previously (+/- 11 frames rather than +/-1 frame), covering the whole of the movie. This very useful panel addition should be considered as an indicative example only, since some of the strain rates calculated over this longer time window were above the ~20% strain limit recommended for the methods that we use (explained in Blanchard et al., 2009). Nevertheless, the radial organization of intercalation (convergence is biased strongly to circumferential) in the panel is striking.

4) See also response below to Reviewer 4 for visual representations of where T1s occur in the placode, and their orientation.

2) The relative simplicity of the system should be exploited more to generate additional controls of the method, to simplify the understanding of the results, and possibly to obtain more precise outcomes. Currently, there is a focus on average strain rates and their accumulation. However, since the system has no apoptosis or division and relatively small strain changes, it should, at least for cellular strain rates, be possible to calculate total changes in strain directly for each cell, by comparing shapes in a certain time step directly with those in the first time step. For calculating the change in area, this would be very straightforward. It seems as if the cumulative area change is now calculated indirectly (but it is not clearly described whether this is indeed the case): ellipses are fitted to cells in such a way that the ellipses have the same areas as the cells. Then for each cell a strain rate tensor is obtained that is constrained to conserve the cell's area change, as assessed by the tensor's trace, which is a first order approximation of area change. Then the cumulative area change is obtained by adding up the traces, thus approximating total relative area change (1-0.1-0.1 is not the same as 1*0.9*0.9). Relative area change can also be obtained by only comparing areas at the first time step with those at a later time step.

We thank the reviewer for very helpful suggestions on the presentation of the strain rate data. We have made the following changes:

We have changed all cumulative plots from proportional strains to strains (or stretch ratios), which means they will all start at 1.0 and deviate over time from 1, at the end of the study period showing the proportion of its original size the tissue has reached (due to a particular behaviour). This could be more intuitive, as the reviewer suggests.

With regards to calculating total strain for individual cell versus our use of average cumulative strains:

For the embryo case study (ExpID0356) we introduced in response the above point, we compare firstly the radial cell lengths of cells at the beginning and at the end of the movie (Author response image 3) to show the range of changes in size of cells over the whole movie (from t= -7.5 min to t= +20.0 min).

**Author response image 3. respfig3:** 

Author response image 3 show predicted (y-axis) versus actual (x-axis) radial cell lengths of cells, using two different cell shape strain rates to calculate the predicted length. For Author response image 3, the cell shape strain rate is the radial projection of the tensorial strain rate that mapped best-fit cell shape ellipses at the start and end of the movie, over a +/- time window of 13.75 mins (half the movie). For Author response image 3, the strain rate used for each cell is the average of the cell shape strain rates calculated for each of the 22 frames of the movie, where each strain is calculated over a smaller time window of +/- 1.25 mins (neighbouring movie frames).

The correlation in both cases is very strong. Note that, as explained for the above point, the strain rate calculation methods we use rely on small changes per time step of less than 20% strain (Blanchard et al., 2009) and we chose the imaging frame rate to ensure this was the case. Calculating start to end strain rate measurements as in Author response image 3 will therefore introduce small errors due to the larger strain rates in the larger time window, and this explains the minor discrepancies. There are also small imprecisions because the mapping of two ellipses onto each other is not perfect if they are elongated and their ellipse orientations are not aligned. The error though is expected to be and looks here unstructured (Gaussian) and so data averages will be accurate.

Thus, we strongly feel that our methodology correctly captures individual cell events in average cumulative strains. We hope that the change in depicting strains as well as the addition of further panels illustrating changes for an example placode will make the changes more intuitive to grasp.

The authors should compare these values as a control of the method. In addition, they should use the direct method for area change, since it is easier to understand and a reader doesn't have to wonder for example why the cumulative area change is in pp per minute and not just in pp in Figure 1H. In addition, the total relative area change is actually the biologically relevant value in my opinion. Now it should not be a problem to do something similar for the strain change of the cells: here the fitted ellipse of a cell at a certain time step can be directly compared with the fitted ellipse of the same cell at the beginning and this should be used as a control. For calculating tissue shape change, the situation is a bit more complicated, since neighbors, and thus calculations, change during the time sequence. In order to do a direct comparison, the same cells should be compared at the beginning and the end. This should in principle be possible though, as long as the neighboring cells in the beginning (mostly) stay together. If other cells mix too much with the initial cells, definitions of strain do not really make sense any more. The authors should judge whether the direct approach is useful for tissue shape change. If it is, it would be useful to replace cumulative strain rates by these values, since they are a bit more straightforward.

We thank the reviewer for these suggestions and have changed our cumulative strain rate plots to cumulative strains (or cumulative stretch ratios), which we agree are on balance more intuitive. (One downside is that the y-axis scale is now no longer symmetrical about the starting value. For example, for intercalation, which involves no area change, a doubling in length in one orientation (from 1 to 2) and a halving of length in the perpendicular orientation (from 1 to 0.5) are no longer symmetrical about the line y=1.)

3) The presentation of the figures should be clearly improved. Generally speaking, figures are currently missing that give a clear and intuitive overview of what is happening where in the tissue.

We have added a new summary Figure 10 that sums up the changes both for 2D cell behaviours observed within the apical domain only as well as the 3D cell behaviours that our 3D proxy analysis revealed. We hope the reviewer finds this a helpful addition.

In addition, tissue shape change, cell shape change, and cell intercalation are currently visualized by coloring according to strain rates, which is not a very intuitive quantity for many biologists. Instead, the figures should be such that readers can make a direct connection between quantitative values and shapes.

As for Reviewer 4’s point (2) above, the new cumulative strains on the y-axes are more intuitive in that one can immediately see the fractional change in size of the tissue attributable to each cell behavior over the course of developmental time.

List of detailed suggestions to improve figures:– Figures 1F and 3C are not very intuitive. First, a longer line is usually not associated with contraction. This should be changed.

These are only illustrative panels to explain the methodology used and use the convention to depict strain rates that is also used and explained in Figure 2B. The colour-coding of the original Figures 1F and 3C was not the same as for Figure 2B, but we have now changed this and they all use magenta to indicate expansion and green to indicate contraction. The length of the lines in these illustrative panels is proportional to the amount of strain, as explained in Figure 2B. We have also added the original small example domain of cells from which the illustrative strain rates in Figure 2B were calculated, in which vertical cell shortening and cell rearrangement leading to further convergence vertically, as expressed in the strain rate motifs, can clearly be seen.

For example, arrows could be used, or the line length could be 1 as a standard and then be shortened or lengthened based on contraction and expansion, respectively. Secondly, the figure shows cell shapes, even though the strain rates are calculated based on a cell and its neighbors. It would therefore be better to make the cell outlines grey or something.

We agree with the general concern here, but our preference, based on a long experience of displaying strain rates going back to 2009 and before is that the fewer the number of lines the better, for ease of digest while also showing detail. Other publications with strain rate nematics showing deviations from 1 or deviations from a circle of standard radius (e.g. by Johanns Bellaîche/Francois Graner & Suzanne Eaton/Frank Jülicher) need quite strong strains to be obviously different and are better suited for data that have been either heavily averaged over space or accumulated over long times.

As the reviewer states, strains are calculated for small domain, i.e. a central cell and its first corona of neighbours, but this strain is then associated with the identity of the central cell, and is drawn for that cell. We think it should be clear from the cartoons representing the strain rate methods in Figure 2A, B and Figure 4—figure supplement 1A-D that this is what we are doing throughout.

Third, as far as I understand it, instantaneous strain rates are shown, so that there is much noise in the results and looking at the single figures may therefore not give much information. This can be improved by taking the total strain change (see point 2 of main points) or otherwise the cumulative strain rate, which should thus be calculated per cell.

We have changed plots in main figures to cumulative strains as suggested.

– At the moment, Figure 1 does not clearly show what happens to the shapes of the radial stripes, even though this would be useful to better understand the movements in the tissue. This should be improved. One way could be to replace Figure 1G. This figure gives a good impression about time and position dependent cell shape changes, but may be replaced by another one that contains this information together with information at tissue level: one could for example replace it by a figure that contains a segmented image at the beginning, at t=0 and at the end. The boundaries between the radial stripes are thicker, so that it can be seen directly how their stripe shapes change. Cell shapes are then directly visible and can thus be assessed directly. However, in order to stress differences in area and compensate for interpretation difficulties due to the 2D projection, it would then still be useful to color individual cells according to area change. If desirable, cell aspect ratios could be color coded in additional images.

We hope that our new panel of the placode separated into 5 radial stripes in Figure 1—figure supplement 1, colour coded by cell area at the start and end of an example movie, addresses this point satisfactorily.

– The interpretation of Figure 2C, Figure 2—figure supplement 1B, Figure 3D, Figure 3—figure supplement 1B, and Figure 6—figure supplement 1A, B is not straightforward. The reader has to look up what the different colors mean and then couple that to the small inlet, which indicates whether the radial or circumferential direction is at play and then read the type of change that the figure shows. Even though the colors are nice to recognize patterns quickly, they don't confer any intuition on the extent of strain. In order to get such an idea, the data in the color bar need to be combined with reading the text to find out what kind of strain is visualized exactly and what the squares mean exactly. The authors should make these figures more intuitive. For example, they could use a segmented image of the end of a video. Each cell could then have arrows indicating strain changed in the radial (or circumferential) direction. Again, the total strain change (or the cumulative strain rate) can be used. In this way, it is immediately clear whether radial or circumferential strains are visualized and they allow for a direct more quantitative comparison between cells. In order to see quickly what is happening where, cells can be colored according to strain changes similarly to how the squares are colored now. Because of the presence of the arrows, it is then also clearer which color codes what. Depending on what the figure looks like exactly in the end, it may be useful to add a segmented image of the start as well and add radial stripe boundaries, so that the shape changes of single cells and their environment can also be looked at. In this way, shape changes of a cell and its neighbors could be directly compared to the length of the arrows and thus create an intuition for strain change. This would of course only show data of one embryo, but the quantification is in other figures anyway.

We found it difficult to improve these plots as suggested. We hope that a combination of more intuitive cumulative strains in the main figures, the addition of panels of snapshots from a single movie in various figures, and better clarity in the text in response to various helpful points made here and by other reviewers, will make it easier for the reader to interpret these plots.

– Figure 4 does not give an intuitive overview of where which shape changes are present between the two layers. This should be improved. For example, two segmented z-projections could be shown in different colors in one figure. Then the reader could directly look at shape changes between the levels and see whether he can distinguish the wedging, interleaving and tilt.

To explain better what type of changes the z strain analysis is measuring and how the methods works in comparison to single plane time-strain analysis, we have added cross section panels to the schematics explaining the 3D behaviours, and have also added a whole section to the Figure 4—figure supplement 1that explains the z-strain rate method in comparison to t-strain rate analysis (Figure 4—figure supplement 1A-D).

– Figure 5 does not give any intuitive overview of where which neighbor exchanges occur. Such an overview should be added. For example, segmented images of the beginning and the end of a movie can be shown. Cell boundaries that will disappear may for example be drawn grey in the first image. New cell boundaries may for example be drawn blue in the image of the end of the movie. To get an idea of the number of reversals, each boundary that disappeared and then appeared again, could get another color.

We have added two panels (Figure 5D) for an example movie that show the interfaces that will be lost over the duration of the movie coloured in blue and labeled on a segmented image of the first time frame, and the interfaces that were gained over the course of the movie labeled on a segmented image of the last time point coloured in red. In addition, a variant of this is shown in Figure 5—figure supplement 1, where future neighbour losses and past neighbour gains are plotted, this time colour-coded for radial and circumferential orientation.

[Editors' note: further revisions were requested prior to acceptance, as described below.]The manuscript has been improved but there are some remaining issues that need to be addressed before acceptance, as outlined below:In the Abstract and in the manuscript text, there are several remaining instances of misleading statements (italicised) inferring a causative role of polarised cell behaviours in tube budding, where no such causative role is supported by experimental evidence:

*Last sentence of the Abstract: "Thus, tube budding involves radially-patterned pools of apical myosin, medial as well as junctional,* leading to *radially-patterned 3D-cell behaviours."*

We changed the sentence to:

“Thus, tube budding involves radially-patterned pools of apical myosin, medial as well as junctional, and radially-patterned 3D-cell behaviours, with a close mechanical interplay between invagination and intercalation.”

*Introduction section: "In addition, across the placode junctional myosin II is enriched in circumferential junctions* leading to *polarised initiation of cell intercalation through junction shrinkage."*

We changed this sentence to:

“In addition, across the placode junctional myosin II is enriched in circumferential junctions, suggesting polarised initiation of cell intercalation through active junction shrinkage.”

*Subsection “Analysis of signatures of active versus passive cell intercalation”: "*This shows that *circumferential junctions, and T1s in particular, are likely to be* driven by *an intrinsic contractile mechanism, and are not the result of cells being pulled away from each other radially."*

We changed this sentence to:

“This suggests that circumferential junctions, and circumferential T1s in particular, were contracted, possibly by an intrinsic contractile mechanism, rather than the cells being pulled away from each other radially by the pit.“

*Discussion section: "in our system they are utilised within a radial coordinate system, thereby* leading to *the formation of a narrow tube of epithelial cells from a round and flat placode primordium."*

We changed the sentence to:

“…but in our system they are utilised within a radial coordinate system, with the morphogenetic outcome being the formation of a narrow tube of epithelial cells from a round and flat placode primordium.”

and, former sentence:

“This finding therefore also supports an active intercalation mechanism, where the remaining local increases in junctional myosin II still drive formation of T1 vertices and rosettes, but without any overall directionality to their resolution.”

changed to:

“This finding therefore also suggests an active intercalation mechanism, where the remaining local increases in junctional myosin II still support formation of T1 vertices and rosettes, but without any overall directionality to their resolution.”